# Radiative corrections and Monte Carlo tools for low-energy hadronic cross sections in $e^+e^-$ collisions

RadioMonteCarLow 2 Working Group: Riccardo Aliberti [1], Paolo Beltrame [2], Ettore Budassi [3,4], Carlo M. Carloni Calame [4], Gilberto Colangelo [5], Lorenzo Cotrozzi [2], Achim Denig [1], Anna Driutti [6,7], Tim Engel [8], Lois Flower [2,9], Andrea Gurgone [3,6,7], Martin Hoferichter [5], Fedor Ignatov [2], Sophie Kollatzsch [10,11], Bastian Kubis [12], Andrzej Kupść [13,14,*], Fabian Lange [11,10], Alberto Lusiani [15,7], Stefan E. Müller [16], Jérémy Paltrinieri [2], Pau Petit Rosàs [2], Fulvio Piccinini [4], Alan Price [17], Lorenzo Punzi [7,15], Marco Rocco [10,18], Olga Shekhovtsova [19,20], Andrzej Siódmok [17], Adrian Signer [10,11,*], Giovanni Stagnitto [21], Peter Stoffer [10,11], Thomas Teubner [2], William J. Torres Bobadilla [2], Francesco P. Ucci [3,4], Yannick Ulrich [2,5,*], and Graziano Venanzoni [2,7,*]

[1]Institute for Nuclear Physics, Johannes Gutenberg University Mainz, Germany
[2]University of Liverpool, Liverpool L69 3BX, U.K.
[3]Dipartimento di Fisica, Università di Pavia, Via A. Bassi 6, 27100, Pavia, Italy
[4]INFN, Sezione di Pavia, Via A. Bassi 6, 27100, Pavia, Italy
[5]Albert Einstein Center for Fundamental Physics, Institute for Theoretical Physics, University of Bern, Sidlerstrasse 5, 3012 Bern, Switzerland
[6]Dipartimento di Fisica, Università di Pisa, Largo B. Pontecorvo 3, 56127, Pisa, Italy
[7]INFN, Sezione di Pisa, Largo B. Pontecorvo 3, 56127, Pisa, Italy
[8]Albert-Ludwigs-Universität Freiburg, Physikalisches Institut, D-79104 Freiburg, Germany
[9]Institute for Particle Physics Phenomenology, Durham University, South Road, Durham, DH1 3LE, U.K.
[10]PSI Center for Neutron and Muon Sciences, 5232 Villigen PSI, Switzerland
[11]Physik-Institut, Universität Zürich, 8057 Zürich, Switzerland
[12]Helmholtz-Institut für Strahlen- und Kernphysik (Theorie) and Bethe Center for Theoretical Physics, Universität Bonn, 53115 Bonn, Germany
[13]Department of Physics and Astronomy, Uppsala University, Box 516, SE-75120 Uppsala, Sweden
[14]National Centre for Nuclear Research, Pasteura 7, 02-093 Warsaw, Poland
[15]Scuola Normale Superiore, Pisa, Italy
[16]HZDR Dresden, Germany
[17]Jagiellonian University, ul. prof. Stanisława Łojasiewicza 11, 30-348 Kraków, Poland
[18]Università degli Studi di Torino & INFN, Via Pietro Giuria 1, Torino 10125, Italy
[19]NSC KIPT Institute for Theoretical Physics, Kharkov, Ukraine
[20]INFN Sezione di Perugia, 06123, Perugia, Italy
[21]Università degli Studi di Milano-Bicocca & INFN, Piazza della Scienza 3, Milano 20126, Italy
*Coordinator and corresponding author

## Abstract

We present the results of Phase I of an ongoing review of Monte Carlo tools relevant for low-energy hadronic cross sections. This includes a detailed comparison of Monte Carlo codes for electron–positron scattering into a muon pair, pion pair, and electron pair, for scan and radiative-return experiments. After discussing the various approaches that are used and effects that are included, we show differential cross sections obtained with AfkQed, BabaYaga@NLO, KKMC, MCGPJ, McMule, Phokhara, and Sherpa, for scenarios that are inspired by experiments providing input for the dispersive evaluation of the hadronic vacuum polarisation.

# Contents

# 1 Introduction

Monte Carlo codes are essential tools for the analysis of low-energy scattering experiments at electron–positron colliders. Accordingly, there is an extensive and long-standing effort by the community to provide and improve codes that are able to produce fully differential predictions for processes related to $e^+ e^- \to$ hadrons at centre-of-mass energies up to a few GeV. Through the community effort described in this article we collect such tools and facilitate their access and usage. In addition, we present a comprehensive comparison of the physical effects included, methods used, and their likely effect on the accuracy of the theoretical predictions. We update the report of the *Working Group on Radiative Corrections and Monte Carlo Generators for Low Energy* [1] and highlight the developments of the past ten years. However, the report presented here is more focused. We restrict ourselves to the processes listed in (1.1) and (1.2), augmented by some remarks about $3\pi$ production and contrary to [1] do not consider issues related to luminosity measurements.

As in [1] we adopt the concept of *tuned comparisons* when presenting the results of different Monte Carlo programmes. Such tuned comparisons use the same set of input parameters and sometimes codes are compared without vacuum-polarisation corrections. These comparisons must also use the same experimental cuts. For the latter, we include realistic acceptance selection, while we refrain from taking into account additional kinematic selection (like kinematic fit) and detector effects. Correspondingly, we do not compare to experimental data at all. At this stage, this is a purely theoretical effort and the presented results require careful interpretation. We consider the current article to be Phase I of an ongoing community effort [2–5] and plan for a long-term continuation of the programme.

One of the main reasons to consider these low-energy processes is of course their impact on the determination of the hadronic vacuum polarisation (HVP) corrections $a_\mu^{\mathrm{HVP}}$ to the anomalous magnetic moment of the muon (also called muon $g-2$). We hope that a coordinated effort from the Monte Carlo community will help to shed light on possible shortcomings or future improvements of these codes and, hence, to their impact on data-driven calculations of the HVP contribution to the muon $g - 2$. Thus, there is a close link to the Muon $(g - 2)$ Theory Initiative [6]. However, there is also a justified interest in these processes as such. After all, the situation regarding the process $e^+ e^- \to \pi^+ \pi^-$, which provides the dominant contribution to $a_\mu^{\mathrm{HVP}}$, is very unclear [7].

There are large tensions among different experiments but also between most experiments and lattice results. A critical assessment of the Monte Carlo tools will be an important component for a better understanding of these fundamental issues.

There has been considerable progress in the evaluation of radiative corrections to scattering processes since [1]. New groups have entered the field, either providing new tools for low-energy scattering processes or maintaining and further developing existing codes. In this article we consider in detail AFKQED, BABAYAGA@NLO, KKMCEE, MCGPJ, MCMULE, PHOKHARA, and SHERPA. There are other codes that can play an important role for the processes under consideration and we encourage their future inclusion. The codes considered here use partly overlapping and partly complementary approaches. This offers a rich environment to obtain a better understanding of the applicability and reliability of different approximations.

The core purpose of this work is to assess the importance of various contributions in the theoretical description of fully differential cross sections. More concretely, in Phase I we are concerned with the $2 \to 2$ processes

$$e^+ e^- \to \pi^+ \pi^- \,, \tag{1.1a}$$

$$e^+ e^- \to \mu^+ \mu^- \,, \tag{1.1b}$$

$$e^+ e^- \to e^+ e^- \,, \tag{1.1c}$$

relevant for energy scan experiments and the $2 \to 3$ processes

$$e^+ e^- \to \pi^+ \pi^- \gamma \,, \tag{1.2a}$$

$$e^+ e^- \to \mu^+ \mu^- \gamma \,, \tag{1.2b}$$

$$e^+ e^- \to e^+ e^- \gamma \,, \tag{1.2c}$$

relevant for radiative-return experiments. We refer to (1.2) as radiative processes (sometimes they are also called initial-state radiation (ISR) processes) and it is understood that the photon is hard, i.e., it has sufficient energy to be detectable. Of course, the inclusion of higher-order corrections implies the consideration of additional – possibly soft – photons in the final state. For all six processes, we consider a collection of observables within several scenarios. The latter consist of particular centre-of-mass energies and acceptance cuts. Since we do not include detector effects nor additional kinematic selection, the results presented here are meant to illustrate a comparison between different theory approaches. They should not be used to compare directly with experimental data.

Our results can also serve as benchmark for future theory developments and as toolbox for experimental collaborations. In the spirit of open science, through [8]

https://radiomontecarlow2.gitlab.io

we make publicly available all source codes that have been used for this report. To ensure reproducibility, the exact configurations that have been used to obtain the results are also given. This includes the precise definition of the observables, the input parameters (usually through a run card), the Monte Carlo version, and analysis and plotting pipelines. It is foreseen that this repository will be updated continuously and possibly extended with additional Monte Carlo codes and additional processes. We hope this facilitates the use of the tools by experimental collaborations as well as the theory community.

This article is not meant to be a complete review of all activities related to low-energy hadronic cross sections. It is only meant to provide the foundation for a detailed analysis of the current theory status for the processes listed in (1.1) and (1.2) at energies up to $10\,\mathrm{GeV}$. Accordingly,

in Section 2 we start with a brief overview over the colliders and experiments that are most relevant for these processes, as well as for $3\pi$ production. The different computational approaches that are used for these processes are described in Section 3. This includes fixed-order QED calculations up to next-to-next-to-leading order (NNLO), approaches to include logarithmically enhanced higher-order terms through parton showers or soft-photon resummation, and the inclusion of non-perturbative hadronic effects due to insertions of vacuum polarisation (VP) or the hadronic light-by-light (HLbL) four-point function. The treatment of pions in the final state is rather delicate and, hence, receives particular attention. With these methods in hand, Section 4 provides a brief description of which contributions are included in the various Monte Carlo tools. Section 5 contains our main results. For five scenarios with different centre-of-mass energies and acceptance cuts we present detailed Monte Carlo comparisons. This is done for selected observables with electron, muon, and pion final states. A more comprehensive list of observables can be found in the repository [8]. If two codes include the same higher-order corrections, the comparisons are used for the technical validation of the codes. Where different approaches have been taken, the comparison is used to study the importance of particular components of higher-order corrections. An executive summary of our findings and an outlook towards further improvement of the codes is given in Section 6.

## 2   Experiments at electron–positron colliders

There are two classes of experiments at electron–positron colliders related to the processes (1.1) and (1.2). Energy scan experiments access cross sections of the processes (1.1) by setting the beam energies of a collider to a given centre-of-mass (c.m.) energy, $\sqrt{s}$, which can be achieved with good precision. The observed number of events for a given integrated luminosity is corrected for the acceptance and radiative effects due to, for example, soft photon emission and virtual corrections. To collect data at other $\sqrt{s}$ requires changes and adjustments of accelerator settings. Radiative-return (also called ISR) experiments use (1.2) to extract the energy dependence of the processes (1.1) with just a single working point $\sqrt{s}$ of the collider. The emission of a hard photon from initial state allows these experiments to collect events with continuously distributed invariant masses $w$ of the final-state system from threshold to the c.m. energy of the collider. The mass $w$ is determined by a measurement of the momenta of the final particles.

Many experiments have contributed to the measurements of the $e^+e^- \to \pi^+\pi^-$ channel over the years, as shown in Figure 1. Information about two-pion cross section measurements is collected in the database

https://precision-sm.github.io/2pi-db.

Table 1 gives a summary of the experiments included in the database including the references and links to the HEPData[1] records.

There are variations of the radiative-return experiments which have different sensitivity to the radiative corrections. The configurations adopted by the different experiments are shown in Table 2. All experiments fall into one of two categories regarding the process used for normalising the $e^+e^- \to \pi^+\pi^-$ cross section: either $e^+e^- \to e^+e^-$ is used, or $e^+e^- \to \mu^+\mu^-\gamma$. The table also details whether each experiment requires explicit detection of a photon in the calorimeter, which will impact the sensitivity to radiative corrections.

In the following we give an overview of the main experiments ordered by collider c.m. energy while for more details we refer to [1] and the literature.

---

[1]https://www.hepdata.net/

| Accelerator | Exp. | Year | References | HEPData |
|---|---|---|---|---|
| BEPC (Beijing) | BESIII | 2016 | [10] | ins1385603 |
| SLAC (Standford U.) | BaBar | 2012 | [11] | ins1114155 |
| CESR (Cornell U.) | CLEO | 2018 | [12] | ins1643020 |
| | | 2013 | [13] | ins1189656 |
| | | 2005 | [14] | ins693873 |
| DAPHNE (LNF) | KLOE | 2017 | [15] | |
| | | 2012 | [16] | |
| | | 2010 | [17] | ins859660 |
| | | 2008 | [18] | ins797438 |
| | | 2004 | [19] | ins655225 |
| Adone (LNF) | MEA | 1980 | [20] | ins158283 |
| | | 1977 | [21] | ins124109 |
| | BCF | 1975 | [22] | ins100180 |
| CERN | NA007 | 1984 | [23] | ins195944 |
| ACO (Orsay) | | 1976 | [24] | ins109771 |
| | | 1972 | [25] | ins73648 |
| DCI (Orsay) | DM2 | 1989 | [26] | ins267118 |
| | DM1 | 1978 | [27] | ins134061 |
| VEPP-2000 (Novosibirsk) | CMD-3 | 2023 | [28] | |
| | SND | 2021 | [29] | ins1789269 |
| VEPP-2M (Novosibirsk) | SND | 2005 | [30]; erratum: [31] | ins686349 |
| | CMD-2 | 2007 | [32] | ins728302 |
| | | 2006 | [33] | ins728191 |
| | | 2005 | [34] | ins712216 |
| | | 2002 | [35]; erratum: [36] | ins568807 |
| | OLYA | 1984 | [37] | ins221309-Table1 |
| | CMD | 1983 | [37] | ins221309-Table2 |
| | TOF | 1981 | [38] | ins167191 |
| VEPP-2 (Novosibirsk) | VEPP-2 | 1972 | [39] | ins75634 |
| | | 1971 | [40] | ins69313 |
| | | 1969 | [41] | ins57008 |
| | | 1967 | [42] | ins1392895 |

Table 1: Summary of the published $e^+e^- \to \pi^+\pi^-$ measurements with links to the datasets.

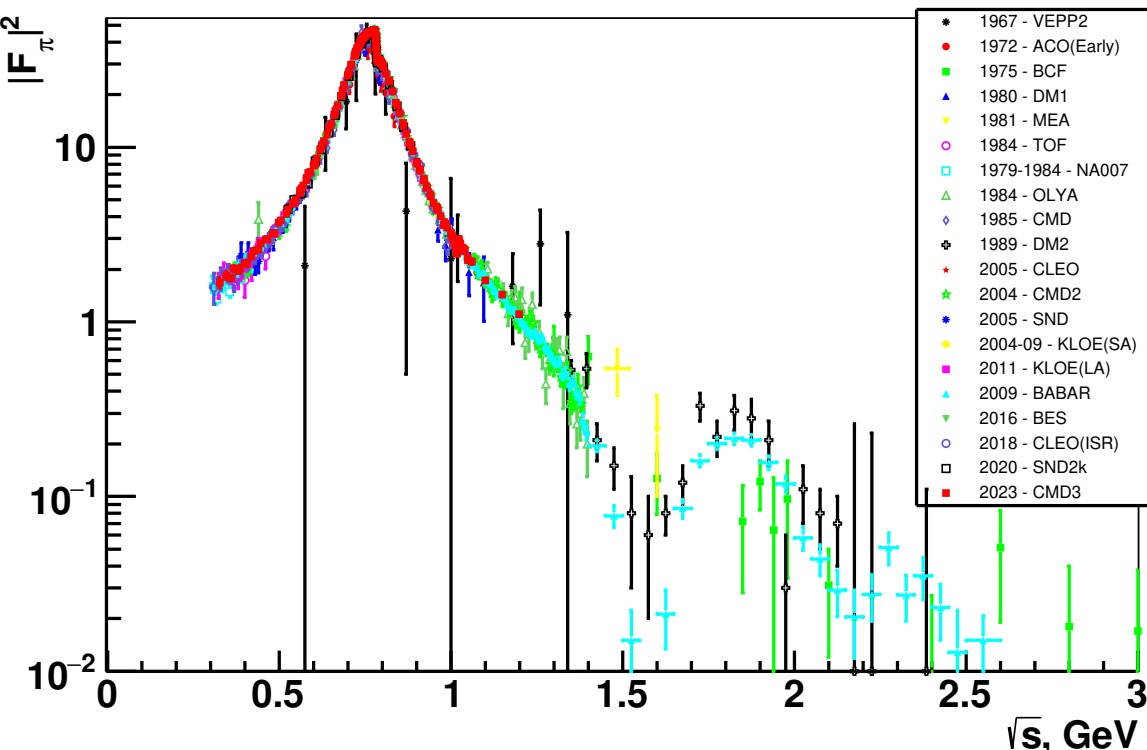

Figure 1: Status of the $e^+e^- \rightarrow \pi^+\pi^-$ cross-section measurement with different experiments contributing over the years. Plot updated from [9].

## 2.1 BINP, Novosibirsk: VEPP-2M,VEPP-2000 and SND, CMD-2, CMD-3

The VEPP-2M electron–positron collider in Novosibirsk [43, 44] has been in use for more than 25 years, starting from 1974, with several generations of detectors. The latest cycle of experiments from 1992 to 2000 was performed by the CMD-2 [45] and SND [46] detectors, installed in two interaction regions of the VEPP-2M collider. The peak luminosity reached by the collider was $3 \times 10^{30}$ cm$^{-2}$s$^{-1}$, and the two detectors together collected about 60 pb$^{-1}$ of the total integrated luminosity, covering the energy range from 0.36 GeV up to 1.38 GeV. Both detectors are general purpose detectors which include central tracking systems, calorimetry and additional auxiliary subsystems such as a muon veto. CMD-2 was operated in a 1 T magnetic field, while SND was non-magnetic and was not able to distinguish charges of particles. Instead, the SND detector has the advantage in detecting neutral modes, with a more comprehensive 3-layer spherical electromagnetic calorimeter. These scan experiments produced data on $e^+e^-$ annihilation to hadrons, covering the production from $\pi^0\gamma$, $\pi^+\pi^-$ to $4\pi$, and $K\bar{K}$ final states [47]. The two-pion measurements were based on about $1.1 \times 10^6$ and $4.5 \times 10^6$ selected $\pi^+\pi^-$ events by CMD-2 and SND, respectively. The achieved precision on the combined total hadronic cross section $R(s)$ was about 1% at c.m. energies around the $\rho$-resonance and 3.5% at 1.38 GeV. The systematic uncertainties in the main $e^+e^- \rightarrow \pi^+\pi^-$ channel were 0.6% for the CMD-2 measurement and 1.3% for SND. The statistics of the two experiments were limited and contributed almost as much as the systematic uncertainty to the overall accuracy of the dispersive integral for $a_\mu^{\text{had}}$. As a result of both experiments, the accuracy of the value $a_\mu^{\text{HVP}}$ at that time was improved by a factor of 3.

The VEPP-2M collider was decommissioned in 2000 to make way for the new storage ring VEPP-

2000 [48, 49], which started to provide data for new experiments in 2010. The machine covers the wider c.m. energy range from $\sqrt{s} = 0.32\,\text{GeV}$ to $2.0\,\text{GeV}$. It employs the novel technique of round beams to reach a luminosity of up to $10^{32}$ cm$^{-2}$s$^{-1}$ at $2\,\text{GeV}$, a world record of single bunch luminosity at these low energies. Today, VEPP-2000 is the only collider able to scan energies below $2\,\text{GeV}$ for the measurement of exclusive $e^+e^- \to$ hadrons channels. A special system based on the Compton backscattering of laser photons is used to measure the beam energy with relative accuracy better than $10^{-4}$ [50, 51]. Two new generation detectors, CMD-3 [52, 53] and SND [54], are installed at opposite interaction regions of the collider. A major upgrade of all subsystems was performed, including completely new modernised electronics and more elaborate triggers in comparison to previous experiments. For example, in the case of the CMD-3 detector, a new drift chamber, which provides higher efficiency and more than twice better the momentum resolution, and a new LXe calorimeter, with multi-layer tracking capabilities and shower profile measurement, were constructed.

The main goals of experiments at VEPP-2000 include the high-precision measurement of cross sections of various modes of $e^+e^- \to$ hadrons in the whole available c.m. energy range up to $2\,\text{GeV}$. All major channels are under analysis with final states of up to 7 pions, or 2 kaons and 3 pions [55]. Many results have already been published by the CMD-3 and SND experiments, but many more are still under analysis.

The most demanding final state, due to the required precision, is the $e^+e^- \to \pi^+\pi^-$ process. The first energy scan below $1\,\text{GeV}$ for the $\pi^+\pi^-$ measurement was performed at the VEPP-2000 collider in 2013, which collected an integrated luminosity of $17.8\,\text{pb}^{-1}$. In 2014–2016, there was a long shutdown for the collider and detector upgrades. In particular, a new electron and positron injector facility was commissioned, which allowed a significant increase in luminosity. The next energy scan in the $\rho$-meson c.m. energy region was carried out during the 2017–2018 data-taking season, when about $45.4\,\text{pb}^{-1}$ were collected. At the end of 2019, an additional $1\,\text{pb}^{-1}$ data sample was collected near the threshold region at c.m. energies $\sqrt{s} < 0.6\,\text{GeV}$. The latest scan below $1\,\text{GeV}$ was performed in the first half of 2024 in the energy range between the $\omega$ and $\phi$ resonances.

The first measurement of the $e^+e^- \to \pi^+\pi^-$ cross section at VEPP-2000 was presented by the SND experiment, covering the energy range $0.525 < \sqrt{s} < 0.883\,\text{GeV}$ with a systematic uncertainty of about 0.8% [29]. This result was based on partial data of the first $\rho$ scan, corresponding to about 10% of the total collected statistics below $1\,\text{GeV}$. The CMD-3 experiment has recently performed the full statistics analysis from 0.32 to $1.2\,\text{GeV}$ with the conservative estimation of systematic uncertainty 0.7% in the dominant $\rho$-resonance region [28, 56]. The CMD-3 analysis

| | Detected "photon" | Undetected "photon" |
|---|---|---|
| Normalisation to $e^+e^-$ | KLOE10 ($1.0\,\text{GeV}$, $\pi^+\pi^-$) | KLOE08 ($1.020\,\text{GeV}$, $\pi^+\pi^-$) |
| | BES-III ($3.773\,\text{GeV}$, $\pi^+\pi^-$) | BaBar ($10.580\,\text{GeV}$, $p\overline{p}$) |
| | BaBar ($10.580\,\text{GeV}$, most channels) | |
| Normalisation to $\mu^+\mu^-\gamma$ | BaBar ($10.580\,\text{GeV}$, $\pi^+\pi^-$) | KLOE12 ($1.020\,\text{GeV}$, $\pi^+\pi^-$) |
| | CLEO-c ($3.671\,\text{GeV}$, $\pi^+\pi^-$) | |

Table 2: Summary of the choices made by radiative-return experiments, comparing the choice of normalisation process and whether a calorimeter-detected photon is required. All energies are given in the c.m. system.

was based on the largest ever dataset at the $\rho$-resonance region, with $34 \times 10^6$ selected $\pi^+\pi^-$ events at $\sqrt{s} < 1\,\mathrm{GeV}$. The large statistics were crucial to study various systematic effects in detail. The main features of the analysis include three independent procedures for measuring the number of detected $\pi^+\pi^-$ events: using momentum distributions of two particles measured in the tracking system, or using detected energy depositions in the LXe calorimeter, or using the polar angular distribution. Two of these methods, using momenta and angles of tracks, relied heavily on differential cross section predictions from Monte Carlo generators. The collected statistics allowed to perform comparisons of measured momentum distributions with the predicted spectra from the Monte Carlo, where a discrepancy in tails was observed. This was traced to the limitations of a collinear jet approximation used in the MCGPJ generator, and required upgrading the generator to take into account the angular distributions of photons in jets. Another feature of the $\pi^+\pi^-$ analysis by CMD-3 was a comprehensive study of the detector acceptance systematic uncertainty due to the determination of the polar angle. The forward–backward charge asymmetry of $\pi^+\pi^-$ was measured with an integrated statistical precision of about 0.025%. A 1% level deviation of the data from the theoretical predictions was observed. This highlighted a limitation of the commonly used sQED approach for the calculation of radiative corrections from the pion final state [57, 58].

The overall collected integrated luminosity per detector surpassed the projected value of $1\,\mathrm{fb}^{-1}$ in spring 2024. The experiments will continue to collect data in the current configurations for the next few years. After this, a moderate upgrade of the detector subsystems is expected. In the longer term, further collider upgrades are being discussed to achieve even higher luminosity in the threshold region, and new dedicated detectors are foreseen with the potential to improve the systematic accuracies of the cross-section measurement.

## 2.2 LNF-INFN, Frascati: DAΦNE and KLOE

DAΦNE at Frascati LNF-INFN is an electron–positron collider optimised to run at the c.m. energy corresponding to the $\phi(1020)$ meson mass. KLOE is a multipurpose detector for the DAΦNE collider. Its main component is a cylindrical drift chamber of $3.3\,\mathrm{m}$ length and $2\,\mathrm{m}$ diameter, with an internal radius of $25\,\mathrm{cm}$. Together with a lead-scintillating fibre electromagnetic calorimeter the chamber is embedded in the $0.52\,\mathrm{T}$ field of a superconducting solenoid. For a more detailed description of the detector we refer to [59–63]. KLOE took data from 2000 to 2006 and acquired $2.5\,\mathrm{fb}^{-1}$ of data at the $\phi$ peak, plus a further $250\,\mathrm{pb}^{-1}$ at other energies. Reviews of the KLOE experimental results using data collected until 2005, including analyses finished until 2008, are given in [64, 65], while the physics programme for KLOE-2 was laid out in [66].

KLOE made use of the radiative-return method for the measurement of the hadronic cross section, focusing in particular on the $\pi\pi$ hadronic channel. KLOE published four hadronic cross section measurements [16–19], which for convenience are called KLOE05[2], KLOE08, KLOE10, and KLOE12, respectively. The KLOE08 analysis measures the differential cross section for $e^+e^- \to \pi^+\pi^-\gamma$ as a function of the $\pi^+\pi^-$ invariant mass, $s_{\pi\pi}$, for radiative events. The di-pion cross section $\sigma(e^+e^- \to \pi^+\pi^-)$, which for shorthand we write as $\sigma_{\pi\pi}$, is derived using

$$s\frac{d\sigma(ee \to \pi\pi\gamma)}{ds_{\pi\pi}}\bigg|_{\mathrm{ISR}} = \sigma_{\pi\pi}(s_{\pi\pi})H(s_{\pi\pi}, s)\,, \tag{2.1}$$

where $H(s_{\pi\pi}, s)$ is the radiator function. In the KLOE08 analysis, $H(s_{\pi\pi}, s)$ was obtained with the PHOKHARA (Version 5) Monte Carlo generator.

KLOE08 was performed using $240.0\,\mathrm{pb}^{-1}$ of on-peak data ($\sqrt{s} = 1019.48\,\mathrm{MeV}$) corresponding to about 3 million events. Small-angle selection cuts were applied in the analysis: photons were

---

[2]This measurement was superseded by KLOE08.

restricted to a cone of $\theta_\gamma < 15°$ ($> 165°$) around the beam line. The two charged tracks, on the other hand, were detected in the range $50° < \theta^\pm < 130°$ and any photons in this region were not detected.

The measurement of the $\pi\pi\gamma$ cross section was normalised to the DA$\Phi$NE luminosity using large-angle Bhabha scattering (using the BABAYAGA@NLO Monte Carlo generator [67]), with 0.3% total systematic uncertainty. The pion vector form factor (VFF) $F_\pi(s)$ and $a_\mu^{\rm HVP}$ were derived using 60 points in the $s_{\pi\pi}$ region between 0.35 and 0.95 GeV$^2$.

KLOE10 used so-called large-angle cuts: both the photon and the charged tracks are detected at large angles $50° < \theta^\pm, \theta_\gamma < 130°$. With this selection it is possible to reach the dipion threshold, but at the price of reduced signal yield and enhancing FSR and $\phi \to \pi^+\pi^-\pi^0$ backgrounds. Therefore the KLOE10 analysis was performed using 232.6 pb$^{-1}$ of data taken at the c.m. energy $\sqrt{s} = 1$ GeV, corresponding to 0.6 million events. The analysis of $a_\mu^{\pi\pi}$ has 75 points in the region (0.10–0.85) GeV$^2$.

KLOE08 and KLOE10 were both normalised to the DA$\Phi$NE luminosity, and used the radiator function to obtain the pion VFF and $a_\mu^{\rm HVP}$. KLOE12, on the other hand, was normalised with respect to the muon radiative differential cross section. In this approach many systematic effects cancel out, including some related to theoretical uncertainties. KLOE12 was published using the same 240 pb$^{-1}$ data sample from 2002 used in KLOE08, and implemented identical small-angle cuts. Like KLOE08, the differential cross section was measured across 60 points in the energy region between 0.35 and 0.95 GeV$^2$.

While KLOE08 and KLOE12 used KLOE on-peak data from 2002, there exists around 1.7 fb$^{-1}$ of on-peak data from 2004 and 2005 which has never been analysed for the measurement of the $2\pi$ cross section. A new experimental effort has begun with the aim of measuring $a_\mu^{\pi\pi}$ with greater precision by using the full 2004–2005 dataset. In order to minimise biases from the published analysis, the new analysis will be conducted blindly. In addition, KLOE is analysing the three-pion cross section using the radiative return method with 1.7 fb$^{-1}$ of data collected at the $\phi$ meson mass [68].

## 2.3   IHEP, Beijing: BEPCII and BESIII

The BESIII experiment [69] is located at Institute for High Energy Physics (IHEP) in Beijing. It records symmetric $e^+e^-$ collisions provided by the BEPC-II collider with c.m. energies in the range between 1.8 and 5 GeV and luminosity exceeding $10^{33}$ cm$^{-2}$s$^{-1}$. The BESIII detector adopts an onion-shape structure, which allows to cover 93% of the full solid angle. It consists of a spectrometer, based on a multilayer drift chamber and (starting from fall 2024) three cylindrical gas electron multiplier layers [70] in the inner region; a time-of-flight system made of plastic scintillators; a CsI(Tl) electromagnetic calorimeter; and a muon identification system provided by resistive plate chambers, which instrument the return-flux joke of the superconducting magnet, enclosing the detector and providing a 1 T magnetic field. Starting from 2008, BESIII has collected the world's largest datasets of electron–positron collisions in the $\tau$-charm energy region, accounting for about 50 fb$^{-1}$. Highlights of the BESIII datasets are:

- $10^{10}$ $J/\psi$ decays,

- $2.7 \times 10^9$ $\psi(3686)$ decays,

- 20 fb$^{-1}$ at the $\psi(3770)$ resonance,

- 130 scan points between 2 and 4.6 GeV.

In summer 2024, the BEPCII accelerator underwent an upgrade, in order to extend the energy range over which peak luminosity can be reached. Profiting from the increased luminosity, the BESIII Collaboration plans to collect large data samples, mainly in the energy region above 4 GeV [71].

The BESIII Collaboration is very active in hadronic cross section measurements for the muon $(g-2)$ effort. In 2015, a first measurement of the $e^+e^- \to \pi^+\pi^-$ cross section with 0.9% systematic uncertainty has been achieved [10], based on a data sample of 2.9 fb$^{-1}$ collected at 3.773 GeV. The measurement used the radiative return technique to access the energy region 600–900 MeV, thus including the dominant contribution from the $\rho$ resonance. In this analysis, the selection cuts required both pion tracks to be reconstructed in the drift chamber, and for the hard photon to be detected at large angle in the electromagnetic calorimeter. A kinematic fit was then performed to constrain the four-momentum of the reconstructed $\pi^+\pi^-\gamma$ final state to the centre-of-mass energy. For this result, the event yield was normalised to the integrated luminosity, obtained by measuring Bhabha events, and the radiator function in (2.1) was obtained using the PHOKHARA event generator.

Preliminary radiative-return results have also been presented for the three-pion channel [72]. Furthermore, a series of cross section measurements of multi-hadronic states have been carried out by energy scan above 2 GeV. Most notably, the $e^+e^- \to \pi^+\pi^-\pi^0$ cross section has been measured in the energy range between 2.0 and 3.08 GeV [73]. In addition, a precision measurement of the total hadronic cross section has recently been measured with the world's highest accuracy via an energy scan between 2.23 and 3.08 GeV [74].

In the future, a new precision analysis of the channel $e^+e^- \to \pi^+\pi^-$ using the 20 fb$^{-1}$ dataset at the $\psi(3770)$ resonance is foreseen. The statistics of this data sample are sufficient to allow normalisation to the di-muon sample, which will improve the systematic uncertainty as the error on the radiator function and the luminosity decreases. Additional radiative-return analyses concern the processes $e^+e^- \to K^+K^-$ and $e^+e^- \to K^0\bar{K}^0$. Exploiting the high-statistics energy scan sample between 2.0 and 4.6 GeV, the uncertainty of the total hadronic cross section measurement will be further improved. A detailed description of the BESIII programme is given in [71].

## 2.4  SLAC, Stanford: PEP-II and BABAR

The BABAR experiment [75] operated a general-purpose quasi-full-solid-angle detector at the SLAC PEP-II asymmetric electron–positron collider from 1999 to 2008, collecting data for $e^+e^-$ collisions at and around the $\Upsilon(4S)$ resonance for a total integrated luminosity of 424 fb$^{-1}$.

BABAR measured the cross section of the process $e^+e^- \to \text{hadrons}(\gamma)$ for a large number of exclusive channels, using the radiative return method [76], which provides good experimental access to hadronic final states with invariant masses from the production threshold to typically $3-5$ GeV. The reported measurements entirely cover energies up to $1.8-2.0$ GeV, the energy beyond which the number and multiplicity of the exclusive hadronic modes and the small size of the total hadronic cross section make inclusive measurements and theory predictions more convenient for the calculation of the $a_\mu^{\text{HVP}}$ dispersive integral. BABAR measured all relevant hadronic channels except for the part of the $\pi^+\pi^-4\pi^0$ channel that does not proceed through $\eta 3\pi$. Using isospin symmetries, it has been estimated that this unmeasured contribution amounts to only $(0.016 \pm 0.016)\%$ [77].

Almost all BABAR measurements select events requiring a detected large-angle hard photon as ISR photon candidate (with a $\sim 10\%$ efficiency) [11, 78–91], relying on the relative large energy of the ISR photon over the whole range of the measured invariant mass of the hadronic system at the BABAR beam energies. By constraining the whole event reconstructed candidates to the

well known position, invariant mass and momentum of the colliding electron-positron system, this analysis strategy suppresses background contamination, and also permits the measurement of extra higher-order radiation consisting of additional detected photons and either one undetected large-angle photon or two opposite-angle beam-collinear photons [11, 78, 79]. Furthermore, requiring a hard photon that is well contained in the detector increases the probability that also the hadronic recoil system is fully detected on the other side, improving the quality of particle identification, and reducing systematic uncertainties related to the simulation of the selection and reconstruction efficiency, particularly for high multiplicity channels. Due to the relatively high involved hadronic invariant mass of the measured final state, no detected ISR photon is required for one $p\bar{p}$ channel measurement [92].

Most BABAR measurements [80–91] normalise the measured yields using a radiator function Monte Carlo simulation at NLO QED and the estimated integrated luminosity (typically order 1% precise) of the analysed sample. Signal and background ISR processes are simulated with an event generator based on EVA, additional ISR photons that are collinear to the beams are simulated with the structure function method [93], and additional FSR photons are simulated with PHOTOS. For checks and additional studies, BABAR uses the generators PHOKHARA and AFKQED.

The most precise BABAR measurements, on $\pi^+\pi^-(\gamma)$ [11, 78] and $K^+K^-(\gamma)$ [79], measure the ratio between the hadronic and the $\mu^+\mu^-(\gamma)$ cross sections, which is not affected by the uncertainties on the integrated luminosity. Furthermore, this approach permits using loose event selections that include higher-order radiative events in both the measurement and the normalisation channel, since these contributions mostly cancel in the ratio, greatly reducing systematic uncertainties related to the simulation of higher-order radiative events.

Using a data sample corresponding to an integrated luminosity of $232\,\mathrm{fb}^{-1}$, BABAR measured the $e^+e^- \to \pi^+\pi^-(\gamma)$ cross section from threshold to $3.0\,\mathrm{GeV}$, performing on the same sample a simultaneous measurement of the $e^+e^- \to \mu^+\mu^-(\gamma)$ cross section as normalisation [11, 78]. The corresponding $a_\mu^{\mathrm{HVP}}$ contribution has a precision of 0.74%, the best of all BABAR measurements. The $e^+e^- \to \mu^+\mu^-(\gamma)$ cross section agrees over the whole range with the NLO QED prediction normalised with the integrated luminosity measured by BABAR, within a 1.1% uncertainty, which is dominated by the luminosity uncertainty. In the $\rho$ peak region, the estimated systematic uncertainty is 0.5%. The reported measurement of the $K^+K^-(\gamma)$ channel, dominated by the $\phi$ resonance, has been performed with the same analysis strategy [79]. Regarding the $e^+e^- \to \pi^+\pi^-(\gamma)$ channel, there is an on-going effort to complete an improved measurement that will use the whole collected statistics and will statistically separate pions from muons using their kinematic distributions in order to reduce the systematic uncertainties related to particle identification.

BABAR has precisely measured the 3-pion and 4-pion final states, which correspond to the largest multi-hadronic cross sections below $2\,\mathrm{GeV}$: $\pi^+\pi^-\pi^0$ [80], $2\pi^+2\pi^-$ [81, 82] and $\pi^+\pi^-2\pi^0$ [85]. The measurements reported by BABAR also include the proton–antiproton final state [89, 90, 92], final states with kaons $K_S^0 K_L^0$, $K\bar{K} + n$ pions with $n = 1, 2$, $K\bar{K}K^+K^-$ [86–88, 94–96], final states including $\eta$ mesons, $\eta\pi^+\pi^-$ [83, 97], $\eta 2\pi^+2\pi^-$ [83], $\eta\pi^+\pi^-2\pi^0$ [98], and final states with up to 6 quasi-stable hadrons [84].

## 2.5 KEK, Tsukuba: SuperKEKB and Belle-II

The Belle-II experiment [99] is located at the SuperKEKB collider [100], an upgrade of the KEKB accelerator, running at a c.m. energy of $10.58\,\mathrm{GeV}$ (near the $\Upsilon(4S)$ resonance) with $7\,\mathrm{GeV}$ electron and $4\,\mathrm{GeV}$ positron beams. The Belle-II experimental programme [101] is mainly related to $B$-meson physics. The design peak luminosity is $6 \times 10^{35}\,\mathrm{cm}^{-2}\mathrm{s}^{-1}$. The experiment started in 2019 and plans to collect $50\,\mathrm{ab}^{-1}$ of data by 2032 [102]. Radiative-return measurements were not

possible at the previous Belle experiment [103] running between 1999 and 2010 due to trigger efficiency, where one of the reasons was that the Bhabha veto condition used only polar angle information [104]. Belle-II implements Bhabha veto based on both polar and azimuthal angles. Recent Belle-II results on the $e^+e^- \to \pi^+\pi^-\pi^0\gamma$ channel [105] use $191\,\text{fb}^{-1}$ of data. In the reconstruction the polar angle of the detected (ISR) photon in the c.m. system is in the range from $48°$ to $135°$. The invariant mass of the $\pi^+\pi^-\pi^0\gamma$ system is required to be greater than $8.0\,\text{GeV}$ to suppress the effect of extra emitted photons. PHOKHARA9.1 is used to simulate the signal [106], with a claimed systematic uncertainty for radiative corrections of 0.5% [107]. In the future, a measurement of the $e^+e^- \to \pi^+\pi^-\gamma$ cross section is planned, with a target precision of 0.5% for $a_\mu^{\pi\pi}$. The measurement will follow BaBar methods as a baseline, but compared to BaBar it will use a much larger integrated luminosity to control systematic effects, which dominate the uncertainty.

# 3 Computational setup

In this section we provide a conceptual overview of the various contributions that are potentially required to obtain a precise theoretical prediction, with a special focus on $e^+e^- \to X^+X^-$ and the radiative processes $e^+e^- \to X^+X^-\gamma$. If $X \in \{e, \mu\}$, the only effects that cannot be computed within standard QED perturbation theory are due to internal light quark loops leading to hadronic loops. If this hadronic loop is attached to two photons, we get HVP contributions. In the case of a loop attached to four photons, we have HLbL contributions. However, the latter only contribute beyond NNLO. For $X = \pi$ the situation is more delicate and requires adapted techniques for different parts of a cross section. As a result, complete higher-order corrections to processes (1.1) and (1.2), taking into account all terms of a particular order in $\alpha$, are the exception rather than the rule. Instead, partial corrections are assembled to obtain the best possible theoretical description.

The split of a cross section into different contributions can be dictated by the applied techniques or by physical considerations. We broadly identify three different techniques, illustrated in Figure 2. Fixed-order QED contributions are depicted as the red blob. Tremendous progress has been made in the past years in the ability to perform such computations. An overview of the required parts and the current status will be given in Section 3.1. The green blob represents approximate higher-order radiative corrections and includes potential resummation of multiple collinear and soft photon emission. This is the topic of Section 3.2. Finally, the blue blob involves techniques beyond pure QED, as it has to take into account the nonpointlike structure of pions. They will be described Section 3.4. The HVP and HLbL effects will be discussed in Section 3.3.

From a physical point of view, it is also possible to split contributions of a cross section into

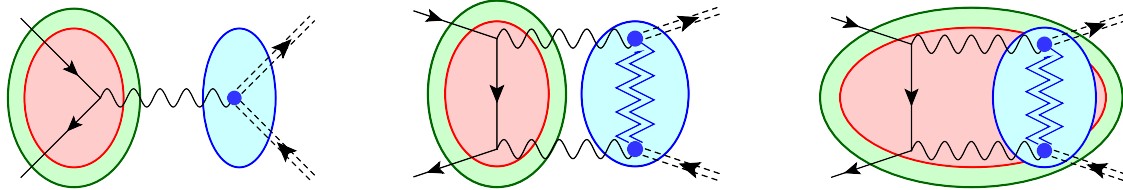

Figure 2: Illustration of various categories of corrections to $e^+e^- \to \pi^+\pi^- (+\gamma)$. The red blob stands for fixed-order QED corrections, while the green blob indicates potentially resummed approximate QED radiation effects. Since the pion is not pointlike, effects beyond standard QED perturbation theory are present, as indicated by the blue blob. The emission of additional real photons is not depicted, but understood.

initial-state corrections (ISC), final-state corrections (FSC), and mixed corrections (interference initial-final) parts. Such a split is partly motivated by technical aspects (different techniques used, different computational complexity) but also by different numerical impact. Looking at the left panel of Figure 2, where there is a clear separation into ISC (red and green) and FSC (blue), the ISC are often dominant. They include higher-order corrections enhanced by large collinear logarithms $\alpha^n L_c^n \equiv \alpha^n \log^n(Q^2/m_e^2)$ with the typical scale of the process $Q \sim \sqrt{s}$ much larger than the electron mass $m_e$. Since in our case the initial state always consists of an electron–positron pair, ISC can be systematically improved through QED perturbation theory. For ISC, the initial state is linked to the final state through the exchange of a single photon and we will adopt the term one-photon exchange (1PE) contribution familiar from lepton–proton scattering for this topology.

In a pure QED process such as (1.1b), FSC can be obtained by recycling the ISC computation. If the final state contains hadrons, different techniques are required, as mentioned above. The mixed corrections are technically much more demanding even for purely leptonic processes. They involve two-photon exchange (2PE) and at NNLO even three-photon exchange (3PE) contributions, i.e., contributions where the inital and final states are connected by two or at NNLO even three photons. Only recently the computation of such corrections at NNLO for $2 \to 2$ processes including mass effects became feasible.

In the following subsections we will go through the various techniques mentioned above. We will not give a detailed description of how the computations are done, but refer to the literature for such aspects. However, we will describe in more detail what contributions exist and set up a more precise notation for them. This will facilitate the description of what precisely is included in the various codes and the relative numerical importance of these partial corrections.

## 3.1 Fixed-order QED

In this section we focus on fixed-order QED contributions. The impact of heavy electroweak gauge bosons is typically negligible for the low-energy processes we are considering here. In order to establish our notation and present the basics of the computational framework, we focus on the process $e^+ e^- \to \mu^+ \mu^-$. At the end of the subsection we briefly comment on the complications with pions in the final state, as a preparation for Section 3.4.

We start with the tree-level amplitude, denoted by $\mathcal{A}_{mm}^{(0)}(q_e\, q_m) \sim q_e\, q_m$. The subscript indicates the final state ($m$ for muon), suppressing the initial state that is always $e^+ e^-$ in this paper. As the argument we have given the power of the electron charge $q_e$ and the muon charge $q_m$, respectively. We formally distinguish between these charges to identify various gauge-invariant contributions in more complicated situations. As generic symbol we use $q_\ell$. The (differential) LO cross section $\mathrm{d}\sigma_{mm}^{(0)}(q_e^2\, q_m^2)$ is then obtained by integrating the tree-level squared matrix element

$$\mathcal{M}_{mm}^{(0)}(q_e^2\, q_m^2) = \left| \mathcal{A}_{mm}^{(0)}(q_e\, q_m) \right|^2 \tag{3.1}$$

over the two-body phase space.

For cross sections that correspond to realistic experimental situations the phase-space integration needs to be done through numerical Monte Carlo methods. Some Monte Carlo codes are so-called integrators, where directly histograms for arbitrary IR safe observables are produced. Experimental cuts are taken into account during the numerical integration. More common are so-called generators, where in a first step generic events are generated. These events are then used in a second analysis step to obtain differential distributions. Generators offer a greater flexibility to include detailed detector simulations. For this to be efficient, however, it is important to keep the number of events with negative weight under control.

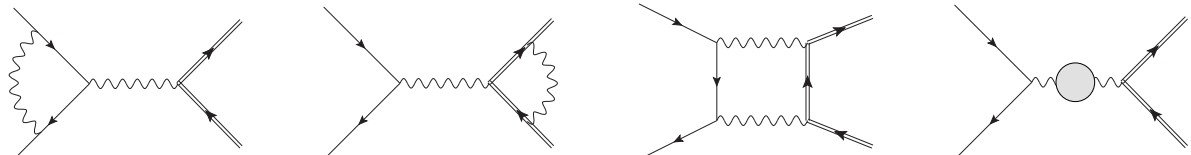

Figure 3: Some diagrams for the NLO amplitude of $e^+ e^- \to \mu^+ \mu^-$, corresponding to the four gauge-invariant parts $\mathcal{A}_{mm}^{(1)}(q_e^3\, q_m)$, $\mathcal{A}_{mm}^{(1)}(q_e\, q_m^3)$, $\mathcal{A}_{mm}^{(1)}(q_e^2\, q_m^2)$, and $\mathcal{A}_{mm}^{(1)}(q_e\, q_m\, \Pi^{(1)})$ of (3.2).

Going beyond LO, we need the one-loop amplitude that we split into four parts

$$\mathcal{A}_{mm}^{(1)}(q_e\, q_m\, q_\ell^2) = \mathcal{A}_{mm}^{(1)}(q_e^3\, q_m) + \mathcal{A}_{mm}^{(1)}(q_e\, q_m^3) + \mathcal{A}_{mm}^{(1)}(q_e^2\, q_m^2) + \mathcal{A}_{mm}^{(1)}(q_e\, q_m\, \Pi^{(1)}) \qquad (3.2)$$

as illustrated in Figure 3. The first (second) term on the right corresponds to corrections to the electron (muon) line with three couplings of the photon to the electron (muon) and only one coupling to the muon (electron). The third term is due to box diagrams with two couplings each to the electron and muon. Together they make up the photonic corrections. Finally, the last term is due to an insertion of the (one-loop leptonic and hadronic) VP, denoted by $\Pi^{(1)} = \Pi_\ell^{(1)} + \Pi_h$ with $\Pi_\ell^{(1)} \sim q_\ell^2$. We will call fermionic corrections all corrections that contain a closed lepton or hadron loop.

As alluded to above, the insertion of $\Pi_h$ is a contribution beyond strict QED and the counting in terms of LO, NLO, etc, is not fully applicable any longer. Related to this, the VPC are often treated by resumming VP insertions. In this approach, the (complex) VP can be evaluated to the highest possible precision. Up to $n = 2$ the leptonic VP can be split into $\Pi_\ell^{(n)}(q_\ell^{2n}) = \Pi_e^{(n)} + \Pi_m^{(n)} + \Pi_\tau^{(n)}$ and it can be obtained with full mass dependence from [108,109]. Three-loop results also exist in different forms of mass-term expansions [110,111]. However, beyond $n = 2$, hadronic contributions mix into the leptonic part and a strict separation is not possible any longer. The determination of $\Pi_h$ from experimental data will be addressed in Section 3.3. In Section 5.1.1 we will discuss the determination of the full VP from experimental data and compare various packages available in the literature.

The complete virtual NLO corrections to $e^+ e^- \to \mu^+ \mu^-$ are obtained by integrating

$$\mathcal{M}_{mm}^{(1)}(q_e^2\, q_m^2\, q_\ell^2) = 2\, \text{Re}\big(\mathcal{A}_{mm}^{(1)}(q_e\, q_m\, q_\ell^2)\, \mathcal{A}_{mm}^{(0)*}(q_e\, q_m)\big) \qquad (3.3)$$

over the two-body phase space. Here we introduced the notation $\mathcal{M}_{xx}^{(n)}$ to include all terms of the squared matrix elements with a power $\mathcal{O}(\alpha^n)$ relative to the LO term. After removing the UV singularities in (3.3) through renormalisation (usually in the on-shell scheme) there are still IR singularities. They cancel for physical (IR-safe) observables if the virtual corrections are combined with real corrections. The latter are obtained by integrating

$$\mathcal{M}_{mm\gamma}^{(0)}(q_e^2\, q_m^2\, q_\ell^2) = \big|\mathcal{A}_{mm\gamma}^{(0)}(q_e\, q_m\, q_\ell)\big|^2 = \big|\mathcal{A}_{mm\gamma}^{(0)}(q_e^2\, q_m) + \mathcal{A}_{mm\gamma}^{(0)}(q_e\, q_m^2)\big|^2 \qquad (3.4)$$

over the three-body phase space. Here, $\mathcal{A}_{mm\gamma}^{(0)}(q_e^2\, q_m)$ and $\mathcal{A}_{mm\gamma}^{(0)}(q_e\, q_m^2)$ are the gauge-invariant part of the tree-level amplitude for $e^+ e^- \to \mu^+ \mu^- \gamma$ where the photon is radiated off the initial and final state, respectively.

Keeping finite fermion masses, there are no collinear singularities in QED. Instead, collinear logarithms $L_c$ appear. However, there are still soft singularities. They can be regularised either by introducing a fictitious photon mass $m_\gamma$ or by using dimensional regularisation also for the IR case. The regularised singularities then appear as $\log(Q^2/m_\gamma^2)$ or $1/\epsilon$ poles in intermediate

expressions. The extraction of the IR singularities of the real corrections can be done in a process- and observable-independent way, using either the slicing method or a subtraction method. A short description of the two can be found, e.g., in [112]. Once real and virtual corrections are combined, it is safe to remove the regulator $m_\gamma \to 0$ or $\epsilon \to 0$. While not strictly necessary, typically the slicing method is combined with photon-mass regularisation, whereas the subtraction method is used with dimensional regularisation.

As for (3.2), the NLO corrections to the cross section, $\mathrm{d}\sigma_{mm}^{(1)}(q_e^2 \, q_m^2 \, q_\ell^2)$, can also be split into four parts. If we were to restrict ourselves to initial-state photonic corrections only, we would use

$$\mathcal{M}_{mm}^{(1)}(q_e^4 \, q_m^2) = 2 \operatorname{Re}\big(\mathcal{A}_{mm}^{(1)}(q_e^3 \, q_m) \, \mathcal{A}_{mm}^{(0)*}(q_e \, q_m)\big)\,, \tag{3.5}$$

$$\mathcal{M}_{mm\gamma}^{(0)}(q_e^4 \, q_m^2) = \big|\mathcal{A}_{mm\gamma}^{(0)}(q_e^2 \, q_m)\big|^2\,, \tag{3.6}$$

and integrate (3.5) and (3.6) over the two-body and three-body phase space, respectively, to obtain $\mathrm{d}\sigma_{mm}^{(1)}(q_e^4 \, q_m^2)$. Analogously, we can obtain $\mathrm{d}\sigma_{mm}^{(1)}(q_e^2 \, q_m^4)$, the final-state NLO corrections to $e^+ \, e^- \to \mu^+ \, \mu^-$. Finally, there are the mixed corrections consisting of $\mathrm{d}\sigma_{mm}^{(1)}(q_e^3 \, q_m^3)$ (interference between emission of initial state and final state) and $\mathrm{d}\sigma_{mm}^{(1)}(q_e^2 \, q_m^2 \, \Pi^{(1)})$ (VP corretions). The sum of these four parts is denoted by

$$\mathrm{d}\sigma_{mm}^{(1)}(q_e^2 \, q_m^2 \, q_\ell^2) = \underbrace{\mathrm{d}\sigma_{mm}^{(1)}(q_e^4 \, q_m^2)}_{\text{ISC}} + \underbrace{\mathrm{d}\sigma_{mm}^{(1)}(q_e^2 \, q_m^4)}_{\text{FSC}} + \underbrace{\mathrm{d}\sigma_{mm}^{(1)}(q_e^3 \, q_m^3)}_{\text{mixed}} + \underbrace{\mathrm{d}\sigma_{mm}^{(1)}(q_e^2 \, q_m^2 \, \Pi^{(1)})}_{\text{VPC}} \tag{3.7}$$

and corresponds to the integration of the full matrix elements (3.3) and (3.4) over the two- and three-body phase space. In (3.7), the ISC and VPC make up the red blob on the left panel of Figure 2 for the $\mu^+ \, \mu^-$ final state.

Of course, we can also integrate (3.4) and explicitly require a resolved photon, i.e., a photon with energy above a certain threshold such that in principle it could be detected. In this case we obtain

$$\mathrm{d}\sigma_{mm\gamma}^{(0)}(q_e^2 \, q_m^2 \, q_\ell^2) = \underbrace{\mathrm{d}\sigma_{mm\gamma}^{(0)}(q_e^4 \, q_m^2)}_{\text{ISC}} + \underbrace{\mathrm{d}\sigma_{mm\gamma}^{(0)}(q_e^2 \, q_m^4)}_{\text{FSC}} + \underbrace{\mathrm{d}\sigma_{mm\gamma}^{(0)}(q_e^3 \, q_m^3)}_{\text{mixed}}\,, \tag{3.8}$$

the LO cross section to the process $e^+ \, e^- \to \mu^+ \, \mu^- \, \gamma$. This process is often called an ISR process, a terminology that is somewhat misleading. The photon can be from ISC, but also from FSC or interference (mixed contribution). Hence, we will refer to such a process simply as a radiative process. How important the non-ISC terms are compared to the ISC part for a cross section like (3.8) depends crucially on the precise definition of the observable and is a question to be addressed in Section 5.

The split of fixed-order corrections to $e^+ \, e^- \to X^+ \, X^-$ into ISC, FSC, mixed, and VPC can be extended to any order. At NNLO, double virtual corrections have to be combined with real-virtual and double real in order to obtain a meaningful IR finite result. The corrections $\mathrm{d}\sigma_{mm}^{(2)}(q_e^2 \, q_m^2 \, q_\ell^4)$ contain at least two powers of $q_e$ and $q_m$. We call VPC those corrections that contain precisely a factor $q_e^2$ and a factor $q_m^2$. They are obtained by VP insertions to the photon propagator present in the Born diagram, namely one insertion of the two-loop VP, $\mathrm{d}\sigma_{mm}^{(2)}(q_e^2 \, q_m^2 \, \Pi^{(2)})$, and two insertions of the one-loop VP, $\mathrm{d}\sigma_{mm}^{(2)}(q_e^2 \, q_m^2 \, (\Pi^{(1)})^2)$, including those from the one-loop amplitude squared, $|\mathcal{A}_{mm}^{(1)}(q_e \, q_m \, \Pi^{(1)})|^2$. The VPC have only double virtual corrections. These corrections are a subset of the resummed VPC discussed above.

ISC also contain precisely a power $q_m^2$ at NNLO, i.e., the minimal possible coupling to the final state (the muon in this case), but they contain $q_e^n$ with $n > 2$. Some representative examples for the squared matrix elements are shown in Figure 4. In the top row, there are double virtual

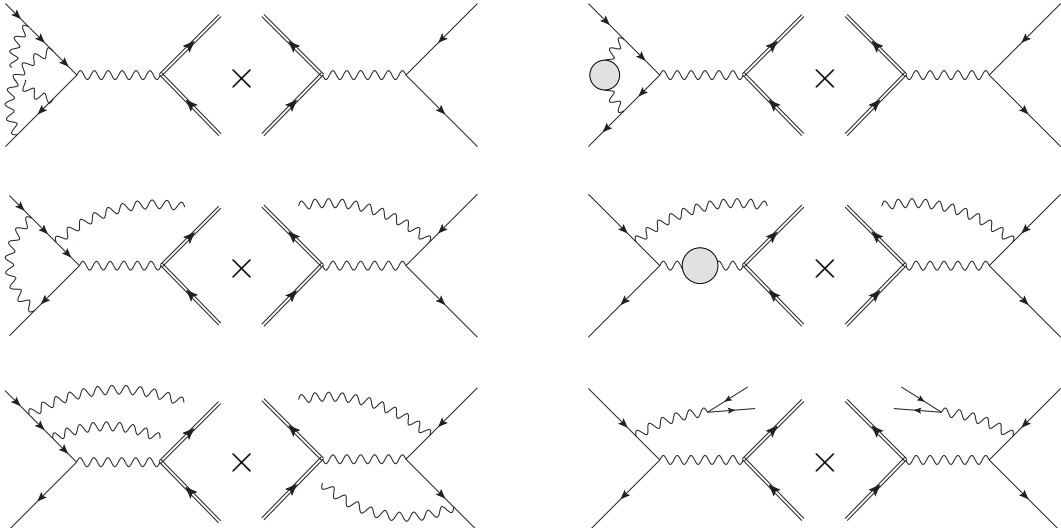

Figure 4: Representative diagrams contributing to $\mathrm{d}\sigma^{(2)}_{mm}(q_e^6\,q_m^2)$ (left panels) and $\mathrm{d}\sigma^{(2)}_{mm}(q_e^4\,q_m^2\,\Pi^{(1)})$ (right panels), the pure ISC of the NNLO corrections for $e^+e^- \to \mu^+\mu^-$. The double real VP contribution at the bottom of the right panel corresponds to a measurably different process $e^+e^- \to \mu^+\mu^-\,e^+e^-$.

corrections. Apart from purely photonic corrections $\mathrm{d}\sigma^{(2)}_{mm}(q_e^6\,q_m^2)$ (example in left panel) there are also VP contributions to ISC, $\mathrm{d}\sigma^{(2)}_{mm}(q_e^4\,q_m^2\,\Pi^{(1)})$ (right panel). Not shown are terms due to the one-loop amplitude squared, $|\mathcal{A}^{(1)}_{mm}(q_e^3\,q_m)|^2$. Also for the real-virtual corrections illustrated in the middle row there are photonic contributions (left panel) and VP contributions (right panel). Finally, there are double real contributions with examples in the bottom row. The associated VP contributions are related to the process with an additional lepton pair in the final state. Keeping the mass of this lepton different from zero, this results in a separately finite contribution and a measurably different process. Still, the cross section for such a process is logarithmically enhanced (as a remnant of the would-be IR singularities in the limit of massless leptons) and, hence, its impact has to be carefully considered.

All ISC at NNLO are by construction 1PE contributions between the incoming leptonic state and the final state. This makes them simpler from a technical point of view. The required two-loop amplitudes are covered by the two-loop form factor for massive particles [113] requiring two-loop integrals with at most three external legs. For the real-virtual part one-loop box diagrams are sufficient. The same holds for the FSC. They can be obtained by recycling the ISC computation, exchanging the roles of $q_e$ and $q_m$.

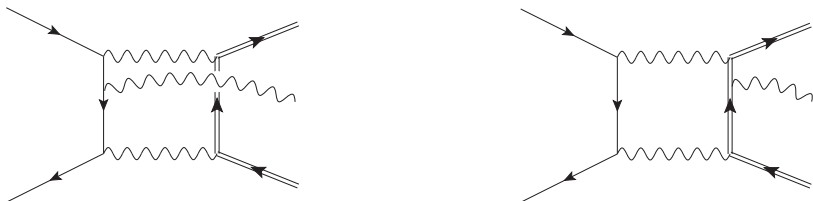

Figure 5: Representative 2PE diagrams contributing to 2PE-ISR $\mathcal{A}^{(1)}_{mm\gamma}(q_e^3\,q_m^2)$ (left) and 2PE-FSR $\mathcal{A}^{(1)}_{mm\gamma}(q_e^2\,q_m^3)$ (right) as part of the NLO amplitude for the process $e^+e^- \to \mu^+\mu^-\,\gamma$.

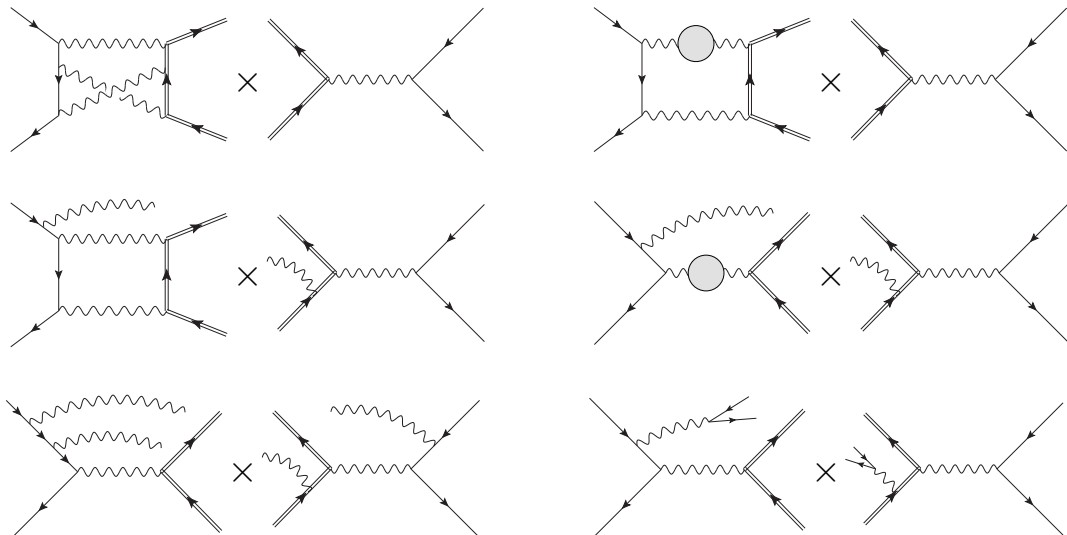

Figure 6: Representative diagrams contributing to mixed NNLO corrections $d\sigma_{mm}^{(2)}(q_e^i \, q_m^f \, q_\ell^x)$ (left panels) and $d\sigma_{mm}^{(2)}(q_e^i \, q_m^f \, \Pi^{(1)})$ (right panels), with $i > 2$ and $j > 2$ for $e^+ e^- \to \mu^+ \mu^-$. The double real VP contribution at the bottom of the right panel corresponds to a measurably different process $e^+ e^- \to \mu^+ \mu^- e^+ e^-$.

Finally, the mixed corrections are all those that are neither VPC, ISC, or FSC. In principle they can be disentangled further according to powers of $q_e$ and $q_m$ [114]. For example, the 2PE part of the one-loop amplitudes for the radiative process $e^+ e^- \to \mu^+ \mu^- \gamma$ can be split into additional emission from the electron line, $\mathcal{A}_{mm\gamma}^{(1)}(q_e^3 \, q_m^2)$, and muon line $\mathcal{A}_{mm\gamma}^{(1)}(q_e^2 \, q_m^3)$, as illustrated in Figure 5. The former (latter) we call 2PE-ISR (2PE-FSR) contributions. For processes with pointlike particles this split is mainly done for computational reasons. Indeed, for the mixed corrections, the computations are much more involved than for ISC, even for pointlike final states. As depicted in the first row of Figure 6 two-loop integrals with four external legs are required. This is a serious complication, in particular for massive fermions. Only recently all two-loop four-point integrals with a massive muon and massless electron have been computed [115,116]. The integrals with two equal nonvanishing masses involve elliptic functions [117]. The real-virtual corrections, illustrated in the second row of Figure 6, now involve pentagon one-loop diagrams that often lead to numerical complications. Fortunately, there is by now a lot of experience how to deal with them [118–120].

With the NNLO contributions discussed above, we also have all ingredients for $d\sigma_{mm\gamma}^{(1)}$, the NLO cross section of the radiative process $e^+ e^- \to \mu^+ \mu^- \gamma$. Of course, the double virtual part, i.e., the two-loop and one-loop squared diagrams, are not required in this case. However, the split into ISC, FSC, and mixed contributions carries through. As illustrated in Figure 5, a further separation of mixed contributions into 2PE-ISR and 2PE-FSR is sometimes useful.

For pointlike final states, the situation described so far is roughly speaking the current state of the art. Next steps that are expected to be completed in the foreseeable future are N$^3$LO ISC for $e^+ e^- \to X^+ X^-$ and NNLO corrections to the radiative process $e^+ e^- \to \mu^+ \mu^- \gamma$. For the former, the three-loop triple virtual corrections are available [121–123]. However, a consistent combination with the various real corrections is still a formidable task. In particular the two-loop amplitudes with an additional photon are currently only known for massless electrons [124,125]. For the latter, the two-loop amplitude is the key ingredient that requires five-point two-loop integrals with an internal mass [126–128].

We end this subsection with a few comments regarding processes with a $\pi^+\pi^-$ pair in the final state. For the ISC there are no conceptual difficulties in their evaluation, since in this case the pions couple through the VFF with a single photon. Care has to be taken to avoid double counting of the VPC, as they are included in the pion VFF. However, FSC and mixed corrections are much more delicate. From a technical point of view it is possible to compute all contributions in scalar QED, order by order in perturbation theory, but the phenomenological reliability of this procedure is questionable. Including form factors at pion–photon vertices, i.e., using FsQED, can improve the situation is specific cases. For example, the 2PE-ISR contribution depicted in the left panel of Figure 5 can be reasonably approximated through FsQED. Additional photon emission from the pions is, however, more delicate. The 2PE-FSR part illustrated in the right panel of Figure 5 for instance cannot be described well through FsQED. This leads to serious limitations in our ability to describe processes with pions in the final state. In particular, the effect of higher-order perturbative corrections on the leptonic side might be drowned in nonperturbative hadronic uncertainties appearing already at lower orders in $\alpha$. A more detailed discussion of processes with pions in the final state is given in Section 3.4.

## 3.2 QED beyond fixed order

As detailed in the previous section, to a first approximation, the most logical way to perform calculations of observables is *order-by-order*, that is all the contributions beyond a fixed order are exactly equal to zero. In this section, a description of the methods to provide theoretical predictions beyond fixed order in QED will be given.

It is possible to write the complete set of QED perturbative leading and subleading corrections as in Table 3. The *order-by-order* criterion corresponds to the sequential calculation of rows in Table 3 and accounts for increasing powers of the perturbative expansion constant $\alpha$. For example, the LO contribution $\mathrm{d}\sigma_{xx}^{(0)}$ includes the only term that is present in the first row, whereas the NLO contribution $\mathrm{d}\sigma_{xx}^{(1)}$ accounts for the terms proportional to $\alpha L_c$ and to $\alpha$. However, since for $s \gg m_e^2$ the collinear logarithm can be large, $L_c = \log\left(Q^2/m_e^2\right) \sim \mathcal{O}(10)$, the terms proportional to powers of $L_c$ break the perturbative expansion in $\alpha$. Moreover, corrections in the LL column give relevant results, phenomenology-wise. Thus, a resummation procedure of the leading logarithms on top of a fixed-order calculation is crucial to reach high-precision Monte Carlo predictions. In formulae, the merging of the LL approximation on top of the LO calculation – corresponding to the first column of Table 3 – is

$$\mathrm{d}\sigma_{xx}^{(0)} + \mathrm{d}\sigma_{xx}^{(LL\geq 1)}. \tag{3.9}$$

|      | LL | NLL | NNLL | N$^3$LL | $\cdots$ |
|------|-----|-----|-----|-----|-----|
| LO | 1 | | | | |
| NLO | $\alpha L_c$ | $\alpha$ | | | |
| NNLO | $\alpha^2 L_c^2$ | $\alpha^2 L_c$ | $\alpha^2$ | | |
| N$^3$LO | $\alpha^3 L_c^3$ | $\alpha^3 L_c^2$ | $\alpha^3 L_c$ | $\alpha^3$ | |
| $\vdots$ | $\vdots$ | $\vdots$ | $\vdots$ | $\vdots$ | $\ddots$ |

Table 3: Contributions of the QED perturbative series, where $\alpha$ is the expansion constant and $L_c = \log\left(Q^2/m_e^2\right)$ is the collinear logarithm at the typical scale of the process $Q \sim \sqrt{s}$, which is much larger than the electron mass. The power counting of $\alpha$ is normalised to the LO. Rows represent corrections in increasing perturbative orders. Columns represent corrections in increasing logarithmic approximations.

Presently, Monte Carlo event generators which are used at flavour factories rely on exact fixed-order QED corrections and the LL approximation of higher-order effects, together with their consistent matching.

The most used techniques to account for multiple photon radiation are the QED structure function approach and the YFS exponentiation. As more thoroughly explored in Section 4, the first approach is used in the BABAYAGA@NLO (Monte Carlo parton shower algorithm), MCGPJ (analytical QED structure functions in the collinear approximation) and AFKQED [93,129] (similarly to MCGPJ) generators, whereas the second approach is used in KKMC and SHERPA.

### 3.2.1 QED parton distribution functions

In the *collinear factorisation* approach, in order to resum the large $L_c$ terms, one relies on a factorisation formula [130–132], similar to the standard QCD factorisation formula adopted at hadron colliders,

$$
\mathrm{d}\sigma_{e^+e^-\to X} = \tag{3.10}
$$
$$
\sum_{ij} \int \mathrm{d}x_+\mathrm{d}x_- \, D_{i/e^+}(x_+,\mu^2,m_e^2) \, D_{j/e^-}(x_-,\mu^2,m_e^2) \, \mathrm{d}\hat{\sigma}_{ij\to X}(x_+,x_-,Q^2,\mu^2) + \mathcal{O}\left(\frac{m_e^2}{Q^2}\right),
$$

valid up to power corrections in the mass of the electron $m_e^2$ over some hard scale $Q^2$. The term on the l.h.s. is the *particle-level* cross section for the process $e^+\,e^-\to X$, computed with massive electrons, whereas $\mathrm{d}\hat{\sigma}_{ij\to X}$ appearing under integration on the r.h.s. is the *parton-level* or *short-distance* cross section for the process $i\,j\to X$, which is free of electron mass singularities. The collinear logarithms are resummed by means of the Parton Distribution Functions (PDF) $D_{i/e^\pm}(x^\pm)$. The partons entering the short-distance cross section are rescaled by a longitudinal momentum fraction $x_\pm$, and their nature can coincide with that of the incoming particles i.e., $(i,j)=(e^+,e^-)$, or it can be different, e.g., $(i,j)=(\gamma,e^-),(e^-,e^-),\ldots$. Finally, a factorisation scale $\mu^2$ appears both in the $D_{i/e^\pm}$ and in $\mathrm{d}\hat{\sigma}_{ij\to X}$. A suitable factorisation scheme must be introduced in order to remove the collinear logarithms (or collinear divergences if massless electrons are used) present in the parton-level cross section.

At variance with hadronic PDFs, QED PDFs are entirely calculable with perturbative techniques. One option is to resort to an iterative Monte Carlo procedure as detailed in Section 3.2.2. Alternatively, one can solve numerically or analytically the Dokshitzer–Gribov–Lipatov–Altarelli–Parisi (DGLAP) evolution equation [133–136], given the initial condition for the evolution at the initial scale $\mu_0^2\simeq m_e^2$. Analytical solutions with LL accuracy have been extensively used for numerical simulations at LEP and flavour factories. Neglecting the running of the QED coupling constant, the nonsinglet QED PDF $D(x,\mu^2)\equiv D_{e^-/e^-}^{\mathrm{LL}}(x,\mu^2)$ (also called QED structure function in the literature) is solution of the DGLAP evolution equation

$$
\mu^2\frac{\partial}{\partial\mu^2}D\left(x,\mu^2\right) = \frac{\alpha}{2\pi}\int_x^1\frac{\mathrm{d}z}{z}P_+(z)D\left(\frac{x}{z},\mu^2\right), \tag{3.11}
$$

with initial condition $D(x,m_e^2)=\delta(1-x)$ and $P_+(z)$ the regularised $\ell\to\ell\gamma$ splitting function. The structure function takes into account how radiation is emitted at all orders in the collinear limit and can be interpreted as the probability density of having a lepton with momentum $p'=xp$ and virtuality $\mu^2$ in a parent lepton with momentum $p$. In the literature, different analytical solutions have been obtained with different logarithmic accuracies: purely-soft Gribov–Lipatov [135], hybrid additive [130,132,137,138], and hybrid factorised [139–141] solutions. The typical expression of

the hybrid additive solutions is of the form

$$D^{\mathrm{LL}}_{e^-/e^-}(x,Q^2) = \frac{\exp\left[(3/4-\gamma_E)\frac{\eta}{2}\right]}{\Gamma(1+\frac{\eta}{2})}\frac{\eta}{2}(1-x)^{-1+\frac{\eta}{2}} - \frac{1}{4}\eta(1+x) + \mathcal{O}(\alpha^2)\,, \quad \eta = \frac{2\alpha}{\pi}L_c\,, \quad (3.12)$$

while in the hybrid factorised solutions the Gribov–Lipatov exponentiated term is multiplied by finite-order terms. Such LL results are built out of an additive matching between a recursive solution up to some order in $\alpha$, and an all-order $\alpha$ solution valid in the region $x \to 1$. With $Q$ in the region of GeV we have $\eta \sim \mathcal{O}(10^{-2})$. Therefore, the $(1-x)^{-1+\frac{\eta}{2}}$ factor results in a PDF that is very peaked towards $x = 1$, where it diverges with an integrable singularity. A last comment is in order: for an $e^+e^-$ annihilation, the choice $\eta \to \beta = \frac{2\alpha}{\pi}\left[\log(s/m_e^2) - 1\right]$ in (3.12) allows to correctly resum all the IR structure of the cross-section calculated with (3.10) [142].

The next-to-leading logarithmic approximation for QED structure functions was first developed in [143]. Recently, there has been some additional progress to extend the accuracy to NLL. In [144], the electron-in-electron, positron-in-electron, and photon-in-electron PDFs have been calculated at NLL accuracy in the $\overline{\mathrm{MS}}$ factorisation and renormalisation scheme. The PDFs have been derived by solving the DGLAP equations both numerically and analytically, by using as initial conditions for the evolution the ones derived in [145]. In [146], these results have been improved in several directions: first, with a DGLAP evolution featuring multiple fermion families (leptons and quarks) in a variable flavour number scheme, i.e., by properly including the respective mass thresholds; second, by taking into account an alternative factorisation scheme, the $\Delta$ scheme [147], where the NLO initial condition are maximally simplified; third, by considering two alternative renormalisation schemes, $\alpha(m_Z)$ and $G_\mu$ schemes (where $\alpha$ is fixed). Finally, in [148], the framework has been extended to include QCD and mixed QED-QCD effects in the DGLAP evolution.

NLL PDFs ready for phenomenology can be obtained with the public code EMELA [146]. EMELA is a stand-alone code, and can be linked to any external programme. EMELA can also provide PDFs with beamstrahlung effects according to the procedure presented in [149].

### 3.2.2 Parton shower

The parton shower procedure considered in this section is a Monte Carlo algorithm that provides an exact numerical solution to the DGLAP evolution equation for the nonsinglet LL QED structure function (3.11). A regularised $\ell \to \ell\gamma$ splitting function, suitable for a Monte Carlo implementation, can be defined as

$$P_+(z) = \frac{1+z^2}{1-z}\Theta(1-\varepsilon-z) - \delta(1-z)\int_0^{1-\varepsilon}\mathrm{d}y\frac{1+y^2}{1-y}\,, \quad (3.13)$$

where $\varepsilon \ll 1$ is an IR cutoff. In this way, (3.11) can be rewritten as

$$\mu^2\frac{\partial}{\partial\mu^2}D\left(x,\mu^2\right) = \frac{\alpha}{2\pi}\int_x^{1-\varepsilon}\frac{\mathrm{d}z}{z}\frac{1+z^2}{1-z}D\left(\frac{x}{z},\mu^2\right) - \frac{\alpha}{2\pi}D\left(x,\mu^2\right)\int_x^{1-\varepsilon}\mathrm{d}z\frac{1+z^2}{1-z}\,. \quad (3.14)$$

One can introduce the Sudakov form factor [150]

$$\Delta(s_1,s_2) = \exp\left[-\frac{\alpha}{2\pi}\int_{s_2}^{s_1}\frac{\mathrm{d}s'}{s'}\int_0^{1-\varepsilon}\mathrm{d}z\frac{1+z^2}{1-z}\right]\,. \quad (3.15)$$

It represents the probability that the lepton evolves from virtuality $s_2$ to virtuality $s_1$ without emitting photons whose energy fraction is larger than $\varepsilon$. Then, (3.14) can be reworked in the integral form as

$$D(x,\mu^2) = \Delta\left(\mu^2,m_\ell^2\right)D\left(x,m_\ell^2\right) + \frac{\alpha}{2\pi}\int_{m_\ell^2}^{\mu^2}\frac{\mathrm{d}s'}{s'}\Delta\left(\mu^2,s'\right)\int_x^{1-\varepsilon}\frac{\mathrm{d}z}{z}\frac{1+z^2}{1-z}D\left(\frac{x}{z},s'\right)\,. \quad (3.16)$$

Iteratively, it becomes

$$D(x, \mu^2) = \sum_{i=0}^{\infty} \prod_{j=1}^{i} \left[ \frac{\alpha}{2\pi} \int_{m_\ell^2}^{s_{j-1}} \frac{\mathrm{d}s_j}{s_j} \Delta(s_{j-1}, s_j) \int_{x/(z_1 \cdot \ldots \cdot z_{j-1})}^{1-\varepsilon} \frac{\mathrm{d}z_j}{z_j} \frac{1 + z_j^2}{1 - z_j} \right]$$
$$\times \Delta(s_i, m_\ell^2) D\left( \frac{x}{z_1 \cdot \ldots \cdot z_i}, m_\ell^2 \right). \quad (3.17)$$

The previous equation allows calculating the structure function $D(x, \mu^2)$ via a Monte Carlo method, according to an iterative procedure [151]. This way, a shower of photons is generated and the simulated $x$ distribution exactly follows the structure function. For leptonic processes, if the scale is set to $\mu^2 = st/u$, this prescription also exponentiates the dominant contribution due to initial-final-state interference. Then, assuming that both initial-state and final-state particles emit photons, the corrected cross section at LL can be written as

$$\mathrm{d}\sigma_{xx}^{(LL \geq 1)}(s) = \int \left( \prod_{i=1}^{4} \mathrm{d}x_i \, D(x_i, \mu^2) \right) \mathrm{d}\sigma_{xx}^{(0)}(x_1 x_2 s) \Theta(\text{cuts}), \quad (3.18)$$

where $x_{1,2}(x_{3,4})$ refer to initial(final)-state particles. For the $\gamma\gamma$ final state, the structure function is set to 1.

It is possible to exclusively generate the transverse momentum of electrons $p_\perp$ and photons, to go beyond the strictly collinear formulation of this method [152]. This can be done by generating the angle of the $k$-th photon according to the YFS formula [153]

$$\cos\vartheta_k \propto -\sum_{i,j}^{N} \eta_i \eta_j \frac{1 - \beta_i \beta_j \cos\vartheta_{ij}}{(1 - \beta_i \cos\vartheta_{ik})(1 - \beta_j \cos\vartheta_{jk})}. \quad (3.19)$$

In the previous equation, $N$ is the total number of generated photons, $\beta_i$ is the speed of the $i$-th emitter, $\vartheta_{ij}$ is the angle between the $i$-th and the $j$-th particles and the symbol $\eta_i$ is equal to $1(-1)$ for incoming lepton (antilepton) or outgoing antilepton (lepton). Moreover, the diagonal terms of the YFS eikonal current account for terms of the kind $m_\ell^2/(p \cdot k)^2$ in the cross section. This makes the inclusion of finite mass corrections in the splitting functions unnecessary.

Through a matching procedure it is possible to combine the cross section (3.18) with corrections up to a certain fixed order, thereby also including higher-order terms without double counting. At present, the exact NLO cross section has been matched with higher-order terms

$$\mathrm{d}\sigma_{xx}^{(0)} + \mathrm{d}\sigma_{xx}^{(1)} + \mathrm{d}\sigma_{xx}^{(LL \geq 2)}. \quad (3.20)$$

According to Table 3, this amounts to calculating the first two rows and the first column.

In BABAYAGA@NLO [67], it is performed as follows. With no loss of generality, one can write the LL exponentiation for the emission of a single leg as

$$\mathrm{d}\sigma_{xx}^{(LL \geq 1)} = \Delta(Q^2, \varepsilon) \sum_{k=0}^{\infty} \frac{1}{k!} \mathcal{M}_{xx\gamma_\varepsilon^k}^{(LL)} \, \mathrm{d}\Phi_k, \quad (3.21)$$

where $\mathcal{M}_{xx\gamma_\varepsilon^k}^{(LL)}$ is the squared tree-level amplitude for the emission of $k$ photons with energy fraction larger than $\varepsilon$ in LL approximation and $\mathrm{d}\Phi_k$ is the exact $k + 2$-body phase space. The expansion of (3.21) at order $\mathcal{O}(\alpha)$

$$\mathrm{d}\sigma_{xx}^{(LL@1)} = \left[ 1 - \frac{\alpha}{2\pi} \log \frac{Q^2}{m_\ell^2} \int_0^{1-\varepsilon} \mathrm{d}z \frac{1 + z^2}{1 - z} \right] \mathcal{M}_{xx}^{(0)} \, \mathrm{d}\Phi_0 + \mathcal{M}_{xx\gamma_\epsilon}^{(LL)} \, \mathrm{d}\Phi_1$$
$$\equiv \left[ 1 + C^{(LL@1)}(\varepsilon) \right] \mathcal{M}_{xx}^{(0)} \, \mathrm{d}\Phi_0 + \mathcal{M}_{xx\gamma_\varepsilon}^{(LL)} \, \mathrm{d}\Phi_1, \quad (3.22)$$

does not coincide with the exact

$$\mathrm{d}\sigma_{xx}^{(1)} = \left[1 + C^{(1)}(\varepsilon)\right] \mathcal{M}_{xx}^{(0)} \, \mathrm{d}\Phi_0 + \mathcal{M}_{xx\gamma_\varepsilon}^{(0)} \, \mathrm{d}\Phi_1 \,. \tag{3.23}$$

The coefficient $C^{(1)}(\varepsilon)$ contains all the virtual and soft (i.e., energy fraction less than $\varepsilon$) real squared matrix elements at $\mathcal{O}(\alpha)$ and $\mathcal{M}_{xx\gamma_\varepsilon}^{(0)}$ is the exact tree-level amplitude with the emission of a photon with energy larger than $\varepsilon$. It is crucial that $C^{(LL@1)}$ have the same logarithmic structure as $C^{(1)}$ and $\mathcal{M}_{xx\gamma_\varepsilon}^{(LL)}$ have the same singular behaviour as $\mathcal{M}_{xx\gamma_\varepsilon}^{(0)}$. This makes it possible to define two finite and IR- and collinear-safe coefficients

$$
\begin{aligned}
F_{SV} &= 1 + \left(C^{(1)} - C^{(LL@1)}\right) \,, \\
F_H &= 1 + \frac{\mathcal{M}_{xx\gamma_\varepsilon}^{(0)} - \mathcal{M}_{xx\gamma_\varepsilon}^{(LL)}}{\mathcal{M}_{xx\gamma_\varepsilon}^{(LL)}} \,.
\end{aligned}
\tag{3.24}
$$

These two correction factors can be made explicit in (3.23), which becomes

$$\mathrm{d}\sigma_{xx}^{(1)} = F_{SV}\big(1 + C^{(LL@1)}\big)\mathcal{M}_{xx}^{(0)} \, \mathrm{d}\Phi_0 + F_H \mathcal{M}_{xx\gamma_\varepsilon}^{(LL)} \, \mathrm{d}\Phi_1 \,. \tag{3.25}$$

Thus, the master formula (3.21), can be improved by writing the resummed matched cross section

$$\mathrm{d}\sigma_{xx}^{(0)} + \mathrm{d}\sigma_{xx}^{(1)} + \mathrm{d}\sigma_{xx}^{(LL\geq 2)} = F_{SV} \, \Delta\big(Q^2,\varepsilon\big) \sum_{k=0}^{\infty} \frac{1}{k!} \left(\prod_{i=1}^{k} F_{H,i}\right) \mathcal{M}_{xx\gamma_\varepsilon^k}^{(LL)} \, \mathrm{d}\Phi_k \,, \tag{3.26}$$

where the $i$ subscript refers to the $i$-th emitted photon. The expansion at $\mathcal{O}(\alpha)$ of (3.26) exactly corresponds to (3.23). Moreover, the resummation of higher-order terms is conserved without any double counting.

### 3.2.3 YFS resummation

The Yennie–Frautschi–Suura theorem [153] details how logarithms associated with soft photons, both real and virtual, can be resummed to all orders, which in turn renders the entire perturbative expression IR finite order-by-order. This subsequent expression can be interpreted as a subtraction scheme, which allows us to include higher-order IR finite corrections in a systematic form. Another feature of the YFS theorem is how it treats the multi-photon phase space. In contrast to the collinear approach of an electron structure function, the YFS approach explicitly generates resolved photons, with a resolution criterion given by an energy and angle cut-off. In so doing, the full kinematic structure of scattering events is reconstructed, which leads to a straightforward implementation of the YFS method as both cross-section calculator and an event generator. Of course, the same properties can be achieved in the collinear picture with the supplementation of an appropriate parton shower algorithm. In essence, the YFS method allows us to take the cross-section expression for a generic process with the addition of infinitely many photons, both real and virtual, and rewrite it into an IR-finite expression, suitable for implementation in a Monte Carlo event generation. For the specific case of $e^+e^- \to xx$ considered here, where $xx$ represents any two-particle final state, the cross section is given by[3]

$$\mathrm{d}\sigma_{xx}^{\mathrm{YFS}} = \sum_{k=0}^{\infty} \frac{1}{k!} \, \mathrm{d}\Phi_Q \left[\prod_{i=1}^{k} d\Phi_i^\gamma\right] (2\pi)^4 \, \delta^4 \left(\sum_{a=1}^{2} p_a - \sum_{b=3}^{4} p_b - \sum_{c=1}^{k} p_c^\gamma\right) \left|\sum_{n=0}^{\infty} \mathcal{A}_{xx\gamma^k}^{(n)}\right|^2 \,, \tag{3.27}$$

---

[3]For clarity and ease of understanding, the notation used in this section has been adapted from the standard YFS format used in [154].

where $n$ denotes the number of virtual photons (for a process which is tree-level at LO this is equal to the number of loops in QED), and $\gamma^k$ denotes the emission of $k$ real photons.

If we consider the addition of one virtual photon, the amplitude will factorise, in the soft limit, as

$$\mathcal{A}_{xx}^{(1)} = \alpha \mathcal{B} A_0^0 + A_0^1 \,, \tag{3.28}$$

where $\alpha$ is the QED coupling, and the remainders $A_k^n$ are the IR-subtracted residuals from the emission of $n$ virtual photons ($k$ is the number of real photons, which have associated IR divergences; these will be subtracted later). Therefore, $A_0^0 = \mathcal{A}_{xx}^{(0)}$ is the LO matrix element. The remainder $A_k^n$ is of relative size $\mathcal{O}(\alpha^{n+k/2})$ with respect to the LO amplitude. We introduce the factor $\mathcal{B}$, which is an integrated off-shell eikonal encoding the universal soft-photon limit. For a single dipole, $\mathcal{B}$ is given by

$$\mathcal{B}_{ij} = -\frac{i}{8\pi^3} Z_i Z_j \theta_i \theta_j \int \frac{\mathrm{d}^4 p_\gamma}{p_\gamma^2} \left( \frac{2p_i \theta_i - p_\gamma}{p_\gamma^2 - 2(p_\gamma \cdot p_i)\theta_i} + \frac{2p_j \theta_j + p_\gamma}{p_\gamma^2 + 2(p_\gamma \cdot p_j)\theta_j} \right)^2 , \tag{3.29}$$

where $Z_i$ and $Z_j$ are the charges of particles $i$ and $j$ in units of the positron charge, and $\theta_{i,j} = 1\,(-1)$ for final (initial) state particles.

If we consider the corrections due to up to two virtual photons, we see

$$\begin{aligned}
\mathcal{A}_{xx}^{(0)} &= A_0^0, \\
\mathcal{A}_{xx}^{(1)} &= A_0^1 + \alpha \mathcal{B} A_0^0, \\
\mathcal{A}_{xx}^{(2)} &= A_0^2 + \alpha \mathcal{B} A_0^1 + \frac{(\alpha \mathcal{B})^2}{2!} A_0^0 \,.
\end{aligned} \tag{3.30}$$

This generalises to any number $n$ of virtual photons,

$$\mathcal{A}_{xx}^{(n)} = \sum_{r=0}^{n} A_0^{n-r} \frac{(\alpha \mathcal{B})^r}{r!} \,, \tag{3.31}$$

and resumming all virtual photon emissions therefore gives

$$\sum_{n=0}^{\infty} \mathcal{A}_{xx}^{(n)} = \exp(\alpha \mathcal{B}) \sum_{n=0}^{\infty} A_0^n \,. \tag{3.32}$$

Due to the Abelian nature of QED, this factorisation can be further generalised to produce squared matrix elements that include any number of additional real photon emissions, such that

$$\left| \sum_{k=0}^{\infty} \mathcal{A}_{xx\gamma^k}^n \right|^2 = \exp(2\alpha \tilde{\mathcal{B}}) \left| \sum_{n} A_k^n \right|^2 . \tag{3.33}$$

To show this, we consider first the case of a single real photon. This can be expressed as

$$\frac{1}{2(2\pi)^3} \left| \sum_{n=0}^{\infty} A_1^n \right|^2 = \tilde{S}(p_\gamma) \left| \sum_{n=0}^{\infty} A_0^n \right|^2 + \sum_{n=0}^{\infty} \tilde{\beta}_1^n(p_\gamma) \,. \tag{3.34}$$

In this expression, all the singularities, due to the emission of soft real photons, are contained within the eikonal,

$$\tilde{S}(p_\gamma) = \sum_{i,j} \frac{\alpha}{4\pi^2} \, Z_i Z_j \theta_i \theta_j \left( \frac{p_i}{p_i \cdot p_\gamma} - \frac{p_j}{p_j \cdot p_\gamma} \right)^2 , \tag{3.35}$$

while the remaining term $\tilde{\beta}_k^n$ are the IR-finite residuals. Extracting all real-emission soft photon divergences through eikonal factors, the squared matrix element for any $k$ real emissions, summed over all possible virtual photon corrections, can be written as

$$
\left(\frac{1}{2(2\pi)^3}\right)^k \left|\sum_{n=0}^{\infty} A_k^n\right|^2
$$

$$
= \quad \tilde{\beta}_0 \prod_{i=1}^{k}\left[\tilde{S}(p_i^\gamma)\right] + \sum_{i=1}^{k}\left[\frac{\tilde{\beta}_1(p_i^\gamma)}{\tilde{S}(p_i^\gamma)}\right]\prod_{j=1}^{k}\left[\tilde{S}(p_i^\gamma)\right] + \sum_{\substack{i,j=1\\i<j}}^{k}\left[\frac{\tilde{\beta}_2(p_i^\gamma,p_j^\gamma)}{\tilde{S}(p_i^\gamma)\tilde{S}(p_j^\gamma)}\right]\prod_{l=1}^{k}\left[\tilde{S}(p_l^\gamma)\right] + \dots
$$

$$
+ \tilde{\beta}_{k-1}(p_1^\gamma,\dots,p_{i-1}^\gamma,p_{i+1}^\gamma,\dots,p_k^\gamma)\sum_{i=1}^{k}\tilde{S}(p_i^\gamma) + \tilde{\beta}_k(p_1^\gamma,\dots,p_k^\gamma). \quad (3.36)
$$

Within this expression, all $\tilde{\beta}_i$ are free from all IR divergences due to either real or virtual photon emissions. For convenience, we have introduced the following notation,

$$
\tilde{\beta}_k = \sum_{n=0}^{\infty} \tilde{\beta}_k^n, \quad (3.37)
$$

in which we suppress the summation over virtual contributions. To recombine all terms into an expression for the inclusive cross section and facilitate the cancellation of all IR singularities, it is useful to define an unresolved region $\Omega$ in which the kinematic impact of any real photon emission is unimportant and the photon itself is undetectable. Integrating over this unresolved real emission phase space gives the integrated on-shell eikonal $\tilde{\mathcal{B}}$, defined by

$$
2\alpha\tilde{\mathcal{B}}(\Omega) = \int \frac{\mathrm{d}^3 p_\gamma}{p_\gamma^0}\,\tilde{S}(p_\gamma)\big[1 - \Theta(p_\gamma,\Omega)\big], \quad (3.38)
$$

which contains all IR poles due to real soft photon emission. With these corrections, we can express the YFS theorem for an arbitrary process as

$$
\mathrm{d}\sigma^{\mathrm{YFS}} = \sum_{k=0}^{\infty}\frac{e^{Y(\Omega)}}{k!}\,\mathrm{d}\Phi_Q\left[\prod_{i=1}^{k}\mathrm{d}\Phi_i^\gamma\,\tilde{S}(p_i^\gamma)\,\Theta(p_i^\gamma,\Omega)\right]\left(\tilde{\beta}_0 + \sum_{j=1}^{k}\frac{\tilde{\beta}_1(p_j^\gamma)}{\tilde{S}(p_j^\gamma)} + \sum_{\substack{j,l=1\\j<l}}^{k}\frac{\tilde{\beta}_2(p_j^\gamma,p_l^\gamma)}{\tilde{S}(p_j^\gamma)\tilde{S}(p_l^\gamma)} + \cdots\right),
$$

$$
(3.39)
$$

with the YFS form factor

$$
Y(\Omega) = 2\alpha\sum_{i<j}\left[\mathcal{B}_{ij} + \tilde{\mathcal{B}}_{ij}(\Omega)\right]. \quad (3.40)
$$

Therein, all IR singularities originating from real and virtual soft photon emission, contained in $\tilde{\mathcal{B}}$ and $\mathcal{B}$ respectively, cancel, leaving a finite remainder.

The most sophisticated variant of YFS is the Coherent Exclusive Exponentiation (CEEX) [155] currently only available in KKMCEE, see Section 4.3. Contrary to the standard YFS approach (EEX), the CEEX variant of YFS exponentiation is implemented in terms of the IR-free $\tilde{\beta}_i$ residuals constructed at the amplitude level, rather than using the spin-summed squared $\tilde{\beta}_i$. CEEX treats correctly to infinite order not only IR cancellations but also QED interferences and narrow resonances. The YFS resummation (EEX) has been implemented in SHERPA [156], see Section 4.7. The YFS approach has also been used in several earlier codes, such as BHWIDE [157], BHLUMI [158] and it has been implemented in HERWIG [159,160]. In a more recent development, the combination of the YFS exponentiation with collinear factorisation has been considered [161].

## 3.3 Internal hadrons

This section provides a brief overview on the methods used to calculate nonperturbative hadronic corrections for the processes (1.1) and (1.2) at NNLO accuracy. These corrections correspond to Feynman diagrams with insertions of Green's functions built from any number of electromagnetic currents

$$j_{\text{em}}^{\mu}(x) = \sum_q Q_q \bar{q}(x) \gamma^{\mu} q(x) \tag{3.41}$$

in pure QCD, where the sum runs over the three lightest quarks $q$ with charge $Q_q$. Correlation functions with an odd number of currents vanish as a consequence of Furry's theorem. Up to NNLO, the only contributing Green's functions are therefore the HVP tensor

$$\Pi_h^{\mu\nu}(q) = \text{\scriptsize\wedge\!\!\wedge\!\!\bigcirc\!\!\wedge\!\!\wedge} = i\,e^2 \int \mathrm{d}^4x\, e^{-iqx} \langle 0|T\{j_{\text{em}}^{\mu}(x) j_{\text{em}}^{\nu}(0)\}|0\rangle = \Pi_h(q^2)(g^{\mu\nu} q^2 - q^\mu q^\nu) \tag{3.42}$$

and the HLbL tensor

$$\Pi_h^{\mu\nu\lambda\sigma}(q_1, q_2, q_3) = \begin{matrix} q_1 \\ q_2 \\ q_3 \end{matrix} \text{\scriptsize\bigcirc}$$

$$= i \int \mathrm{d}^4x\,\mathrm{d}^4y\,\mathrm{d}^4z\, e^{-i(q_1\cdot x + q_2\cdot y + q_3\cdot z)} \langle 0|T\{j_{\text{em}}^{\mu}(x) j_{\text{em}}^{\nu}(y) j_{\text{em}}^{\lambda}(z) j_{\text{em}}^{\sigma}(0)\}|0\rangle. \tag{3.43}$$

The HLbL tensor is only relevant for the process $e^+ e^- \to \gamma\gamma$ at NNLO. For our main processes it only contributes beyond NNLO. In the following, we therefore mainly focus on HVP corrections.

At the energy scales of the experiments discussed in Section 2, these hadronic correlation functions are nonperturbative and cannot be calculated in perturbation theory. Instead they either have to be computed in lattice gauge theory or extracted from experimental data making use of unitarity and analyticity. We only discuss the latter approach. In the case of the HVP function, the optical theorem (unitarity) gives a relation between its imaginary part and the production cross section of hadrons

$$\text{Im}\,\Pi_h = -\frac{s}{4\pi\alpha}\, \sigma(e^+ e^- \to \gamma^* \to \text{had}). \tag{3.44}$$

Analyticity of the HVP function then allows for its reconstruction via the dispersion integral

$$\frac{\Pi_h^{\text{ren}}(q^2)}{q^2} = \frac{1}{\pi} \int_{4m_\pi^2}^{\infty} \frac{\mathrm{d}z}{z} \frac{\text{Im}\Pi_h(z)}{(z - q^2 - i0)}, \tag{3.45}$$

where we introduced the renormalised (subtracted) $\Pi_h^{\text{ren}}(q^2) \equiv \Pi_h(q^2) - \Pi_h(0)$. There is a long history of specialised tools such as `alphaQED` [162–164] and `HVPTools` [165] that compute the HVP function based on the described approach. The recent most precise tabulations of the full VP are provided by the KNT group (v3.1, 2022) [166], Jegerlehner (hadr5x23 from `alphaQED23` package, 2023) [167], and Novosibirsk (NSK) VP (v2.9, 2022) [168, 169].

Similarly, the HLbL tensor has been studied in detail in the context of the anomalous magnetic moment of the muon, $a_\mu^{\text{HLbL}}$, see [6] (including a corresponding prediction for the HLbL contribution based on [170–182]). In particular, a dispersive formalism was developed that, in analogy to (3.45), aims at reconstructing the entire HLbL tensor in terms of its singularities [183–185]. The dominant intermediate states that contribute in such a dispersive representation are pseudoscalar poles $P = \pi^0, \eta, \eta'$ [171, 173, 174, 186], two-meson cuts [172, 187], and the corresponding $S$-wave

rescattering, the latter encoding the effects of scalar resonance exchanges [188], e.g., $S = f_0(500)$, $f_0(980)$, and $a_0(980)$. Subleading corrections arise from axial-vector exchanges, $A = f_1(1285)$, $f_1(1420)$, and $a_1(1260)$, see [189–193] for their incorporation into the dispersive formalism, and tensor mesons [194], where, in the case of $T = f_2(1270)$, corrections beyond a narrow-width approximation can again be captured by two-meson rescattering [195, 196].

In the application to $e^+e^-$ reactions, a dispersive approach to HLbL could be established similarly to the application for $a_\mu^{\mathrm{HLbL}}$, with a few important differences. On the one hand, the dispersive frameworks for $a_\mu^{\mathrm{HLbL}}$ are formulated either in four-point kinematics with fixed $t = q_2^2$ [172, 185], or, alternatively, directly in the soft-photon limit [193, 194], whereas for $e^+e^- \to \gamma\gamma$, one requires doubly-virtual four-point kinematics with Mandelstam variables different from the photon virtualities. Simplifications compared to the case of $a_\mu^{\mathrm{HLbL}}$ occur in the $s$-channel: first, $P$ and $S$ contributions are helicity suppressed, and thus likely irrelevant for the $e^+e^-$ channel due to the resulting scaling with $m_e$. Second, the on-shell two-photon coupling vanishes for $A$ on account of the Landau–Yang theorem [197, 198]. Accordingly, the first resonantly enhanced contribution without helicity suppression in $e^+e^- \to \gamma\gamma$ is due to tensor states in the $s$-channel. Away from the resonance region, the HLbL contribution is likely dominated by $t$-channel exchanges, which are neither helicity suppressed nor forbidden by Landau–Yang, by the charged pion box, as well as short-distance contributions [170, 176, 199, 200]. To estimate the impact of the HLbL contribution, a simple strategy could thus rely on a narrow-width approximation for the $f_2(1270)$, with transition form factors constrained by data on $\gamma\gamma^{(*)} \to \pi\pi$ [201, 202] and their asymptotic behaviour [189], to estimate $s$-channel effects. The $t$-channel contributions could be estimated, e.g., for $P = \pi^0$ using the known transition form factor, whereas the pion box could be evaluated with input for the pion VFF [185]. A dispersive treatment beyond these leading contributions would need to deal with the issue of kinematic singularities in the tensor decomposition, which so far has only been addressed in the kinematic configuration of $a_\mu^{\mathrm{HLbL}}$.

Once the HVP and HLbL tensors have been determined through data, their contribution to the processes can be computed perturbatively. This is trivial for diagrams where the hadronic tensors factorise from loop integrals. In the case of nonfactorisable hadronic corrections, the loop integral has to be performed semi-numerically since the hadronic tensors are only available as numerical routines. In the case of HVP corrections this is rather straightforward. After inserting the dispersive representation (3.45) into the amplitude and exchanging the order of integration, the loop integral can be performed with standard methods where $z$ simply acts as a photon mass. This yields a one-dimensional integration over $z$ that can be done numerically. This *dispersive approach* [203] has been used to calculate NNLO hadronic corrections to the muon decay [204, 205], Bhabha scattering [206–208], and muon–electron scattering [209, 210]. An alternative approach to the dispersive technique is the *hyperspherical method* [211–213]. It is based on the idea that the loop momentum routing can be chosen such that the HVP function only depends on the square of the loop momentum. In this representation the angular integration can be performed analytically. The remaining one-dimensional radial integral over the HVP function can then be computed numerically. Contrary to the dispersive approach, the integration is over space-like momenta in this case. Narrow hadronic resonances are therefore avoided, which represents a significant advantage. However, the method entails subtle analytic continuations from the Euclidean to the Minkowski region, which significantly complicates the application of the method to processes with time-like momentum transfer.

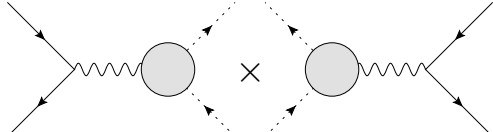

Figure 7: LO contribution to the $e^+e^- \to \pi^+\pi^-$ cross section.

## 3.4 Hadronic final states

### 3.4.1 Strategies for radiative corrections to low-energy hadronic amplitudes

In processes with hadronic final states, the nonperturbative nature of the strong interactions at low energies significantly complicates the description of radiative corrections compared to purely leptonic processes. On the one hand, the strong dynamics itself requires nonperturbative techniques, on the other hand the additional interplay with QED corrections constitutes a highly nontrivial problem. In principle, the QED interaction can still be treated perturbatively. At each order in QED perturbation theory, nonperturbative matrix elements show up, which can be defined in pure QCD. One complication is the fact that the parameters of pure QCD (i.e., strong interaction in the limit of vanishing electromagnetic coupling) need to be expressed in terms of the physical parameters in a scheme- and scale-dependent matching [214–216]. Since the masses of the hadronic states depend on electromagnetic effects, QED corrections also lead to shifted thresholds. In practice, one often chooses a different approach than a strict separation of QED and QCD effects, e.g., the LO HVP contribution to $g-2$ is conventionally defined to be the one-particle-irreducible hadronic two-point function including photonic corrections [6]. The separation by topologies leads to a fully consistent treatment of higher-order insertions of HVP corrections, but does no longer follow a strict power counting in $\alpha$.

For the description of the nonperturbative strong dynamics and the interaction of the hadronic states with photons, different techniques and approximations with varying levels of rigour exist. Lattice QCD provides an approach based on first principles. The restriction of lattice simulations to Euclidean space severely impacts the applicability to Minkowski-space processes [217, 218], but recent years have seen promising conceptual developments. At very low energies, the interaction of the pseudoscalar Goldstone bosons with photons and leptons is described by ChPT [219–223]. While this effective-field-theory framework consistently describes structure-dependent radiative corrections and has been applied to a variety of low-energy processes [224–232], higher-order contributions involve poorly known nonperturbative low-energy constants. Furthermore, the energies of interest are often beyond the range of applicability of ChPT. Dispersion theory provides an approach to the strong dynamics based on the fundamental principles of unitarity, analyticity, and crossing, establishing model-independent relations between different measurable processes. While for the leading hadronic processes in (1.1) dispersion theory is well established [233, 234], dispersive treatments of the corresponding radiative corrections are only partially available or under development [58, 235–237], as will be discussed in more detail in Section 3.4.2, Section 3.4.3, and Section 3.4.4. Often, a combination of dispersive techniques with input from ChPT and, if available, lattice QCD provides the ideal description of radiative corrections in hadronic processes by taking into account as many theoretical and experimental constraints as possible [238, 239].

In practice, for many processes the more rigorous approaches mentioned above are either not applicable or not yet available. In this case, one typically resorts to models of the interaction of hadrons with photons. A hadronic model is a phenomenological approach that is not fundamentally derived from QCD and comes with intrinsic uncertainties that cannot be systematically reduced and can be assessed only by comparison with more rigorous approaches. For $e^+e^- \to \pi^+\pi^-$, exist-

ing Monte Carlo tools are all based on different levels of hadronic modelling, which we categorise in the following.

**"sQED"** Scalar QED: this is the $U(1)$ gauge theory describing the renormalisable interaction of (pseudo-)scalars with photons. Its use is motivated by the fact that sQED corresponds to the subset of renormalisable interactions of LO ChPT. In contrast to ChPT, sQED assumes pions to be pointlike. Note that LO ChPT includes further nonrenormalisable interactions, which at the loop level lead to UV divergences that necessitate the introduction of NLO ChPT counterterms. For loop processes including photons and leptons, these are the counterterms of [221–223]. sQED avoids the chiral counterterms by neglecting the effects of pion structure.

**"F×sQED"** Scalar QED multiplied by form factors for external virtualities [240]: this prescription preserves the nice properties of scalar QED, such as renormalisability and the cancellation of IR divergences, while keeping the dominant form-factor effects, but only the virtualities corresponding to the $e^+e^-$ and $\pi^+\pi^-$ invariant masses can be taken into account in this way.

**"GVMD"** Generalised vector-meson dominance [57]: the photon–pion coupling is modified by (a sum of) vector-meson Breit–Wigner propagators. This approach allows one to include form-factor effects within loops. However, the form factor does not fulfill analyticity constraints and the use of a constant vector-meson width introduces unphysical subthreshold imaginary parts.

**"FsQED"** Form-factor sQED [58, 183, 185]: in this approach, form factors are kept everywhere by isolating pion poles in dispersion theory. This allows one to account for form-factor effects inside loop integrals in a well-defined manner as long as the required matrix elements are sufficiently simple so that pion poles are uniquely defined.

**"full"** Full hadronic matrix elements: the inclusion of a more complete description of the hadronic matrix elements as explained above is not yet available in existing Monte Carlo tools. However, as we will explain below, we expect even FsQED to be insufficient for more complicated matrix elements, such as $\pi^+\pi^- \to 3\gamma^*$. This is on the one hand due to the fact that the definition of pion poles becomes ambiguous as it depend on the choice of dispersion relations, on the other hand FsQED does not account for other potentially important intermediate states beyond pion poles.

### 3.4.2 Structure-dependent radiative corrections in $e^+e^- \to \pi^+\pi^-$

The photon-inclusive process $e^+e^- \to \pi^+\pi^-(\gamma)$ is used to measure the pion electromagnetic form factor in $e^+e^-$ experiments in scan mode. At leading order, the cross section shown in Figure 7 is directly proportional to the squared modulus of the pion vector form factor (VFF) in pure QCD, defined by the current matrix element

$$\langle 0|j_\mu(0)|\pi^+(p_+)\pi^-(p_-)\rangle = (p_+ - p_-)_\mu F_\pi(s)\,, \qquad s = (p_+ + p_-)^2\,. \tag{3.46}$$

Comprehensive dispersive analyses of the form factor are available [233, 236, 241–243]. The representation of [233] can be used in the energy range up to $1\,\text{GeV}$ and reads

$$F_\pi(s) = \Omega_1^1(s) \times G_\omega(s) \times G_{\text{in}}^N(s)\,, \tag{3.47}$$

where

$$\Omega_1^1(s) = \exp\left\{\frac{s}{\pi} \int_{4m_\pi^2}^\infty ds' \frac{\delta_1^1(s')}{s'(s'-s)}\right\} \tag{3.48}$$

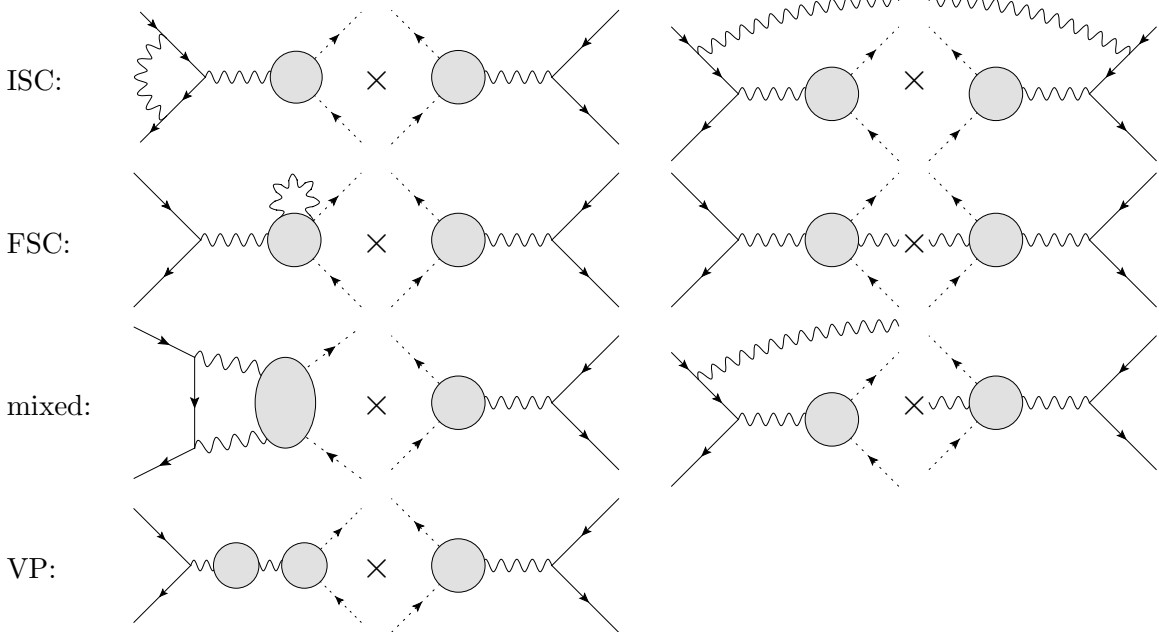

Figure 8: Representative NLO contributions to the $e^+e^- \to \pi^+\pi^-(\gamma)$ cross section: the first line shows ISC, consisting of initial-state virtual corrections and ISR. The second line shows FSC, consisting of virtual and real photon corrections to the pion VFF. The third line shows mixed corrections and the last additional VP.

is the Omnès function with the elastic $\pi\pi$-scattering $P$-wave phase shift $\delta_1^1(s)$ as input [244, 245]. The second factor in (3.47) describes the resonantly enhanced isospin-breaking $\rho$–$\omega$ interference effect

$$G_\omega(s) = 1 + \frac{s}{\pi} \int_{9m_\pi^2}^{\infty} ds' \frac{\operatorname{Im} g_\omega(s')}{s'(s'-s)} \left( \frac{1 - \frac{9m_\pi^2}{s'}}{1 - \frac{9m_\pi^2}{m_\omega^2}} \right)^4, \quad g_\omega(s) = 1 + \epsilon_\omega \frac{s}{(m_\omega - \frac{i}{2}\Gamma_\omega)^2 - s}, \quad (3.49)$$

and additional inelastic contributions are parameterised by a conformal polynomial $G_{\mathrm{in}}^N(s)$ with a cut starting at the $\pi^0\omega$ threshold. The dispersive representation (3.47) respects the constraints of unitarity and analyticity and it depends on only a few free parameters. Although there are systematic differences between the different cross-section measurements, the dispersive representation can be fit with a good $\chi^2$ to almost all major experimental low-energy data sets, see [7] for an overview.

At NLO, within a strict fixed-order counting in $\alpha$ the radiative corrections can be split into the different gauge-invariant contributions introduced in Section 3.1, as shown in Figure 8. All ISC scaling as $\sim q_e^4 q_h^2$, where $q_h$ denotes a generic coupling of the photon to hadronic states, can be treated as described in Section 3.1 and pose no additional conceptual difficulty. The FSC, however, scaling as $\sim q_e^2 q_h^4$ and shown in the second line of Figure 8, include two additional nonperturbative matrix elements: the amplitude $\gamma^* \to \pi^+\pi^-\gamma$ for the real corrections, related by crossing to the pion Compton tensor, as well as virtual corrections to the VFF itself. So far, these corrections were treated in sQED, which leads to the inclusive cross section due to FSC

$$\sigma(e^+e^- \to \pi^+\pi^-(\gamma)) = \left[ 1 + \frac{\alpha}{\pi}\eta(s) \right] \sigma(e^+e^- \to \pi^+\pi^-), \quad (3.50)$$

where the correction factor $\eta(s)$ determined in sQED can be found in [246–249]. The reliability of the sQED determination of the factor $\eta(s)$ is currently under investigation, by making use of

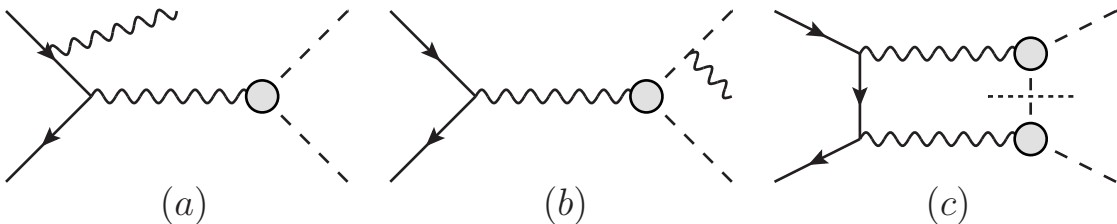

Figure 9: Diagrams for the forward–backward asymmetry $A_{\text{FB}}$ in $e^+e^- \to \pi^+\pi^-$, taken from [58]. The short-dashed line in $(c)$ indicates that only the pion-pole contribution is kept, not the general pion Compton tensor.

dispersion theory for the hadronic matrix elements entering the FSC. The pion Compton amplitude was analysed dispersively in [185,195,196,250–253]. Preliminary results for the virtual corrections to the VFF are available in [235], where no large effects beyond the sQED approximation were observed, which can be explained at least partially by the dominance of the IR-enhanced effects. The correction factor $\eta(s)$ is also used to capture the dominant radiative effects in $e^+e^- \to K^+K^-(\gamma)$, see [254], where, in addition, the comparison to the Sommerfeld–Gamow–Sakharov factor [255–257]

$$Z(s) = \frac{\pi\alpha}{\sigma_K(s)} \frac{1 + \alpha^2/\big(4\sigma_K^2(s)\big)}{1 - \exp\big(-\pi\alpha/\sigma_K(s)\big)}, \qquad \sigma_K = \sqrt{1 - \frac{4m_K^2}{s}}, \tag{3.51}$$

was studied, resumming threshold-enhanced higher-order effects in $\alpha$. However, already for the $K^+K^-$ channel the non-Coulomb corrections contained in $\eta(s)$ prove more important.

The third line in Figure 8 shows the mixed corrections $\sim q_e^3 q_h^3$, where there is an additional photon coupling to both initial and final states. It is important to note that this contribution comes with an odd number of photon couplings to the hadronic states, hence it is $C$-odd and only contributes to the differential cross section and the charge or forward–backward asymmetry, whereas these mixed corrections cancel in the total cross section. Due to one additional photon coupling to the hadronic final state, the mixed corrections to $e^+e^- \to \pi^+\pi^-$ only depend on two hadronic matrix elements: the pion VFF itself as well as the pion Compton tensor. The charge asymmetry was studied in [57] within the GVMD model and in [58] within the FsQED approximation, i.e., taking into account only the dispersively defined pion-pole contribution to the pion Compton tensor, see Figure 9. Both analyses came to similar conclusions and found relatively large deviations from the point-like sQED approximation and a much better agreement with measurements of the asymmetry. Generalisations of these treatments should be possible by including effects in the Compton tensor beyond the pion pole, in particular two-pion rescattering in $S$- and $D$-waves as determined in [185, 195, 196].

Finally, the last line in Figure 8 involves the VP corrections $\sim q_e^2 q_h^2 \Pi^{(1)}$. Conventionally, this correction is included in what is reported by $e^+e^-$ experiments as the form factor, while it is excluded from the so-called "bare cross section," which only includes FSC.

### 3.4.3 Structure-dependent radiative corrections in $e^+e^- \to \pi^+\pi^-\gamma$

Radiative-return experiments measure the process $e^+e^- \to \pi^+\pi^-\gamma$ with a hard photon in the final state. As in (3.8), the LO cross section consists of ISC, FSC, and a mixed contribution, shown in Figure 10. The first line shows the ISC $\sim q_e^4 q_h^2$, which can be determined in standard QED and only depends on the VFF. The FSC $\sim q_e^2 q_h^4$ shown in the second line of Figure 10 only depends on the singly-virtual pion Compton tensor and in principle could be determined using a dispersive

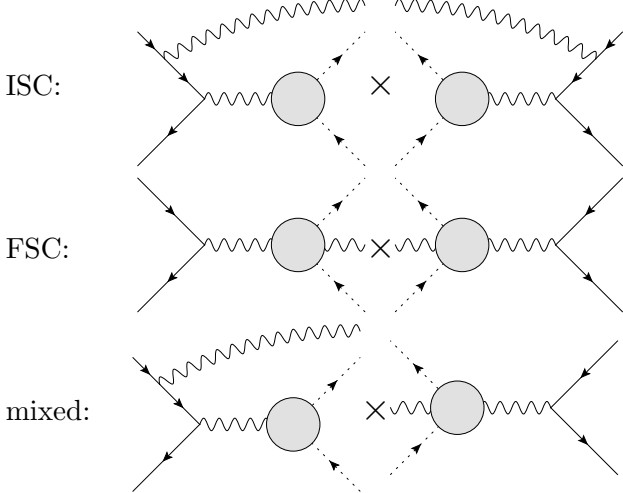

Figure 10: LO contributions to the $e^+e^- \to \pi^+\pi^-\gamma$ cross section.

description of the hadronic sub-amplitude [185, 195, 196, 252]. The mixed contribution $\sim q_e^3 q_h^3$ shown in the third line of Figure 10 depends on both the pion VFF and the Compton tensor. As before, it is $C$-odd and contributes to the charge asymmetry of the process.

The radiative corrections at NLO can again be split into ISC, FSC, and mixed contributions. By definition, the ISC part only depends on the pion VFF as hadronic sub-amplitude and we do not discuss it further. In the following, we also ignore VP corrections, which do not introduce new hadronic sub-amplitudes compared to the LO contributions. The virtual corrections to the radiative amplitude are shown in Figure 11, where we omit the two ISC diagrams. We see that the radiative corrections depend on the pion Compton tensor, which appears in the first diagram (2PE-ISR), the five-point amplitude $\gamma^*\gamma^* \to \pi^+\pi^-\gamma$, which is part of the second diagram (2PE-FSR), as well as on the virtual corrections to the pion VFF and the pion Compton tensor. At present, all these corrections have been studied within sQED, multiplied by external form factors [240, 258, 259].

As previously discussed, dispersive approaches are available for the doubly-virtual pion Compton tensor [185, 195, 196], which consist of the dispersive pion pole and potentially vector-meson left-hand cuts, together with a unitarisation of the two-pion rescattering in the $S$- and $D$-waves. While in the case of $e^+e^- \to \pi^+\pi^-$, [58] includes the pure pion pole in the loop diagram shown in Figure 9, a similar study for the radiative process is not yet available. The rescattering contribution has been studied neither in $e^+e^- \to \pi^+\pi^-$ nor in the radiative process. However, at least at low energies below $1\,\mathrm{GeV}$ the pion pole is known to be a decent approximation of the full Compton amplitude [6]. The radiative corrections to the pion VFF studied dispersively in [235] could be included both in $e^+e^- \to \pi^+\pi^-$ as well as in the radiative process.

At present, no dispersive study exists for radiative corrections to the Compton tensor. The complicated five-point amplitude $\gamma^*\gamma^*\gamma \to \pi^+\pi^-$ appears in a new dispersive approach to HLbL, proposed in [194]. There, the gauge-invariant Bardeen–Tung–Tarrach tensor decomposition into structures free of kinematic singularities was performed, a necessary starting point for a dispersive analysis, which is work in progress in the context of HLbL. There, it is needed only in the next-to-next-to-soft approximation for the real photon [194], whereas in the present context the amplitude is required with a hard real photon.

The NLO cross section is obtained from the three-body phase-space integral over the product of the LO and NLO amplitudes, together with the four-body phase-space integral over the square

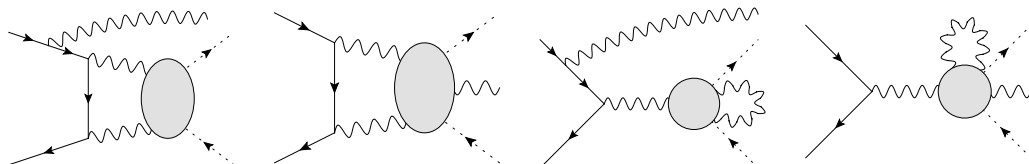

Figure 11: Representative virtual NLO contributions to the $e^+e^- \to \pi^+\pi^-\gamma$ amplitude, omitting ISC. The ISR photon can be attached anywhere at the lepton line.

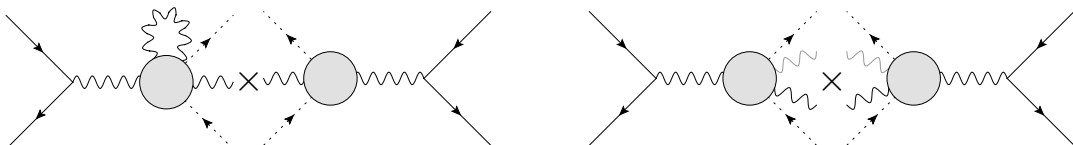

Figure 12: FSC NLO contributions to the $e^+e^- \to \pi^+\pi^-\gamma(\gamma)$ cross section. The grey photon is soft and the grey blobs denote the hadronic matrix elements.

of the doubly-real-emission amplitude. We follow the classification of [240, 258, 259] of the different contributions to the cross section into IR-safe categories, neglecting again pure ISC and VP contributions.

Figure 12 shows pure FSC $\sim q_e^2 q_h^6$, where the soft real photon is shown in grey. Going beyond a model estimate might be difficult: these corrections require virtual-photon and soft-real corrections to the pion Compton tensor, which have not yet been studied dispersively. However, the pure FSC are generally expected to be small [258]. If the conclusions of the preliminary dispersive studies of FSC for the nonradiative process [235] can be transferred, they are well approximated by sQED estimates.

The mixed corrections can be split up further. In Figure 13, we show the $C$-odd contributions, which can be divided into three gauge-invariant and IR-finite sets. The first one, scaling as $\sim q_e^5 q_h^3$, involves a 2PE-ISR diagram in the virtual corrections. The hadronic matrix elements in this class are the pion VFF and the pion Compton tensor. The second class in Figure 13, scaling as $\sim q_e^3 q_h^5$, involves a virtual correction with a 2PE-FSR diagram. The required hadronic sub-amplitudes are the pion Compton tensor, as well as the complicated five-point amplitude $\gamma^*\gamma^* \to \pi^+\pi^-\gamma$. The last class in Figure 13 also scales as $\sim q_e^3 q_h^5$ and contains virtual corrections to the pion VFF and the pion Compton tensor, together with real corrections that cancel the IR divergences of the virtual corrections. Although these different corrections partially involve hadronic matrix elements that are difficult to compute beyond model approximations, the overall $C$-odd contributions will cancel if the experimental cuts are symmetric under change conjugation. Measurements of the charge asymmetry in the radiative process could be used to constrain the hadronic sub-processes [258].

We now turn to $C$-even mixed corrections. The class of diagrams shown in Figure 14 corresponds to virtual and soft real corrections to the pion VFF, overall scaling as $\sim q_e^4 q_h^4$. The hard real photon is emitted from the initial state. For these corrections, the dispersive analysis of [235] should be applied.

This finally leaves us with the class of diagrams shown in Figure 15, which form an IR-finite subset scaling as $\sim q_e^4 q_h^4$. In this class, the hard real photon interferes between initial and final states. While these corrections were previously assumed to be negligible [258], they were later computed within the sQED approximation [259], multiplied by external form factors. The studies of the charge asymmetry in $e^+e^- \to \pi^+\pi^-$ of [57, 58] however indicates that larger corrections

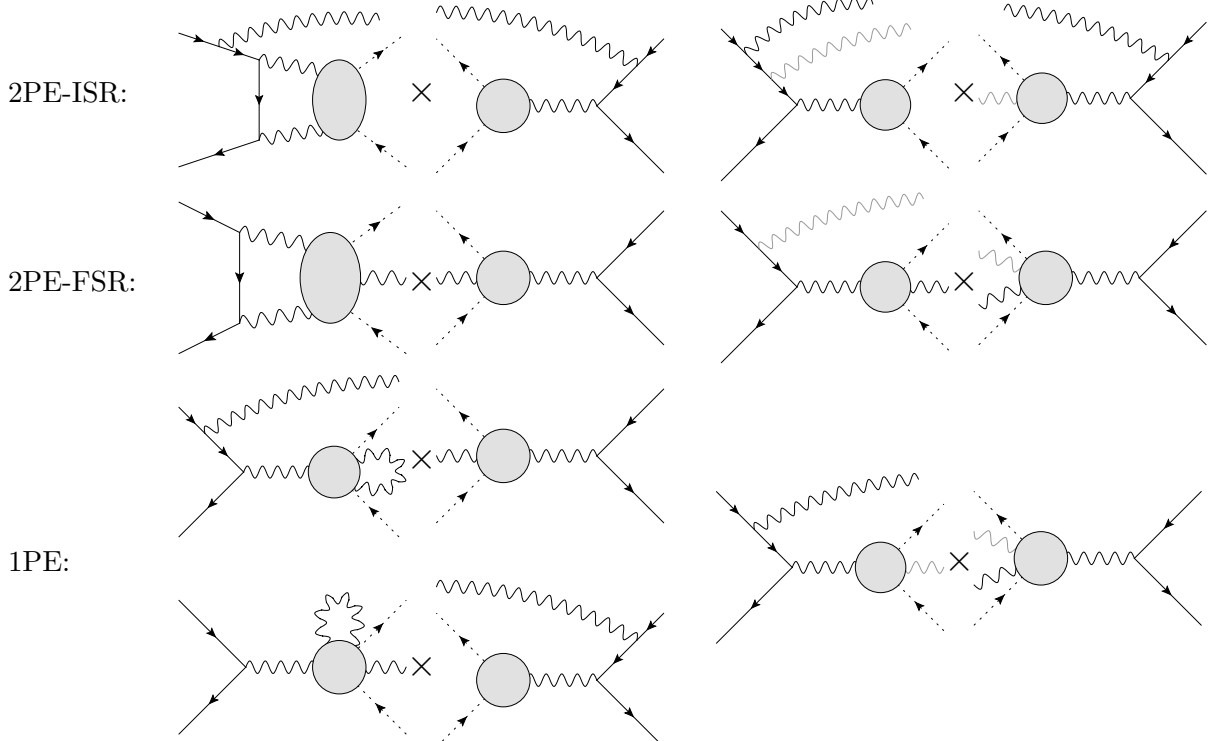

Figure 13: Representative $C$-odd mixed NLO contributions to the $e^+e^- \to \pi^+\pi^-\gamma(\gamma)$ cross section. The grey photon is soft, the grey blobs denote the hadronic matrix elements, and the ISR photons can be attached anywhere at the lepton line.

beyond the point-like approximation are possible [2]. In the case of the 2PE-ISR corrections, a study similar to [57, 58] should be performed. At least at low energies, the pion Compton tensor can be well approximated by a dispersively defined pion pole.

In the case of the 2PE-FSR correction, the situation is more challenging. It is unclear if even a dispersively defined pion-pole approximation is sufficient, since the two-pion state is in a $P$-wave. Final-state rescattering effects giving rise to the $\rho$-resonance call for the inclusion of a unitarisation of the pole contribution. Furthermore, due to the hard real photon additional internal resonance enhancement is possible. At present, these effects have not been estimated in a model-independent framework.

### 3.4.4 $\quad e^+e^- \to \pi^+\pi^-\pi^0$

In analogy to (3.46), the general matrix element for $\gamma^*(q) \to \pi^+(p_+)\pi^-(p_-)\pi^0(p_0)$ is defined by

$$\langle 0|j_\mu(0)|\pi^+(p_+)\pi^-(p_-)\pi^0(p_0)\rangle = -\epsilon_{\mu\nu\alpha\beta}p_+^\nu p_-^\alpha p_0^\beta \mathcal{F}(s,t,u;q^2)\,, \tag{3.52}$$

with $q = p_+ + p_- + p_0$, $s = (p_+ + p_-)^2$, $t = (p_- + p_0)^2$, $u = (p_+ + p_0)^2$, subject to the constraint $s+t+u = 3m_\pi^2 + q^2$. In contrast to the pion electromagnetic form factor, the function $\mathcal{F}(s,t,u;q^2)$ does not only depend on the virtual-photon momentum, but also on the invariant masses of all two-pion subsystems; expressed in terms of resonances, in addition to the isoscalar vector three-pion resonances such as $\omega(782)$, $\phi(1020)$, ..., also two-pion resonances of isospin 1 and odd angular momentum play an important role, most significantly the $\rho(770)$. The total $e^+e^- \to 3\pi$ cross

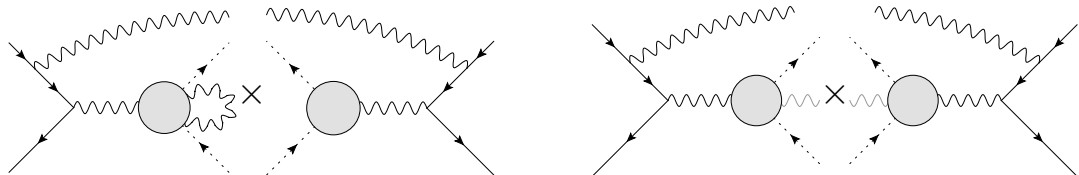

Figure 14: Representative $C$-even mixed NLO contributions to the $e^+e^- \to \pi^+\pi^-\gamma(\gamma)$ cross section, corresponding to virtual and soft real corrections to the pion VFF with an additional initial-state hard photon.

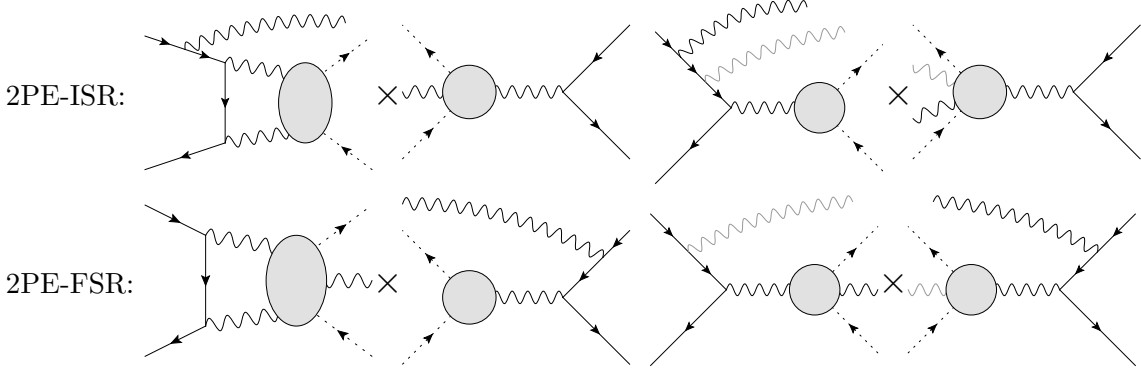

Figure 15: Representative $C$-even mixed NLO contributions to the $e^+e^- \to \pi^+\pi^-\gamma(\gamma)$ cross section, where the hard photon interferes between initial and final states.

section is obtained according to

$$\sigma_{e^+e^- \to 3\pi}(q^2) = \alpha^2 \int_{s_{\min}}^{s_{\max}} ds \int_{t_{\min}}^{t_{\max}} dt \, \frac{s[\kappa(s,q^2)]^2(1-z_s^2)}{768\,\pi\,q^6} \, |\mathcal{F}(s,t,u;q^2)|^2 \,, \qquad (3.53)$$

with the relevant kinematic quantities

$$z_s = \cos\theta_s = \frac{t-u}{\kappa(s,q^2)} \,, \qquad \kappa(s,q^2) = \sigma_\pi(s)\lambda^{1/2}(q^2,m_\pi^2,s) \,,$$

$$\lambda(x,y,z) = x^2 + y^2 + z^2 - 2(xy+yz+xz) \,, \qquad \sigma_\pi(s) = \sqrt{1-\frac{4m_\pi^2}{s}} \,, \qquad (3.54)$$

and the boundaries of the Dalitz-plot integration given by

$$s_{\min} = 4m_\pi^2, \qquad s_{\max} = \left(\sqrt{q^2}-m_\pi\right)^2 \,,$$

$$t_{\min/\max} = (E_-^* + E_0^*)^2 - \left(\sqrt{E_-^{*2}-m_\pi^2} \pm \sqrt{E_0^{*2}-m_\pi^2}\right)^2 \,, \qquad (3.55)$$

and

$$E_-^* = \frac{\sqrt{s}}{2} \,, \qquad E_0^* = \frac{q^2-s-m_\pi^2}{2\sqrt{s}} \,. \qquad (3.56)$$

The function $\mathcal{F}(s,t,u;q^2)$ can be decomposed into simpler ones using a reconstruction theorem when neglecting discontinuities in the two-pion invariant masses of angular momentum 3 and higher, leading to [234, 260–262]

$$\mathcal{F}(s,t,u;q^2) = \mathcal{F}(s,q^2) + \mathcal{F}(t,q^2) + \mathcal{F}(u,q^2) \,, \qquad (3.57)$$

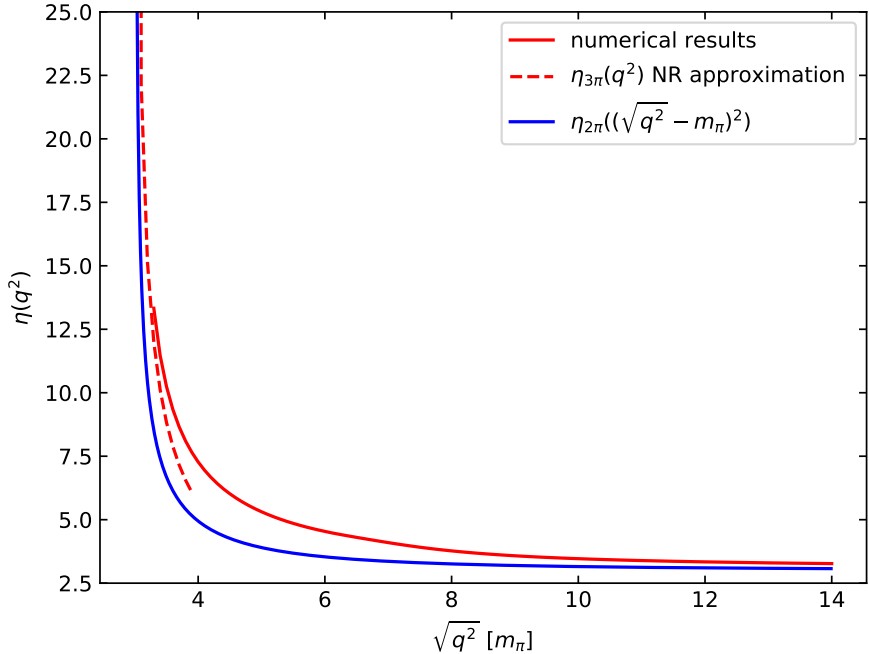

Figure 16: IR-enhanced radiative corrections for $e^+e^- \to 3\pi$ incorporated in $\eta_{3\pi}$ (red solid). The numerical result is compared to a nonrelativistic approximation (red dashed) and the $2\pi$ analogue $\eta_{2\pi} \equiv \eta$ shifted to the $3\pi$ threshold (blue solid). Figure adapted from [237].

which corresponds to a symmetrised partial-wave expansion stopping beyond $P$-waves. The neglected $F$-wave contributions are irrelevant before production of the $\rho_3(1690)$ becomes feasible, which happens around $\sqrt{q^2} \approx 1.83\,\text{GeV}$ [234, 260, 263]. The $s$-dependence of the functions $\mathcal{F}(s, q^2)$ can be calculated dispersively in the Khuri–Treiman formalism [264], which takes two-pion $P$-wave rescattering in all subchannels into account consistently, using the corresponding phase shift. They require a $q^2$-dependent subtraction or normalisation function that incorporates information on all $3\pi$ resonances, which can be modelled in a way consistent with analyticity and fit to experimental data [234, 237, 262]. Its value at $q^2 = 0$ is constrained by the chiral anomaly [265–267], including quark-mass corrections [261, 268, 269].

The assessment of radiative corrections to the $\gamma^* \to 3\pi$ matrix element is complicated by the fact that it is not accessible in the framework of sQED: it is given, at LO in the chiral expansion, by the Wess–Zumino–Witten anomaly [270, 271], which, due to its derivative structure, does not constitute a renormalisable theory, but requires counterterms to render loop corrections UV finite. ChPT calculations [272] or other approaches to radiative corrections near threshold [273, 274] furthermore miss the resonance dynamics of the $3\pi$ final state and are therefore phenomenologically insufficient in the energy region relevant for HVP studies.

While, as a consequence, an unambiguous determination of structure-dependent radiative corrections in the $3\pi$ channel would be a formidable task, the IR-enhanced effects have been studied in [237], owing to the observation that these constitute the by far dominant contributions to the radiative corrections in the $2\pi$ channel [252]. Due to the dominance of pion–pion $P$-waves in the partial-wave decomposition of the $3\pi$ final state, it was observed that the sQED correction factor $\eta(s)$, cf. (3.50), can similarly be applied to the $\pi^+\pi^-$ subsystem in $3\pi$, whose invariant mass is, however, $s$, not $q^2$. The cross section including IR-enhanced radiative corrections can therefore be

written as

$$
\begin{aligned}
&\sigma_{e^+e^-\to 3\pi(\gamma)}(q^2) \\
&= \alpha^2 \int_{s_{\min}}^{s_{\max}} \mathrm{d}s \int_{t_{\min}}^{t_{\max}} \mathrm{d}t\, \frac{s[\kappa(s,q^2)]^2(1-z_s^2)}{768\,\pi\,q^6}\, |\mathcal{F}(s,q^2)+\mathcal{F}(t,q^2)+\mathcal{F}(u,q^2)|^2 \left(1+\frac{\alpha}{\pi}\eta(s)\right), \quad (3.58)
\end{aligned}
$$

and an effective analogue correction factor $\eta_{3\pi}(q^2)$ be defined as

$$
1 + \frac{\alpha}{\pi}\eta_{3\pi}(q^2) = \frac{\sigma_{e^+e^-\to 3\pi(\gamma)}(q^2)}{\sigma^{(0)}_{e^+e^-\to 3\pi}(q^2)}, \tag{3.59}
$$

which has to be calculated numerically. Near threshold $q^2 = 9m_\pi^2$, $\eta_{3\pi}(q^2)$ shows a Coulomb-pole-like divergence $\propto (q^2-9m_\pi^2)^{-1/2}$ inherited from the divergence in $\eta(s) \propto (s-4m_\pi^2)^{-1/2}$, whose coefficient can be calculated analytically from a nonrelativistic expansion [237]. The resulting correction factor is reproduced in Figure 16, compared to its nonrelativistic approximation and the (shifted) $\eta(s)$ correction.

# 4 Generators and integrators

In this section we will briefly review each of the seven codes that are used in this comparison, in alphabetical order. We focus mainly on the physics effects included in each generator, using the language defined in Section 3. This will allow us to study the relative impact of different contributions in Section 5.

The seven codes discussed during Phase I are AFKQED, BABAYAGA@NLO, KKMCEE, MCGPJ, MCMULE, PHOKHARA, and SHERPA. A list of responsible contact persons for each code can be found on the website at

https://radiomontecarlow2.gitlab.io/code-responsible.html

Other codes may be added in Phase II.

## 4.1 AFKQED

AFKQED is a Monte Carlo event generator that simulates radiative processes, i.e., those with a hard photon

$$
e^+e^- \to X\gamma, \tag{4.1}
$$

where $X$ in the final state can represent a $\mu^+\mu^-$ pair, or a number of hadronic processes, including those with pions, kaons and protons.

The simulation of most hadronic processes is based on the EVA event generator [275, 276], which was originally used to generate $2\pi\gamma$ and $4\pi\gamma$ final states. The code was developed in Fortran mainly by V. Druzhinin at Novosibirsk, and further final states were implemented in a modular way. AFKQED was used by the BaBar collaboration in 2009 for acceptance determination [11]. In later analyses, the BaBar collaboration used PHOKHARA for this purpose.

The zipped file `AfkQed.tar.gz` in the repository

https://gitlab.com/radiomontecarlow2/monte-carlo-results/-/tree/root/codes/
afkqed

|  |  | $e^+e^- \to \mu^+\mu^-\gamma$ | $e^+e^- \to \pi^+\pi^-\gamma$ |
|---|---|---|---|
| ISC | LO | Exact matrix elements [278] | EVA [276] |
| | NLO | Collinear structures [93, 129] | |
| FSC | LO | Exact matrix elements [278] including ISR-FSR interference | No FSR at LO |
| | NLO | PHOTOS [279] | |
| HVP | | None, leptonic only [278], leptonic [278] + hadronic, or NSK VP [168, 169] | Customisable form factor |

Table 4: Overview of the modules implemented in AFKQED for $e^+e^- \to X^+X^-\gamma$ processes.

contains all modules necessary for execution, including the stand-alone test programme `Afkcrs.F` in Fortran, the `GfiAfkQed.cc` code written in `C++` which was used within the BaBar software environment, and the stand-alone code `afkrun.cpp`, which was used for the purpose of this report.

For most hadronic channels, including $\pi^+\pi^-\gamma$, the simulation is based on the approach developed in [276]. For the hadronic processes $p^+p^-\gamma$, $3\pi^+3\pi^-\gamma$, $2\pi^+2\pi^-2\pi^0\gamma$, the Bonneau–Martin formula is used [277]. For the $\pi^+\pi^-\gamma$ process, the VFF is implemented through the F×sQED approach described in Section 3.4.1. For the $\mu^+\mu^-\gamma$ process, the calculation of the hard photon matrix element is based on the Born cross section formulae in [278], and the simulation includes ISC, FSC, and mixed corrections.

In AFKQED, the LO hard photon can be generated at large angle, within the detector acceptance range. At higher orders, additional ISR photons are generated first, through the structure function method described in Section 3.2.2. Such photons are assumed to be collinear to the $e^+$ or $e^-$ beams, so they do not have the angular distribution of (3.19), and photon jets radiated from each beam are resummed as a single photon along that beam. After one or two collinear ISR photons have been simulated, the final state, including a photon with an angular distribution, is generated in the boosted system. Additional FSR photons are generated with the PHOTOS package [279]. In all cases, the photon generated at a large angle is assumed to be hard and is not complemented by soft photon emission corrections. For the $\mu^+\mu^-\gamma$ process, HVP is included, where in the original code it could be chosen to include leptonic contributions only (from [278]), or both hadronic ($\rho + \omega + \phi$) and leptonic. By default, the current version on the repository employs the Novosibirsk (NSK) VP [168, 169] applied in the Dyson resummed form.

AFKQED events are built in a factorised way, meaning that the modules displayed in Table 4 for additional ISR or FSR, or for HVP, can be switched on or off at the user's discretion.

For the simulation of different scenarios, the following parameters in the centre-of-mass frame can be specified: the centre-of-mass energy, the minimum energy and polar angle of the LO hard photon, and — in case photons from collinear structures are switched on — the minimum squared invariant mass of observed particles in the final state (i.e., the system composed of hadrons or muons and of the photon emitted to large angle).

AFKQED is not suitable for use in cases when the LO photon at a large angle is not treated as the hardest photon after selection cuts, because this photon is not supplemented by soft and virtual corrections. In the KLOE-like small-angle scenario (see Section 5.3), the collinear structures for

additional ISR will almost always be the hardest photons in all events, whereas the LO photon at a large angle will be a correction. In the KLOE-like large-angle scenario (see Section 5.4), if PHOTOS is switched on, the additional FSR might be detected as the main ISR photon, and for this reason we do not include FSR corrections from PHOTOS in the KLOE-like scenarios. In the BESIII-like scenario, the issue of the full treatment of the LO photon is addressed by applying a soft photon cut of 400 MeV at the generation level, together with the experimental requirement that only one hard photon be detected at large angle. This will force the LO photon to be considered as the main radiative photon. Finally, in the $B$ scenario there is a requirement for a photon energy of at least 3 GeV at the generator and experimental photon selection levels. We also require the invariant mass of all observed final-state particles to be greater than 8 GeV. Together, these requirements allow AFKQED to work well for these scenarios.

## 4.2   BabaYaga@NLO

BABAYAGA@NLO is a fully differential Monte Carlo event generator which was developed for high-precision simulations of QED processes at flavour factories up to $\sqrt{s} \simeq 10$ GeV [67, 151, 152, 280–283]. It has been used by most of the experimental collaborations for luminosity determination and other physics studies. It is based on the matching of exact $\mathcal{O}(\alpha)$ corrections with a QED parton shower algorithm. The BABAYAGA@NLO code allows the generation of fully exclusive events with exact kinematics and phase space. Via the parton shower, the generator is also able to exactly reconstruct all the generated photon momenta, conserving both the total four-momentum and the mass shell conditions of all the particles. A detailed description of the parton shower algorithm as implemented in BABAYAGA@NLO can be found in Section 3.2.2. In addition to resumming all leading logarithmic terms of the form $\alpha^n L_c^n$ for $n \geq 2$, the matching procedure takes into account the next-to-leading logarithmic terms of the form $\alpha^n L_c^{n-1}$ for $n = 2$, as well as the leading mass terms proportional to $m_i^2/(p_i \cdot q)$. The resummation is extended to the ISC, FSC, mixed, and VPC contributions for all channels.

Table 5 shows a list of all the processes one can calculate using BABAYAGA@NLO. All the SM processes are calculated at NLO with a consistent matching of a QED parton shower, without double counting, as specified by the NLOPS tag in Table 5. Pion pair production is implemented according to three of the possible alternative approaches discussed in Section 3.4, namely F×sQED, GVMD, and FsQED [283].

Different pion VFF parametrisations are included for comparative studies. Any other parametrisation can be added to the code, either as an analytical function or as a numerical table. In addition to the SM processes, it is possible to study the production of a vector boson according to the dark matter models discussed in [284–286]. This calculation has been performed at LOPS [282].

The accuracy of the code is estimated to be at the 0.1% level for calculations at NLOPS for the $2 \rightarrow 2$ processes with typical event selections at flavour factories. It is important to note that for radiative events, i.e., $2 \rightarrow 2 + \gamma$ processes the accuracy of BABAYAGA@NLO is LL, estimated at the $\mathcal{O}(1\%)$ level.

BABAYAGA@NLO can generate both weighted and unweighted events. In the former case, each event comes with a different weight which has to be carried throughout the whole detector simulation and analysis. This procedure allows for a very fast generation and a better Monte Carlo error convergence. In the case of unweighted generation, all events come with the same weight. This means that they are distributed according to the cross section. However, the generation of unweighted events can be very slow due to the unweighting procedure, which will be inefficient if there are large fluctuations in weights.

To calculate one-loop amplitudes, for $\mu^+\mu^-$ and $\pi^+\pi^-$ production, BABAYAGA@NLO is in-

|  | $e^+e^- \to e^+e^-$ | $e^+e^- \to \mu^+\mu^-$ | $e^+e^- \to \gamma\gamma$ | $e^+e^- \to \pi^+\pi^-$ |
|---|---|---|---|---|
| Order | NLOPS | NLOPS | NLOPS | NLOPS |
| Accuracy | $\mathcal{O}(0.1\%)$ | $\mathcal{O}(0.1\%)$ | $\mathcal{O}(0.1\%)$ | $\mathcal{O}(0.1\%)$ |

Table 5: All the processes that are calculated by BABAYAGA@NLO, the order of their calculation, and their estimated accuracy. NLOPS means the matching of the full NLO calculation with a PS algorithm, as described in Section 3.2.2.

terfaced to Collier [118] (and optionally to LoopTools [287, 288]), and mass effects are fully included. The mass of the external particles is instead neglected in the non-IR parts $\delta_{\mathrm{virt}}^{\mathrm{non-IR}}$ for the $e^+e^- \to e^+e^-$ and $e^+e^- \to \gamma\gamma$ processes. For all implemented processes, BABAYAGA@NLO keeps the full dependence of the masses in the kinematics and in the real photon radiation processes.

Infrared divergences are treated by giving a vanishingly small mass to real and virtual photons. Moreover, the real photon contribution is computed using the phase space slicing technique. This amounts to imposing an arbitrary cutoff $\varepsilon$ on the photon energy and then separating the two different phase space regions where the photon energy is smaller or larger than the cutoff. The phenomenological results obtained with BABAYAGA@NLO do not depend on the choice of the infrared separator $\varepsilon$.

In BABAYAGA@NLO, the hadronic contribution to the vacuum polarisation can be included by using various routines, for example by KNT [166], Jegerlehner [167, 179] or NSK [169].

## 4.3  KKMC

KKMCEE is a Monte Carlo event generator applicable for electron–positron annihilation processes, accounting for multiple photon emission:

$$e^-e^+ \to f\bar{f} + n\gamma, \qquad f = \mu, \tau, \nu, u, d, s, c, b, \qquad n = 0, 1, \ldots \infty \qquad (4.2)$$

KKMCEE allows the generation of fully exclusive events with exact kinematics and phase space, which is crucial for realistic data analysis. The Monte Carlo integration over the phase-space is executed through the FOAM algorithm [289]. The generator can produce weighted and unweighted events depending on the user's needs. KKMCEE is extensively used in data analysis for most existing electron colliders, such as BES and Belle, and in research for future electron colliders like FCCee, CLIC, and ILC.

Effects due to photon emission from incoming beams and outgoing fermions are calculated in QED up to second order in $\alpha_{\mathrm{QED}}$, including all interference effects, within Coherent Exclusive Exponentiation (CEEX) [155, 290], which is based on Yennie–Frautschi–Suura exponentiation [153], see Section 3.2.3. CEEX not only treats infrared cancellations and the description of soft photons correctly to infinite order, but also includes the correct description of QED interferences and narrow resonances. Contrary to the standard YFS approach, in which higher-order corrections are effected at the level of the amplitude squared, the CEEX scheme is devised in terms of spin amplitudes and can therefore account for ISR-FSR interferences. A detailed description of CEEX and the older EEX scheme can be found in [155, 291]. In addition to higher order QED corrections the weak corrections are provided by Dizet library [292], and polarised $\tau$ decays are included using the TAUOLA programme [293].

The current version of the code, written in C++, was released in 2022 [294] and it is publicly available at

## 4.4 MCGPJ

The Monte Carlo Generator with Photon Jets (MCGPJ) [295] was developed more than 20 years ago by the Dubna–Novosibirsk collaboration, based on [278, 296]. The main purpose of the generator is to simulate processes of electron–positron annihilation to two particles for the energy scan experiments, specifically for the CMD-2 experiment at the VEPP-2M electron–positron collider. This includes Bhabha scattering, production of two charged pions, kaons, lepton pairs, and $3\gamma$. The programme is based on the exact $\mathcal{O}(\alpha)$ differential amplitude supplemented by logarithmically enhanced higher order contributions. The exact NLO amplitude includes emission from the pions and kaons in the pointlike (sQED) approximation. The LL effects are taken into account using the structure function approach in the collinear approximation, where multi-photon jet objects are emitted along the electrons' direction of motion. For the structure function, MCGPJ uses the jet energy emission function $D(z)$ [130, 278], which is an extension of (3.12) to $\mathcal{O}(\alpha^3)$ and to $\mathcal{O}(\eta^2)$ in the exponent. The original paper [130] also includes part of the NLL corrections due to virtual and real $e^+e^-$ pair production. In MCGPJ, the jet emission is performed only from the light electron and positron lines, including final states in the Bhabha process, while for heavier final-state particles no final-state resummation is applied.

Unlike in many other generators, the matching of the resummation to higher-order corrections is achieved additively in MCGPJ. Event generation is performed independently using the exact NLO amplitude or by simultaneous jet emissions from all electron and positron lines. An energy cut $\Delta = \Delta\epsilon/\epsilon$ is applied to the real photon which is generated with NLO accuracy, and isolation criteria are applied to electrons and positrons, meaning the NLO-accurate photon emission does not occur within a narrow cone of width $\theta_0$ around electrons/positrons. Most of the photon radiation comes from the narrow collinear region with angles around $\sim 1/\gamma \sim \sqrt{1-\beta^2}$ (which justifies the use of the collinear jet approximation in most cases), and the auxiliary parameter $\theta_0$ is taken $\sim 1/\sqrt{\gamma}$ to make it outside of the collinear region. The corresponding one-photon contribution, integrated over the volume defined by $\Delta$ and $\theta_0$, is subtracted from jets when only one jet is above the $\Delta\epsilon$ cut energy. The corresponding soft region of all jets is also matched with the one-photon soft and virtual corrections. While the final result does not depend on $\Delta$ or $\theta_0$, a physically-motivated selection of these auxiliary parameters helps to suppress negative weighted events.

The theoretical precision for the integrated cross section is estimated to be better than 0.2%. Until now, MCGPJ was the only generator capable of simulating two-pion events with such precision for scan experiments.

The collinear approximation can limit the precision of the differential cross section prediction in tails, especially in kinematic regions where events with two resolved photon emissions (which are not included in the NLO amplitude) can pass experimental cuts. To improve this situation, the original version of MCGPJ was modified to take into account the angular distribution of photon emissions. The angle distribution of photon jets was generated according to the one-photon approximation

$$f(c = \cos(\theta), x = \omega/E) \sim \frac{1}{1-\beta c} - \frac{1-x}{1+(1-x)^2}\frac{1-\beta^2}{(1-\beta c)^2}\,, \tag{4.3}$$

where $\theta$ is the angle of the jet relative to the radiating particle and $x$ is the emitted energy ratio. This modification was implemented only for the Bhabha process and led to the ability to run the event generation in partially weighted mode, with a significant presence of normalised negative weighted events.

|  | $e^+e^- \to e^+e^-(\gamma)$ | $e^+e^- \to \mu^+\mu^-(\gamma)$ | $e^+e^- \to \pi^+\pi^-(\gamma)$ |
|---|---|---|---|
| NLO | exact amplitude | | |
|  |  |  | F×sQED |
| next LL orders | SF with angles | collinear structures | |
|  | ISC+FSC | ISC | ISC |
| VP | Novosibirsk VP table | | VFF |
| 2PE |  |  | F×sQED, GVMD |
|  |  |  | or dispersive |

Table 6: Overview of the corrections to $e^+e^- \to X^+X^-$ that are available in MCGPJ

The latest version of MCGPJ also includes the contribution of the two-photon exchange diagram calculated in the above F×sQED approximation, taken into account either using the GVMD model or in the dispersive formalism [57,58]. The correction is pre-tabulated in $\sqrt{s}$ and the $\theta_{\mathrm{avg}}$ of final-state particles. The VPC are included according to the Novosibirsk (NSK) compilation [1,168,169].

## 4.5 McMule

In its current version, the Monte Carlo framework McMule [297]

https://mule-tools.gitlab.io

is a parton-level integrator that performs fully differential fixed-order QED calculations to high precision. The available processes are $ee \to XX$ with $X \in \{e, \mu, \pi\}$ and others such as $\ell p \to \ell p$ [298] and $\mu e \to \mu e$ [299], all at NNLO accuracy. Further, McMule also supports electroweak and polarisation effects for selected processes such as $ee \to \mu\mu$ [300]. $2 \to 3$ QED processes, i.e., those with an additional photon in the final state, are implemented at NLO accuracy.

McMule builds upon modern methods developed for higher-order QCD calculations, adapting them to the case of QED with massive fermions.

On the one hand, this translates into a simplification, as the presence of fermion masses makes the handling of infrared divergences solvable to all orders in $\alpha$. In particular, infrared singularities are regularised in $d = 4 - 2\epsilon$ dimensions, and dealt with in phase-space integrations with the FKS$^\ell$ method [301], an adaption of FKS subtraction [302,303] to any loop order in QED. The extension of the subtraction scheme to all orders is made possible by the absence of collinear singularities in QED, allowing for the exponentiation of soft singularities, as in YFS [153] (see Section 3.2.3). The FKS$^\ell$ method introduces an unphysical parameter as a bookkeeper of contributions with different real-photon multiplicities, and the independence of the final result of said parameter is used as a strong consistency check. The method is exact and does not require splitting photon radiation into a soft and a hard part, nor does it introduce a photon mass as a regulator, since dimensional regularisation is instead used. If all squared matrix elements are known, the numerical integration of FKS$^\ell$-arranged squared matrix elements, combined with a measurement function to define IR-safe observable(s), results in a fully-differential Monte Carlo code.

On the other hand, while the presence of non-vanishing fermion masses simplifies the infrared structure, the additional (though typically small) scales can cause serious complications. This is

particularly relevant for the evaluation of loop amplitudes and the numerical stability of phase-space integrations.

For one-loop amplitudes McMule relies on external libraries such as OpenLoops [119,120] and Collier [118]. Two-loop matrix elements are the main bottleneck since no general approach exists, especially because of the higher number of scales. This issue can be mediated using massification [304–307], if the corresponding two-loop matrix element with vanishing fermion masses is known, and if some external fermions have small masses compared to all other scales in the process, such as the common case $m_e^2 \ll Q^2$. Using this small-mass expansion, all terms that are not polynomially suppressed in $m_e^2/Q^2$, i.e., the logarithmically enhanced ones as well as the constant terms, can be recovered.

In particular, the massive result can be related to the massless one via the factorisation formula

$$\mathcal{M}_n(m_e) = \left( \prod_j Z(m_e) \right) \times S \times \mathcal{M}_n(m_e = 0) + \mathcal{O}(m_e), \tag{4.4}$$

where the product is over all external fermion legs with a small mass $m_e$, and $n$ refers to any non-radiative final state such as $n = mm = \mu^+\mu^-$. The factorisation is a consequence of collinear and soft degrees of freedom factorising at leading power, as shown in soft-collinear effective theory (SCET) [308–310]. Fermions with small masses correspond to highly-energetic particles in the external states, thus defining a collinear sector in SCET and contributing one power of the massification constant $Z$. This process-independent factor does not depend on any hard scale and, apart from a trivial factorised $m_e$ dependence, is a constant now known up to three loop [311]. The soft part $S$ is process dependent, starts at NNLO, and is obtained from fermionic corrections only [305]. However, it is often advantageous to compute the fermionic corrections of two-loop squared matrix elements semi-numerically as it includes the HVP data. This has the added advantage of eliminating $S$ in (4.4), thus rendering massification completely universal.

The comparison of massified calculations with their equivalent full-mass calculation has allowed the quantification of the massification error. At the level of the differential NNLO cross section, massification introduces an error of the order $\alpha^2 \times 10^{-3}$ [299]. It was also verified, for the case of the muon decay, that the massified result at NNLO gives a very good approximation to the result with exact $m_e$ dependence [301]. In general, applying massification corresponds at NNLO to a parametric error of order $(\alpha/\pi)^2 \, m_e^2/Q^2$, potentially multiplied by a $L_c^1$.

Non-vanishing fermion masses, in addition to problematic evaluations of loop amplitudes, cause instabilities in the numerical integration, in particular, at NNLO, for the real-virtual contributions. OpenLoops is remarkably stable in the bulk of the phase space, and even in very soft and collinear regions. However, for simultaneously extremely soft and collinear kinematics the numerical stability in double precision, with on-the-fly support in quadruple, is not sufficient. In order to overcome this problem, McMule applies next-to-soft (NTS) stabilisation, i.e. replaces the real-virtual squared matrix elements by numerically adequate squared matrix elements in the problematic soft regions of the phase space [312]. While the leading power eikonal approximation is in general not sufficiently accurate, an expansion in the soft photon energy, $E_\gamma$, up to the next-to-leading power proves to be accurate enough.

The LBK theorem [313,314] can provide the NTS limit at tree level, however for the real-virtual contribution the one-loop NTS limit has to be used. The real-virtual squared matrix element can be written as [315]

$$\mathcal{M}_{n+\gamma}^{(1)} = \left( \mathcal{E} + \mathcal{D} + \mathcal{S}^{(1)} \right) \mathcal{M}_n^{(1)} + \mathcal{O}(E_\gamma^0), \tag{4.5}$$

in terms of the non-radiative one-loop squared matrix element, where $\mathcal{E}$ is the eikonal factor, that scales as $1/E_\gamma^2$. In addition, we must include the LBK operator $\mathcal{D}$, and the one-loop contribution

| | | $e^+e^- \to e^+e^-$ [312, 319] | $e^+e^- \to \mu^+\mu^-$ [299, 300] | $e^+e^- \to \pi^+\pi^-$ [297] |
|---|---|---|---|---|
| ISC | NLO | full mass dependence | full mass dependence | full mass dependence |
| | NNLO | massified OpenLoops+NTS | full mass dependence OpenLoops+NTS | full mass dependence OpenLoops+NTS |
| FSC | NLO | full mass dependence | full mass dependence | |
| | NNLO | massified OpenLoops+NTS | full mass dependence OpenLoops+NTS | |
| mixed | NLO | full mass dependence | full mass dependence | |
| | NNLO | massified OpenLoops+NTS | massified OpenLoops+NTS | |
| VP | | HVP with `alphaQED` or NSK | HVP with `alphaQED` or NSK | HVP with `alphaQED` or NSK and VFF |

Table 7: Overview of the corrections to $e^+e^- \to X^+X^-$ that are available in McMule at NLO and NNLO. At LO and NLO all implementations have full mass dependence. Radiative processes $e^+e^- \to X^+X^-\gamma$ at NLO are deduced from NNLO results of $e^+e^- \to X^+X^-$.

$\mathcal{S}^{(1)}$, which takes into account soft virtual corrections. These contributions scale as $1/E_\gamma$. Recent results [316, 317] have shown that the LBK theorem holds at any loop order in QED, since the soft function $\mathcal{S}^{(1)}$ is one-loop exact, as well as for any number of photon emissions.

From the comparison of results obtained with NTS stabilisation with the same obtained with OpenLoops in full quadruple precision (at the cost of at least ten times longer running times), the error due to the former could be quantified, at the level of the differential NNLO cross section, to $\alpha^2 \times 10^{-2}$ [299].

Similar instabilities in the numerical integration due to the presence of non-vanishing fermion masses are found in the case of tree-level contributions with the emission of one or more photons, for example real corrections at NLO and double-real corrections at NNLO. Since the mass of the fermion is often small compared to its energy, radiative amplitudes exhibit narrow peaks: remnants of the collinear singularities (which have been regularised by the masses). A reliable integration, less hampered by such peaks, is achieved in McMule by a partitioning and tuning of the phase space [318] to directly match the collinearity with a variable of the adaptive integration algorithm.

In McMule, NLO corrections to a radiative process $ee \to XX\gamma$ are obtained as a subset of the NNLO corrections of the non-radiative process $ee \to XX$. As described in Section 3.1, this amounts to simply dropping the pure virtual terms (assuming an IR-safe observable). The availability of a non-radiative process at $N^n$LO corresponds to the availability of the related radiative process at $N^{n-1}$LO. Hence, we only describe the implemented corrections for non-radiative processes in Table 7.

McMule currently does not support any effects related to the hadronic structure of the pion in the final state beyond one-photon exchange contributions. As indicated in Table 7, only ISC and VPC are included together with a customisable pion form-factor vertex. Work to implement the remaining contributions is currently in progress.

With McMule it is possible to separately compute contributions involving the leptonic and

| $e^+e^- \to$ | Order | VP | VFF | Extras |
|---|---|---|---|---|
| $\mu^+\mu^-$ | LO | alphaQED, | | Narrow resonances |
| $\mu^+\mu^-\gamma$ | NLO with full | from [321, 322] | - | of $J/\psi$ and $\psi(2S)$ |
| | mass dependence | or NSK | | |
| $\pi^+\pi^-$ | LO | alphaQED, | F×sQED | Narrow resonances |
| $\pi^+\pi^-\gamma$ | NLO with full | from [321, 322] | choice of | of $J/\psi$ and $\psi(2S)$ |
| | mass dependence | or NSK | 3 VFF | Radiative $\phi$ decays |
| $X$ | $X \in 2\pi^0\pi^+\pi^-,\, 2\pi^+2\pi^-,\, p\bar{p},\, n\bar{n},\, K^+K^-,\, K^0\bar{K}^0,\, \pi^+\pi^-\pi^0,\, \Lambda(\to \pi^- p)\bar{\Lambda}(\to \pi^+\bar{p}),$ $\eta\pi^+\pi^-,\, \pi^0\gamma,\, \eta\gamma,\, \eta'\gamma,\, \chi_{c1} \to J/\psi(\to \mu^+\mu^-)\gamma,\, \chi_{c2} \to J/\psi(\to \mu^+\mu^-)\gamma$ | | | |

Table 8: Overview of the corrections to $e^+e^- \to X$, $e^+e^- \to X^+X^-$, $e^+e^- \to X^+X^- + \gamma$, that are available in PHOKHARA at LO and NLO.

hadronic VP, including at NNLO. The most recent version of the Fortran library alphaQED [162–164, 167], alphaQEDc23, or the NSK VP [168, 169] are employed for the evaluation of the HVP. The leptonic VP can even be split into separate leptons. However, typically McMule combines the full leptonic VP parts with the photonic corrections.

Fermionic corrections are treated differently according to how the VP is inserted in the diagrams. At NNLO, if $\Pi(Q^2)$ factorises from the rest of the amplitude, the correction reduces to quantities that have already been computed at NLO. Non-factorisable contributions need instead a special treatment, which corresponds to a dispersive [203] or hyperspherical [211–213] approach in McMule. In the dispersive approach, the VP in the original integrand is replaced by a massive photon propagator, where the photon mass is the dispersion parameter. This integral is then computed with existing one-loop tools: in McMule, Collier is used due to its stability in the presence of large cancellations for large values of the dispersion parameter. However, for extremely large values of the latter, a sufficiently precise evaluation of the kernel becomes even more difficult. This necessitates expanding the amplitude, typically using the method of regions [320].

### 4.6 Phokhara

The Monte Carlo event generator PHOKHARA was conceived to provide theoretical predictions for cross section measurements at fixed low-energy meson factories. In particular, the current version of the generator PHOKHARA 10.0 focuses on the radiative return processes $e^+e^- \to \mu^+\mu^-\gamma$ (1.2b) and $e^+e^- \to \pi^+\pi^-\gamma$ (1.2a) at NLO. These theoretical predictions include virtual and soft photon corrections to one-photon emission events and the emission of two real hard photons, accounting for the complete NLO corrections.

PHOKHARA can be downloaded from

https://looptreeduality.csic.es/phokhara/.

In addition to these flagship processes, PHOKHARA also contains a variety of hadronic production channels. We summarise in Table 8 all physical processes which are available in PHOKHARA, as well as some features of the main channels.

In the following, we briefly discuss the history and main features of the generator. The first ver-

sions of PHOKHARA were based on EVA [275], a LO Monte Carlo generator for the pion radiative-return process (1.2a), extended to include the muon process (1.2b), as well as the emission of an extra hard photon in the final state from the incoming electrons. In PHOKHARA, ISC and FSC are treated separately as independent contributions using the gauge-invariant splitting method described in Section 3.1. By switching them on or off, an accurate determination of the contributions to the physical region under consideration in the experiment can be explored. When looking at these physical observables within PHOKHARA, one can identify interesting properties of the physical process, which are useful during all phases of an experiment.

The treatment of hadrons in the final state, in particular pions, is carried out under the approximation of the pions being point-like particles described by sQED (see Figure 2). This approach, not being rigorous, is augmented by the use of a non-perturbative form factor. For further details, we refer the reader to [323, Figure 2.10].

The decomposition between perturbative and non-perturbative contributions to the pion production cross section (1.2a) motivated the addition of initial-stae radiative corrections as a first attempt in PHOKHARA to simulate emission of photons at large-angle ($\theta \gg m_e^2/s$) [107, 324, 325], and small-angle ($\theta \ll m_e^2/s$) [326] regions. To further improve theoretical predictions, for final-state radiation off the pions, PHOKHARA adopted the F×sQED approach (see Section 3.4.1). The accuracy of this approach is under scrutiny. Using these methods, PHOKHARA includes the emission of one photon from the initial state and one photon from the final state, requiring one of the photons to be hard [248, 258]. The relevant Feynman diagrams can be seen in the right column of Figure 13 and Figure 14.

Regarding other important channels for the evaluation of the hadronic vacuum polarisation, PHOKHARA considers the hadronic channels of $\pi^+\pi^-\pi^0$, kaon pairs $K^+K^-$ and $K^0\bar{K}^0$ [106, 327], nucleon pairs $p\bar{p}$ and $n\bar{n}$, and the radiative $\phi$ decay contributions to the reaction $e^+e^- \to \pi^+\pi^-\gamma$ relevant when running at $\phi$-factory energy (see Section 2.2).

The version of PHOKHARA which is employed for the Monte Carlo comparisons in Section 5 considers the complete set of Feynman diagrams that contribute to the NLO theoretical prediction of the scattering processes (1.2b) and (1.2a). On top of considering ISC and FSC virtual and real corrections, this version includes, for the first time, the gauge-invariant group of diagrams containing two virtual photons, referred to as the penta-box contribution (see Figure 5). The presence of these diagrams gives a complete NLO theoretical prediction.

In the calculation of the muon production cross section (1.2b), the evaluation of Feynman integrals appearing at intermediate steps of the calculation is achieved by the use of publicly automated software [328–331]. A more detailed discussion on the QED calculation and the implementation of these contributions in PHOKHARA can be found in [332]. From the Feynman diagrams needed at NLO, one can recognise a decomposition in terms of the couplings between the photon and leptons (analogously to (3.2)),

$$\mathcal{A}^{(0)}_{mm\gamma\gamma}\left(q_e q_m\right) = \mathcal{A}^{(0)}_{mm\gamma\gamma}\left(q_e^3 q_m\right) + \mathcal{A}^{(0)}_{mm\gamma\gamma}\left(q_e^2 q_m^2\right) + \mathcal{A}^{(0)}_{mm\gamma\gamma}\left(q_e q_m^3\right) , \tag{4.6}$$

$$\mathcal{A}^{(1)}_{mm\gamma}\left(q_e q_m\right) = \mathcal{A}^{(1)}_{mm\gamma}\left(q_e^4 q_m\right) + \mathcal{A}^{(1)}_{mm\gamma}\left(q_e^2 q_m^3\right) + \mathcal{A}^{(1)}_{\text{PB};mm\gamma}\left(q_e^3 q_m^2\right) + \mathcal{A}^{(1)}_{mm\gamma}\left(q_e^2 q_m \Pi^{(1)}\right)$$
$$+ \left(q_e \leftrightarrow q_m\right) , \tag{4.7}$$

where $\mathcal{A}^{(0)}_{mm\gamma\gamma}\left(q_e q_m\right)$ accounts for the real radiation of two photons, and $\mathcal{A}^{(1)}_{mm\gamma}\left(q_e q_m\right)$ is understood as the virtual correction to emission of a photon from a leptonic line. This decomposition can be elucidated as the following gauge invariant pieces, with $f, F \in \{e, \mu\}$ and $f \neq F$,

- $f^+f^- \to \gamma^* \to F^+F^- + \gamma$,

- $f^+f^- \to \gamma\gamma^* \to F^+F^-$,

- $f^+ f^- \to F^+ F^- + \gamma$ (only diagrams containing two virtual photons),

- Insertion of VP.

ISC and FSC real corrections in (1.2b), needed at NLO, that are accounted by $\mathcal{A}^{(0)}_{mm\gamma\gamma}(q_e q_m)$ with the emission of two hard photons are calculated in terms of helicity amplitudes. This is done, in order to overcome numerical instabilities.

For the process (1.2a) the organisation of virtual and real corrections is less evident within PHOKHARA framework, since the organisation of this calculation has not been documented. However, a similar approach to the muon channel was carried out. Additionally, because PHOKHARA makes extensive use of scalar QED, it always looks for consistency physical checks at various steps of the calculation. In particular, the study of the infrared structure of virtual and real corrections determines if the code needs to switch to running in quadruple precision.

Finally, for both (1.2b) and (1.2a), VP diagrams are not included, and their contribution is instead taken into account by an overall factor, included at NNLO. To calculate it, three routines are available, as summarised in Table 8. Furthermore, for the two main processes and some hadronic final states, PHOKHARA has the option to simulate narrow resonances and their decay into kaon, pion and muon pairs, also listed in the aforementioned table. This allows the generator to produce a description of the experimental data at the peaks of the resonances, as explored in [333].

In the modified version of PHOKHARA 10.0, available in the GitLab repository associated with this manuscript, the generator is now able to incorporate fully customised cuts at the generation level, as well as producing and storing histograms of differential cross sections versus user defined variables. Finally, the four-momenta and weights of all events can be stored in a `.csv` file for further analysis.

## 4.7   Sherpa

SHERPA is a general-purpose Monte Carlo event generator which was originally developed to model high-multiplicity processes at the LHC. Since its inception, the generator has been under active development and recent developments include the addition of dedicated modules for the simulation of processes at lepton colliders, in addition to many other physics improvements. SHERPA includes two inbuilt matrix-element generators, AMEGIC [334] and COMIX [335, 336]. These allow the automated generation of tree-level matrix elements in the complete Standard Model.

A number of QED radiation methods are implemented in SHERPA. Primarily, there are two implementations of the YFS resummation: the PHOTONS module for final-state radiation in decays [156, 337, 338], and the YFS module for full QED radiation from the initial and final states in lepton–lepton collisions [154]. These two modules have been stringently tested and are in very good agreement with each other and with PHOTOS [339]. To produce the results in Section 5, the YFS method has been used to resum all QED IR divergences to all orders (see Section 3.2.3 for a detailed description).

This resummation method can be enhanced further by including higher-order corrections. Two main approaches exist for incorporating these higher-order effects. The first approach, called exclusive exponentiation (EEX), adheres to the framework outlined in the original YFS paper. This approach constructs $\tilde{\beta}_{ij}$ through analytic differential distributions derived from the corresponding Feynman diagrams. These expressions are typically expressed in terms of products of four-vectors; at lower orders, this frequently involves Mandelstam variables or similar quantities.

A key advantage of the EEX method is its relatively straightforward implementation, as well as the ease with which one can verify the behaviour of these expressions in specific limits, such as soft

| Order | ISR Corrections | FSR Corrections | Reference |
|:---:|:---:|:---:|:---:|
| $\tilde{\beta}_0^0$ | Born | Born | |
| $\tilde{\beta}_1^1 + \tilde{\beta}_0^1$ | $\mathcal{O}(\alpha, \alpha L)$ | $\mathcal{O}(\alpha, \alpha L)$ | Eq. (13) |
| $\tilde{\beta}_2^2 + \tilde{\beta}_1^2 + \tilde{\beta}_0^2$ | $\mathcal{O}(\alpha^2 L^2)$ | $\mathcal{O}(\alpha^2 L)$ | Eq. (18,19) |
| $\tilde{\beta}_3^3 + \tilde{\beta}_2^3 + \tilde{\beta}_1^3$ | $\mathcal{O}(\alpha^3 L^3)$ | — | Eq. (26) |

Table 9: The explicit beta terms that have been implemented in SHERPA's EEX. The reference column provides the equation number from [290] where the explicit form of the corrections are given.

or collinear limits. Additionally, constructing terms independently for ISR and FSR contributions simplifies the automation and implementation of ISR effects, particularly for lepton colliders (such as $e^+e^-$ or $\mu^+\mu^-$). However, due to the potential complexity of the final states, automating FSR contributions is challenging and generally requires handling each case individually. The EEX corrections we have included were computed in [130, 143, 340–344], and explicit expressions for the infrared-subtracted terms can be found in [290]. We summarise these corrections in Table 9.

The code is available to download at

https://gitlab.com/sherpa-team/sherpa

# 5 Monte Carlo comparisons

In this section we present results obtained with the Monte Carlo codes described in Section 4. To this end, we define a set of scenarios, i.e., simplified setups consisting of acceptance cuts, reminiscent of the experiments described in Section 2. The purpose is not to precisely match the experimental analyses carried out. Rather, we want to provide simplified but still realistic phenomenological scenarios that can be used to validate the codes and assess the importance of the various contributions that contribute to differential cross sections. The source codes that have been used to obtain these results can be found at

https://radiomontecarlow2.gitlab.io/monte-carlo-results/

Furthermore, these results can also be used to benchmark future theoretical developments. It is foreseen that the repository is updated if new theoretical computations become available.

Throughout this section we use the same colour coding of the seven Monte Carlo codes, as shown in Figure 28. However, for most scenarios not all codes have provided results. Some codes are specialised either to scan $2 \to 2$ or to radiative $2 \to 3$ processes. Furthermore, sometimes particular final states or VPC have not (yet) been implemented.

The errors that are indicated in the plots are only the statistical Monte Carlo errors. We obtain rough indications of the expected theoretical errors by comparing differences in various approaches and approximations. However, a reliable estimate of theoretical error is beyond the scope of the present article. In order to disentangle the impact of input parameters from other differences, in this section we use a standard implementation of the HVP and pion form factor, as discussed in Section 5.1.

In the process of preparing these results we have also carried out numerous technical validations, by comparing identical results produced by different codes.

## 5.1 Input values and observables

We consider the processes

$$e^+ e^- \to X^+(p_+) \, X^-(p_-) \tag{5.1a}$$

$$e^+ e^- \to X^+(p_+) \, X^-(p_-) \, \gamma(p_\gamma) \tag{5.1b}$$

with $X \in \{e, \mu, \pi\}$ and momenta $p_\pm = (E_\pm, \vec{p}_\pm)$. The polar and azimuthal angles are denoted by $\theta_\pm$ and $\phi_\pm$ and we sometimes use the notation $\mathsf{p}_\pm \equiv |\vec{p}_\pm|$. The processes (5.1a) are considered in a CMD-like scenario. The radiative processes (5.1b) are looked at for scenarios related to KLOE, BES III, and B factories. In some of these (tagged cases), the photon is explicitly detected, allowing for cuts on $p_\gamma = (E_\gamma, \vec{p}_\gamma)$. Of course, beyond LO the radiative processes will have more than one photon in the final state. Cuts on $E_\gamma$ below indicate there is at least one photon with energy above this cut. If there are more photons in the final state, we assume they can always be separated. The photon $\gamma(p_\gamma)$ is the hardest photon passing the cut. In other scenarios (untagged cases), the photon is indirectly inferred from missing momentum. In this case we define $\vec{p}_{\widetilde{\gamma}} \equiv -(\vec{p}_+ + \vec{p}_-)$.

Apart from the components of the various momenta we also consider the following quantities:

| | | |
|---|---|---|
| invariant mass | $M_{XX} \equiv \sqrt{(p_+ + p_-)^2} \quad$ or $\quad M_{XX\gamma} \equiv \sqrt{(p_+ + p_- + p_\gamma)^2}$ | (5.2a) |
| (polar angle) acollinearity | $\xi \equiv |\theta^+ + \theta^- - \pi|$ | (5.2b) |
| average polar angle | $\theta_{\text{avg}} \equiv (\theta^- - \theta^+ + \pi)/2$ | (5.2c) |
| azimuthal angle acollinearity | $\left| |\phi^+ - \phi^-| - \pi \right|$ | (5.2d) |

We use on-shell coupling and masses

$$\begin{aligned} \alpha &= 1/137.03599908, \quad m_e = 0.510998950 \,\text{MeV}, \quad m_\mu = 105.658375 \,\text{MeV}, \\ m_\tau &= 1776.86 \,\text{MeV}, \qquad m_\pi = 139.57039 \,\text{MeV} \,. \end{aligned} \tag{5.3}$$

### 5.1.1 Vacuum polarisation

The VP is responsible for the running of the electromagnetic coupling $\alpha(q^2) = \alpha/(1 - \Delta\alpha(q^2))$, and it is an essential ingredient of any radiative corrections calculation. Generalising (3.45) to also include leptonic final states $\Pi^{\text{ren}}(s) = \Pi_\ell^{\text{ren}}(s) + \Pi_h^{\text{ren}}(s)$, the VP can be determined from $e^+e^-$ annihilation cross section using the dispersion relation based on analyticity and unitarity

$$\Pi^{\text{ren}}(s) = \frac{s}{4\pi^2\alpha} \Big[ \text{PV} \int_{4m_l^2}^{\infty} \frac{\sigma^{\text{bare}}_{e^+e^- \to \gamma^* \to X}(s')ds'}{s - s'} - i\pi\sigma^{\text{bare}}_{e^+e^- \to \gamma^* \to X}(s) \Big], \tag{5.4}$$

where the total cross section corresponds to the production of all final states $X$ (including leptons and hadrons) via the one-photon process. The intermediate photon is undressed from the VP contribution itself. In this case, the VP function represents the one-particle irreducible diagram. The Dyson resummation of the VP corresponds to the leading effect of the VPC on the total (say $e^+e^- \to \mu^+\mu^-$) cross section

$$d\sigma_{mm}(q_e^2 \, q_m^2 \, \Pi) = \frac{d\sigma_{mm}(q_e^2 \, q_m^2)}{|1 - \Pi^{\text{ren}}(s)|^2} \,. \tag{5.5}$$

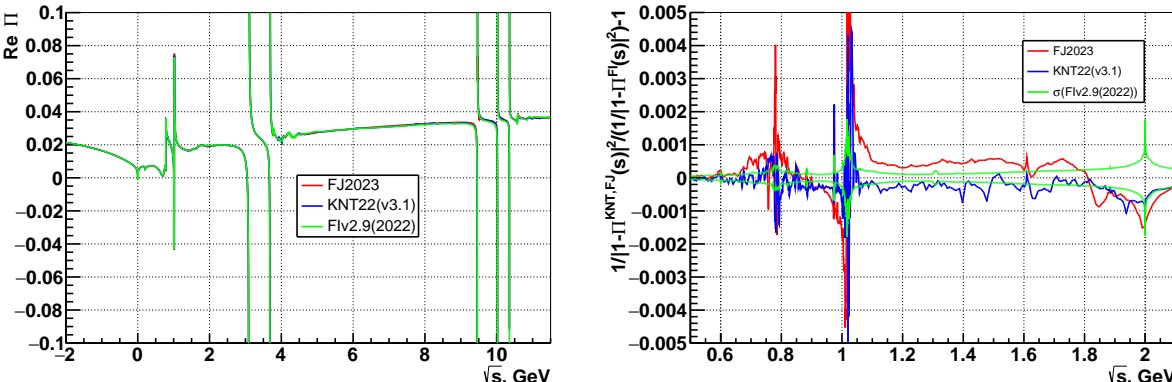

Figure 17: The comparison of the VP function from different packages. The left plot corresponds to $\mathrm{Re}\,\Pi(s)$, where the red line is parameterisation from Jegerlehner (2023), the blue line – KNT v3.1, the green line – NSK VP v2.9. The normalised difference of $1/|1 - \Pi(s)|^2$ relative to the NSK VP is shown on the right plot, where green lines indicate the uncertainty of the NSK VP evaluation.

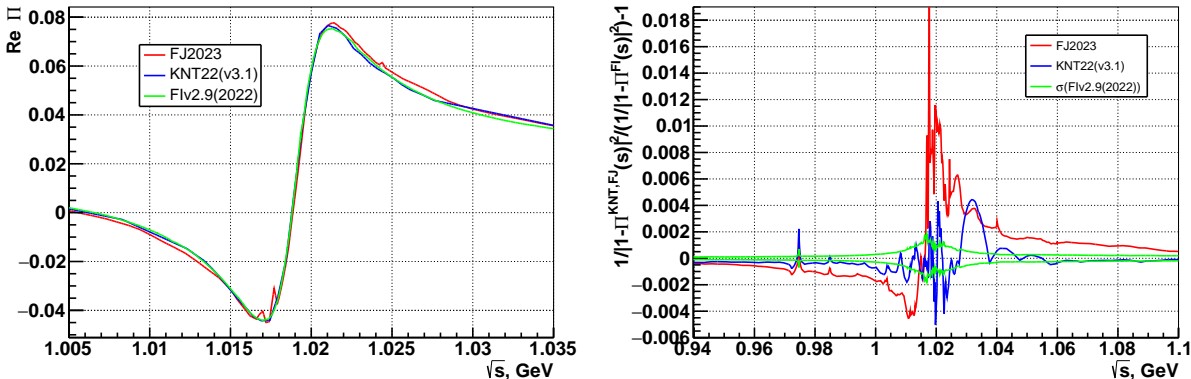

Figure 18: The comparison of the VP function from different packages at the $\phi$ resonance. The left plot corresponds to $\mathrm{Re}\,\Pi(s)$, and right plot the normalised difference of $1/|1 - \Pi(s)|^2$ relative to the NSK VP.

It should be noted that the neglect of the imaginary part of $\Pi(s)$ results in a 1.6% systematic error of the $e^+e^- \to \mu^+\mu^-$ cross section on the peak of the $\phi$ resonance.

While the leptonic part of the VP can be computed in QED as discussed in Section 3.1, the hadronic part can be determined with sufficient precision only by using experimental $e^+e^- \to$ hadrons data at this moment. This requires proper combination and merging of all available experimental $e^+e^-$ datasets, taking into account possible correlations in the systematic uncertainties between different channels. In this subsection we point out some differences between the recent most precise tabulations of the full VP, provided by the KNT group (v3.1, 2022) [166], Jegerlehner (hadr5x23 from `alphaQED23` package, 2023) [167], and Novosibirsk (NSK) VP (v2.9, 2022) [168, 169].

With the current knowledge of the $e^+e^-$ cross sections the precision of the $|1 - \Pi^{\mathrm{ren}}(s)|^2$ normalisation factor is better than 0.05% at c.m. energies below 5 GeV, except at narrow resonances. There the precision is somehow degraded and, for example, the latest combination of experimental data gives 0.2% statistical accuracy at the peak of the $\phi$ resonance. The comparison of different VP compilations is shown in Figure 17. A good consistency is seen outside the $\omega$ and $\phi$ resonances.

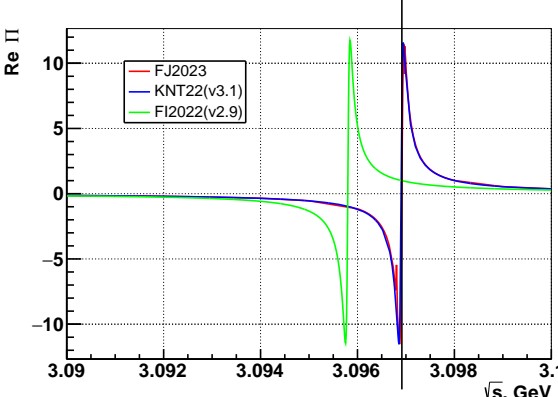

Figure 19: The behaviour of $\mathrm{Re}\,\Pi(s)$ around the $J/\psi$ resonance. The green line corresponds to the proper evaluation using bare resonance parameters, while the blue and red lines were obtained using dressed PDG parameters.

The `alphaQED` version prior to the 2023 release used dressed instead of bare $\phi$ resonance parameters. This resulted in an additional $\sim 2.5\%$ systematic bias of the cross section at these c.m. energies. The latest 2023 release has switched to bare resonance treatment, but it still deviates from KNT and NSK compilations at the level of $-0.4\%$ to $+1\%$ at the left and right side of the $\phi$ peak, as shown in Figure 18. The KNT v3.1 does not have a sufficient number of tabulated points at the fast changing $\phi$ interference, which gives an additional 0.5% error after using linear interpolation between the tabulated points (as seen at $\sqrt{s} = 1.03\,\mathrm{GeV}$ in Figure 18). Both KNT and Jegerlehner's compilations are using dressed PDG mass parameters for resonances like $J/\psi$ and $\psi'$, which gives a much larger effect in theses cases compared to the $\phi$ resonance. For example, the shift from dressed to bare $J/\psi$ mass is about $1.1\,\mathrm{MeV}$, which is much larger than the resonance width itself. The significant changes of the bare full and leptonic widths need to be taken into account as well [168, 345, 346]. The consistent definition of the bare resonance parameters and usage of the resummed VP function leads to the physically observed $\sigma^{\mathrm{bare}}/|1 - \Pi^{\mathrm{ren}}(s)|^2$ cross section on resonances, without convergence problems of the Dyson series which occur when using a real running coupling definition. Improper usage of the dressed resonance parameters in the dispersive integral (5.4) will give unreliable VP energy dependence in the vicinity of them as shown in Figure 19. Using the full VP from such packages is not applicable to energies around narrow resonances. Note that for the KNT compilation, when used close to a very narrow resonance, it was recommended to switch off its particular contribution to the running coupling and instead add the corresponding narrow resonance contribution by hand.

All packages mentioned above include at least the NLO term in the leptonic part of the VP. This is more than sufficient compared to the precision level of the hadronic part. The NSK tabulation of the leptonic $\Pi_\ell(s)$ was calculated numerically through the integral from the total $e^+e^- \to \ell^+\ell^-\gamma$ cross section. This includes the LO FSR correction [249] and the additional enhancement on the threshold from the Coulomb final-state interaction. (the $\Pi_\ell(s)$ by the dispersive integral from the such cross section without the Sommerfeld–Gamow–Sakharov factor gives numerically identical result to the one-two-loop analytical formula of the VP in the paper [109]).

Some Monte Carlo generators still use old implementations of the VP function, which implies additional systematic uncertainties. Comparisons of older VP versions were given in [1]. For example, the original version of KKMC was not supposed to be used at low energies $\sqrt{s} < 2\,\mathrm{GeV}$. The usage of the VP implemented in this package leads up to 10% systematic variations of the total $e^+e^- \to \mu^+\mu^-$ cross section in this case. The original AFKQED generator uses a simplified

evaluation of the VP with precision up to 0.4% of the cross section below $\sqrt{s} < 2\,\text{GeV}$ (except $\phi$-resonance, where it degrades up to 1.8% variation).

In order to disentangle effects from different versions of VP from other differences, we use [169] as default for the hadronic VP for the results presented in this section. However, the procedure of resummation of VPC – or the lack thereof – differs between various codes. These differences are typically beyond NLO, but can be numerically significant, in particular near resonances.

### 5.1.2 Pion form factor

For the pion VFF $F_\pi(q^2)$ we adopt a harmonised parameterisation for all generators. This avoids introducing discrepancies due to different choices of the VFF, which would be already visible at LO. The chosen parameterisation, based on the vector meson dominance (VMD) model [347], is inspired by the fit functions which are usually used by the experiments to fit $F_\pi(q^2)$, for instance in [10, 11, 28]. In particular, we consider the sum of four $\rho$-resonances with the inclusion of the $\rho$–$\omega$ and $\rho$–$\phi$ interference. The explicit parameterisation reads

$$
\begin{aligned}
F_\pi(q^2) = {} & \frac{\text{BW}_\rho^{\text{GS}}(q^2)\left[1 + (q^2/m_\omega^2)\,c_\omega\,\text{BW}_\omega(q^2) + (q^2/m_\phi^2)\,c_\phi\,\text{BW}_\phi(q^2)\right]}{1 + c_{\rho'} + c_{\rho''} + c_{\rho'''}} \\
& + \frac{c_{\rho'}\,\text{BW}_{\rho'}^{\text{GS}}(q^2) + c_{\rho''}\,\text{BW}_{\rho''}^{\text{GS}}(q^2) + c_{\rho'''}\,\text{BW}_{\rho'''}^{\text{GS}}(q^2)}{1 + c_{\rho'} + c_{\rho''} + c_{\rho'''}}
\end{aligned}
\tag{5.6}
$$

The amplitude of each resonance is a complex number, i.e., $c_v = |c_v|e^{i\varphi_v}$. The narrow $\omega$- and $\phi$-resonances are described by a Breit–Wigner (BW) function with a constant width

$$
\text{BW}_v(q^2) = \frac{m_v^2}{m_v^2 - q^2 - im_v\Gamma_v} \qquad v = \omega, \phi
\tag{5.7}
$$

The broad $\rho$ resonances are described by a Gounaris–Sakurai (GS) [348] function

$$
\text{BW}_v^{\text{GS}}(q^2) = \frac{m_v^2 + d(m_v)\,m_v\,\Gamma_v}{m_v^2 - q^2 + f(q^2, m_v, \Gamma_v) - i\,m_v\,\Gamma(q^2, m_v, \Gamma_v)} \qquad v = \rho, \rho', \rho'', \rho'''
\tag{5.8}
$$

where

$$
\Gamma(q^2, m_v, \Gamma_v) = \Gamma_v \frac{m_v}{\sqrt{q^2}}\left[\frac{p_\pi(q^2)}{p_\pi(m_v^2)}\right]^3 \qquad p_\pi(q^2) = \frac{1}{2}\sqrt{q^2 - 4m_\pi^2}
\tag{5.9}
$$

$$
d(m_v) = \frac{3}{\pi}\frac{m_\pi^2}{p_\pi^2(m_v^2)}\log\frac{m_v + 2p_\pi(m_v^2)}{2m_\pi} + \frac{m_v}{2\pi p_\pi(m_v^2)} - \frac{m_v m_\pi^2}{\pi p_\pi^3(m_v^2)}
\tag{5.10}
$$

$$
f(q^2, m_v, \Gamma_v) = \frac{\Gamma_v m_v^2}{p_\pi^3(m_v^2)}\left[p_\pi^2(q^2)\left[h(q^2) - h(m_v^2)\right] + p_\pi^2(m_v^2)\left(m_v^2 - q^2\right)\left.\frac{\text{d}h}{\text{d}q^2}\right|_{q^2=m_v^2}\right]
\tag{5.11}
$$

$$
h(q^2) = \frac{2}{\pi}\frac{p_\pi(q^2)}{\sqrt{q^2}}\log\frac{\sqrt{q^2} + 2p_\pi(q^2)}{2m_\pi} \qquad \frac{\text{d}h}{\text{d}q^2} = \frac{h(q^2)}{8}\left[\frac{1}{p_\pi^2(q^2)} - \frac{4}{q^2}\right] + \frac{1}{2\pi q^2}
\tag{5.12}
$$

The parameter values are inspired by the $F_\pi(q^2)$ measurement of the BaBar, BES III, CMD-2, DM-2, KLOE, and SND experiments. The CMD-3 data are used only to fix the parameters describing the $\phi \to \pi^+\pi^-$ resonance. The chosen numerical values are listed in Table 10, while a comparison between our toy model VFF and experimental data is shown in Figure 20. The parameter values are also chosen to fulfil the dispersive sum rule

$$
\frac{1}{\pi}\int_{4m_\pi^2}^{\infty}\text{d}s'\frac{\text{Im}F_\pi(s')}{s'} = 1
\tag{5.13}
$$

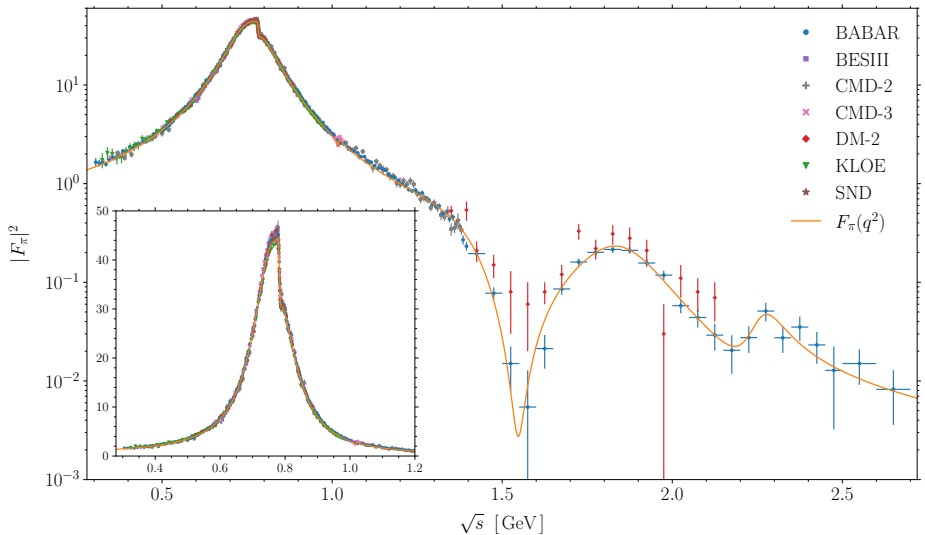

Figure 20: Comparison between our toy model VFF and experimental data.

|  | $\rho$ | $\rho'$ | $\rho''$ | $\rho'''$ | $\omega$ | $\phi$ |
|---|---|---|---|---|---|---|
| $m_v$ (MeV) | 774.56 | 1485.9 | 1866.8 | 2264.5 | 782.48 | 1019.47 |
| $\Gamma_v$ (MeV) | 148.32 | 373.60 | 303.34 | 113.27 | 8.55 | 4.25 |
| $|c_v|$ | - | 0.14104 | 0.0614 | 0.0047 | 0.00158 | 0.00045 |
| $\varphi_v$ (rad) | - | 3.7797 | 1.429 | 0.921 | 0.075 | 2.888 |

Table 10: Input parameters for our model of pion VFF $F_\pi(q^2)$.

and the unitary condition $\mathrm{Im}F_\pi(q^2 < 4m_\pi^2) = 0$ with a permille accuracy.

We do not claim that this expression for $F_\pi(q^2)$ is a proper combination of all experimental data. It is simply a fixed parameterisation inspired by real data and mainly serves the purpose of allowing generator comparisons without impact from VFF variations. Thus, we do not analyse the discrepancy with experimental data or give an error of the VFF parameters. Other parametrisations are possible, such as (3.47), but for the Monte Carlo comparison we adopt the well-known sum of Gounaris–Sakurai functions for simplicity. Although the VFF parameterisation is fixed, each generator implements it following one of the approaches described in Section 3.4, namely factorised sQED, GVMD, and FsQED. While all approaches are equivalent at LO, they can give different results at higher orders. These differences contribute to the theoretical error of the generators.

## 5.2 CMD-like scenario

For the CMD-like scenario we consider the processes $e^+ e^- \to X^+(p_+) X^-(p_-)$ with $X \in \{e, \mu, \pi\}$ at $\sqrt{s} = 0.7\,\mathrm{GeV}$ and apply the following kinematic selection cuts:

$$1\,\mathrm{rad} \leq \theta_\mathrm{avg} \leq \pi - 1\,\mathrm{rad}\,, \tag{5.14a}$$

$$\mathsf{p}_\pm > 0.45 \cdot \sqrt{s}/2\,, \tag{5.14b}$$

$$\left||\phi^+ - \phi^-| - \pi\right| < 0.15\,\mathrm{rad}\,, \tag{5.14c}$$

$$\xi \equiv |\theta^+ + \theta^- - \pi| < 0.25\,\mathrm{rad}\,. \tag{5.14d}$$

In the following subsections we give an example for all three cases $X \in \{e, \mu, \pi\}$.

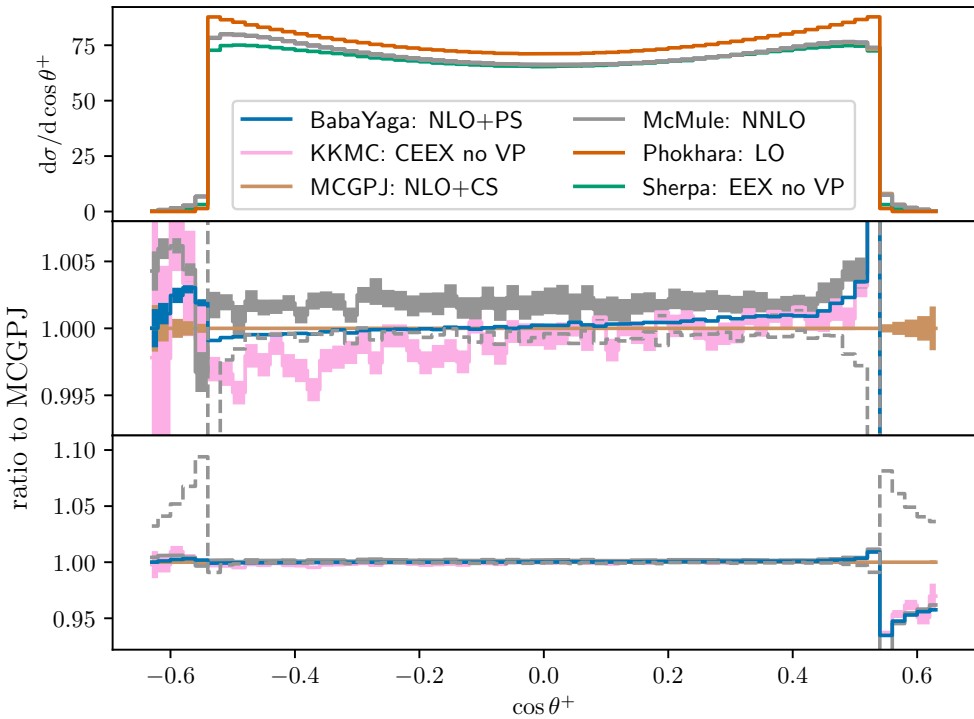

Figure 21: Distribution of the $\mu^+$ angle in the CMD-like scenario for $e^+ e^- \to \mu^+ \mu^-$. The dashed MCMULE line shows the fixed-order NLO result.

### 5.2.1 Muon final state

There are several codes that provide results for $e^+ e^- \to \mu^+ \mu^-$ and we depict their results for the $\cos\theta^+$ distributions in the top panel of Figure 21. Since PHOKHARA is not intended for $2 \to 2$ processes, we use its LO result to illustrate the effect of higher-order corrections. If mixed corrections are neglected, i.e., for LO and the SHERPA result, the distribution is symmetric w.r.t. $\cos\theta^+ = 0$. The middle panel shows the ratio of the more specialised codes to the MCGPJ result. KKMC and SHERPA do not include VPC, neither leptonic nor hadronic. However, at this particular energy it happens that the impact of VPC at NLO is minimal. This is due to the near vanishing of $\Pi(\sqrt{s} = 0.7\,\mathrm{GeV})$, see Figure 17. Hence, a comparison of KKMC with codes including VPC is still meaningful. In the bulk of the distribution where the LO result does not vanish, there is an agreement within less than 0.5% of all specialised codes. The difference between the grey dashed line (MCMULE NLO) and the grey band (MCMULE NNLO) indicates the impact of full NNLO corrections. At NNLO, VPC are not smaller any longer than photonic corrections. In fact, purely photonic NNLO corrections are negative for most values of $\cos\theta^+$. Only by adding NNLO terms with a VP insertion (which amounts to an effect of about 0.5%) the result with a positive total NNLO effect shown as the grey band is obtained. The full NNLO corrections amount to a few permille in the bulk, but are much larger at the boundaries. This is visible in the lower panel (a zoom out of the middle panel), with the NLO far off the parton-shower improved results at the edges of the distribution. However, the NNLO calculation (which in fact is only accurate at NLO in this region) reproduces the bulk of the parton shower. This is an indication that the parton shower is dominated by one additional emission.

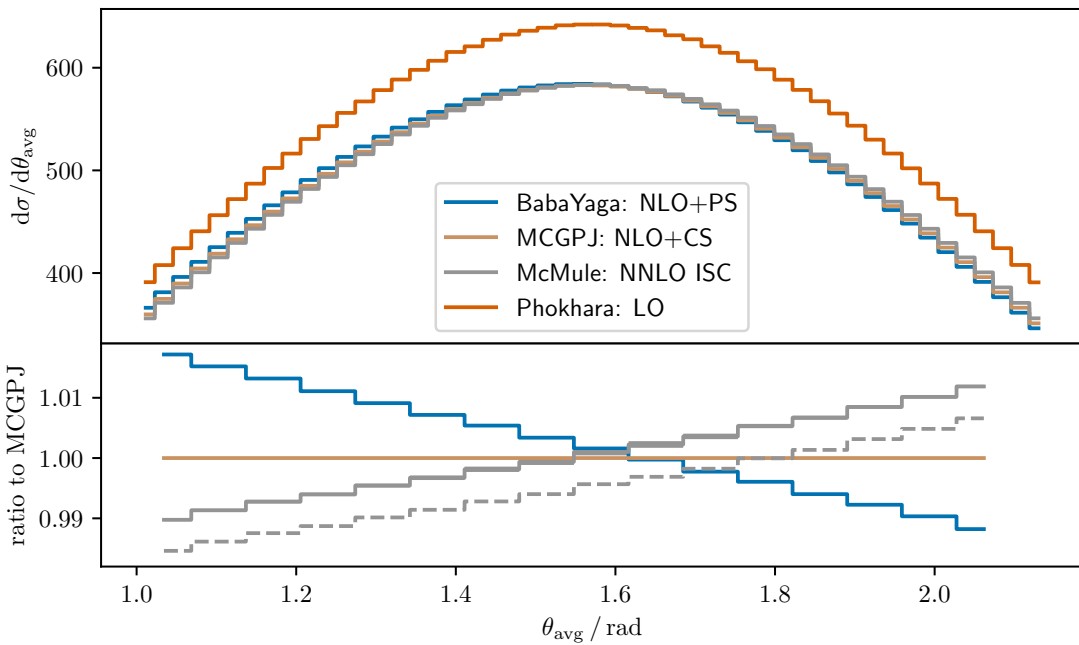

Figure 22: Distribution of $\theta_{\mathrm{avg}}$ in the CMD-like scenario for $e^+ e^- \to \pi^+ \pi^-$. The dashed MCMULE line shows the NLO result (with ISC only).

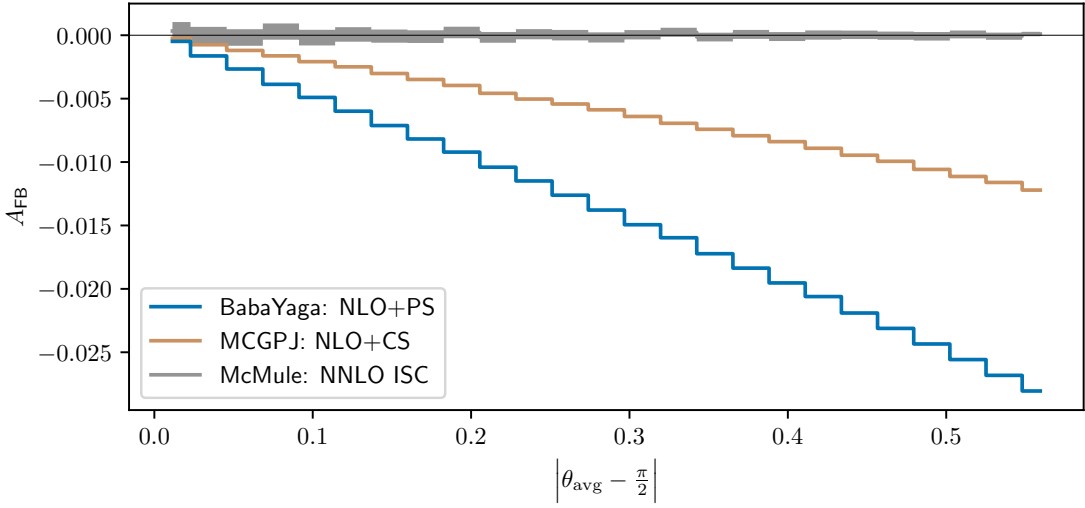

Figure 23: The asymmetry w.r.t. $\theta_{\mathrm{avg}}$ in the CMD-like scenario for $e^+ e^- \to \pi^+ \pi^-$.

### 5.2.2 Pion final state

The reliability of theoretical predictions for $e^+ e^- \to \pi^+ \pi^-$ is unfortunately not at the same level as for $e^+ e^- \to \mu^+ \mu^-$. This is related to the difficulties of treating radiation off pions, as described in Section 3.4. The two codes that provide a full NLO result combined with additional radiation – MCGPJ and BABAYAGA@NLO – differ by more than 1%. This is shown in Figure 22 where we take $\theta_{\mathrm{avg}}$ as an example. Again, we misuse PHOKHARA to provide a LO distribution to indicate the size of higher-order corrections. McMULE includes NNLO corrections, but only ISC. Hence, its $\theta_{\mathrm{avg}}$ distribution is symmetric w.r.t. $\theta_{\mathrm{avg}} = \pi/2$. This is particularly evident in Figure 23 that

shows the forward-backward asymmetry w.r.t. $\theta_{\mathrm{avg}}$

$$A_{\mathrm{FB}} = \frac{\frac{\mathrm{d}\sigma}{\mathrm{d}\theta_{\mathrm{avg}}}\left(\theta_{\mathrm{avg}} > \frac{\pi}{2}\right) - \frac{\mathrm{d}\sigma}{\mathrm{d}\theta_{\mathrm{avg}}}\left(\theta_{\mathrm{avg}} < \frac{\pi}{2}\right)}{\frac{\mathrm{d}\sigma}{\mathrm{d}\theta_{\mathrm{avg}}}\left(\theta_{\mathrm{avg}} > \frac{\pi}{2}\right) + \frac{\mathrm{d}\sigma}{\mathrm{d}\theta_{\mathrm{avg}}}\left(\theta_{\mathrm{avg}} < \frac{\pi}{2}\right)} \,. \tag{5.15}$$

Both full NLO codes produce an asymmetry, as expected, but they differ substantially. This is related to the different treatment of 2PE contributions. MCGPJ uses GVMD, whereas the BabaYaga@NLO results displayed here are obtained with F×sQED. As discussed in Section 3.4.2, the asymmetry has been studied extensively, also in the FsQED approach. While this leads to a reasonable agreement with GVMD, there are notable differences w.r.t. sQED. A recent comparison can also be found in [283]. We have to conclude that in the current codes for fully differential distributions of the process $e^+ e^- \to \pi^+ \pi^-$, there are similar aspects as for the asymmetry. Hence, different options of treating the 2PE contributions lead to a difference above the percent level. As we will see, these problems get more severe in the case of processes with additional photon radiation.

### 5.2.3 Electron final state

The situation for Bhabha scattering $e^+ e^- \to e^+ e^-$ is similar to muon pair production, but there are fewer codes that have implemented this process. Still, we have a full NNLO calculation of McMule and two complete NLO calculations with additional radiation from MCGPJ and BabaYaga@NLO.

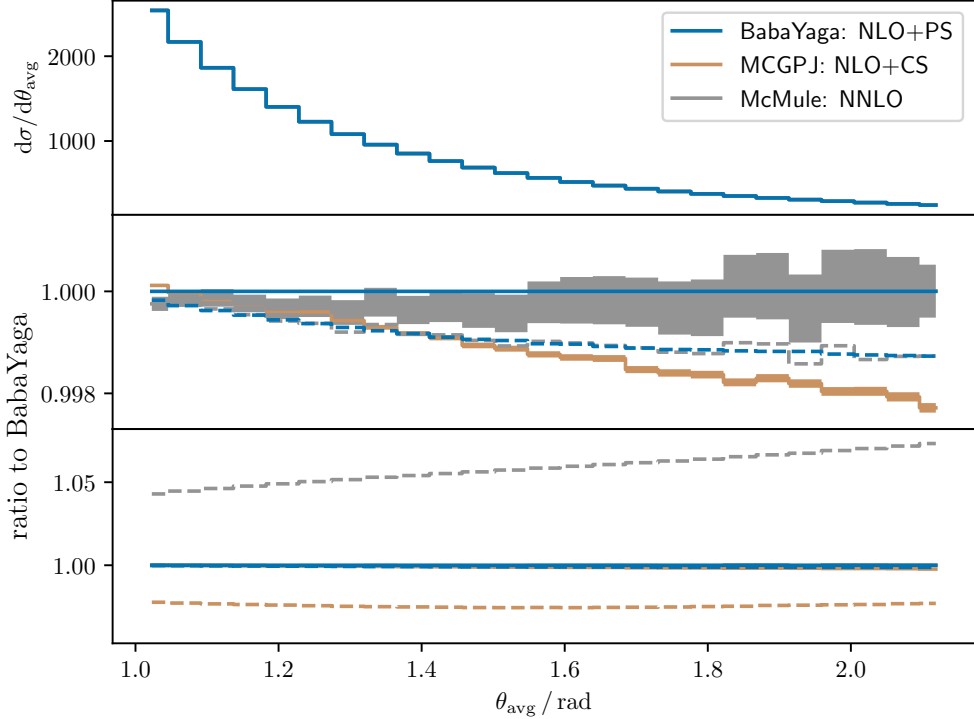

Figure 24: Distribution of $\theta_{\mathrm{avg}}$ in the CMD-like scenario for $e^+ e^- \to e^+ e^-$. In the lower panel we show the LO result (dashed McMule line) and the NLO+CS result without VP (dashed MCGPJ line). In the middle panel the dashed lines show the full NLO result.

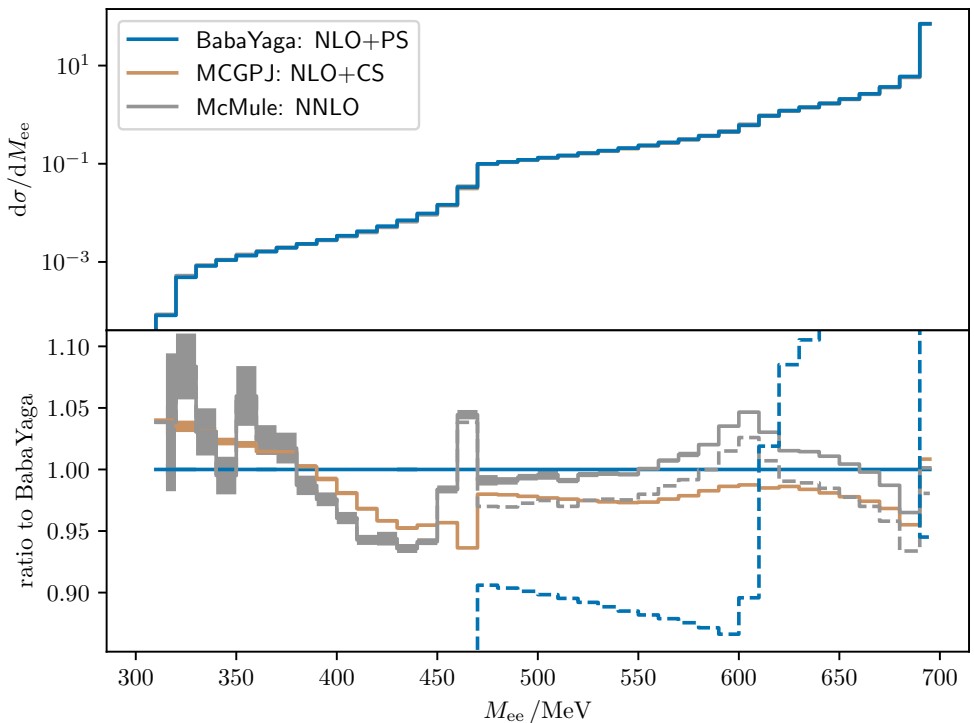

Figure 25: Distribution of the invariant mass of $M_{ee}$ in the CMD-like scenario for $e^+ e^- \rightarrow e^+ e^-$. In the lower panel we also show the NLO result (dashed BABAYAGA@NLO line) as well as the NNLO result without VP contributions (dotted McMULE line).

We first consider an observable that is non-vanishing in the full range, namely $\theta_{\text{avg}}$. The results of Figure 24 show an excellent agreement between the three codes. From the lower panel we can infer that NLO corrections are of the order of 5% whereas VPC at NLO amount to $1-2\%$. Thus, in contrast to $e^+ e^- \rightarrow \mu^+ \mu^-$, VPC are not negligible in this case. The $t$-channel VPC are not affected by the accidental suppression mentioned in Section 5.2.1. In the middle panel we zoom in to an accuracy of a few permille. First, there is perfect agreement between the NLO results of BABAYAGA@NLO and McMULE, depicted as dashed lines. The full NNLO corrections of McMULE are of the order of 0.1% and are almost perfectly reproduced by the additional parton-shower emission in BABAYAGA@NLO. The MCGPJ result also agrees within 0.2%. However, compared to the NLO result, it has a different sign. MCGPJ includes the interference between ISR and FSR only at NLO, whereas BABAYAGA@NLO resums them. Still, we can conclude that for well-behaved observables, non-vanishing at LO, the specialised codes for Bhabha scattering produce results with an error well below 0.5%.

The situation is more complicated for observables with a restricted range at LO. As an example we consider the invariant mass of the two final-state electrons, $M_{ee}$. At tree level, the distribution is a delta peak at $M_{ee} = \sqrt{s}$. Higher-order corrections produce a tail. Thus, the NLO computation produces a LO result for $M_{ee} < \sqrt{s}$. This is illustrated in Figure 25, where the bottom panel shows that the NLO result (dashed blue line) deviates up to 10% for moderately large values of $M_{ee}$ w.r.t. the parton-shower improved results. For smaller values of $M_{ee}$, the NLO result is completely unreliable. However, the NNLO computation reproduces the parton shower for the whole range of $M_{ee}$ within a few percent. The difference between the additional radiation

of BABAYAGA@NLO and MCGPJ is of similar size. A theoretical description of $M_{ee}$ over the whole range at the percent level would require to combine full NNLO corrections with a parton shower. The need of a complete NNLO computation is also confirmed by the impact of NNLO VPC (indicated by the grey dashed line) that are of the order of 2% in the tail.

## 5.3   KLOE-like small-angle scenario

The KLOE-like small-angle scenario considers the radiative processes $e^+ e^- \to X^+ X^- \gamma$ where, however, the photon is not directly detected. The angle $\theta_{\widetilde{\gamma}}$ associated with the 'untagged' photon momentum $\vec{p}_{\widetilde{\gamma}} \equiv -(\vec{p}_+ + \vec{p}_-)$ is assumed to be small w.r.t. the beam. The energy is set to $\sqrt{s} = 1.02\,\text{GeV}$.

More precisely, the cuts we apply are

$$\theta_{\widetilde{\gamma}} \leq 15° \quad \text{or} \quad \theta_{\widetilde{\gamma}} > 165° \,, \tag{5.16a}$$

$$0.35\,\text{GeV}^2 \leq M_{XX}^2 \leq 0.95\,\text{GeV}^2 \,, \tag{5.16b}$$

$$50° \leq \theta^{\pm} \leq 130° \,, \tag{5.16c}$$

$$|\mathsf{p}_{\pm}^z| > 90\,\text{MeV} \quad \text{or} \quad \mathsf{p}_{\pm}^{\perp} > 160\,\text{MeV} \,, \tag{5.16d}$$

where $\mathsf{p}_{\pm}^z$ and $\mathsf{p}_{\pm}^{\perp}$ denote the $z$ and transverse components of the charged final-state particles. In the following subsections we give an example for the cases $X \in \{\mu, \pi\}$ .

### 5.3.1   Muon final state

In Figure 26 we show the $\theta^+$ distribution results for the process $e^+ e^- \to \mu^+ \mu^- \gamma$ of six codes. We do not include the AFKQED result, because the LO photon generated with the exact amplitude is almost never the main ISR photon after selection cuts and it lacks soft and virtual corrections, which would be needed in this case. In addition to the distribution and the ratios to PHOKHARA (middle panel) we also show the normalised distributions in the lower panel to focus on the shape. Indeed, at LO the $\theta^+$ distribution is symmetric w.r.t. $\theta^+ = 90°$, but interference effects beyond LO induce an asymmetry. The two NLO calculations, MCMULE and PHOKHARA agree well, with minor differences due to the treatment of VPC. BABAYAGA@NLO and MCGPJ both include collinear effects using parton showers on top of an LO calculation. However, in the case of MCGPJ, this does not include the angular distribution of the extra photons leading to the differences between the two codes. We note that neither BABAYAGA@NLO nor MCGPJ are designed to be used for this process. KKMC and SHERPA both resum soft emissions to all orders with YFS, the former includes higher-order perturbative corrections using the CEEX formalism and the latter with EEX. There is a significant difference between the two due to the coherence effects that are included in CEEX. In the normalised ratio plot, KKMC agrees well with the NLO codes and BABAYAGA@NLO, whereas SHERPA and MCGPJ have the same shape as the LO result (dashed PHOKHARA line). Also the middle panel clearly illustrates the difference in the shapes. KKMC (with currently no VPC implemented) agrees very well with the NLO result without VPC (dashed MCMULE line). Since VPC amount to roughly 5% their inclusion is essential.

### 5.3.2   Pion final state

For the process $e^+ e^- \to \pi^+ \pi^- \gamma$ we consider the invariant-mass distribution $M_{\pi\pi}$ with the results shown in Figure 27. Also in this case the AFKQED result suffers from the problem of the photon generation being incompatible with the selection cuts. MCGPJ is also shown for illustration only as it is not designed for radiative processes. MCMULE includes ISC at NLO, but does not have any

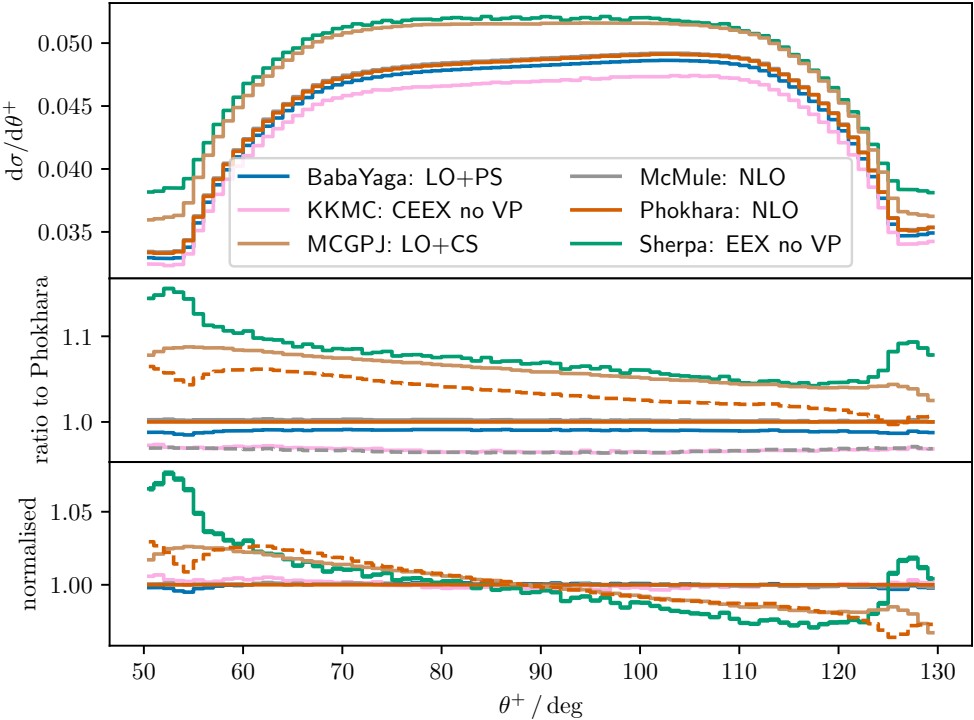

Figure 26: Distribution of the $\mu^+$ angle in the KLOE-like small-angle scenario for $e^+ e^- \to \mu^+ \mu^- \gamma$. The dashed PHOKHARA line is the LO result, the dashed McMule line is the NLO result without VPC. The bottom panel shows the ratio of the normalised distributions.

additional radiation off the pions. At LO, the neglect of FSC has minimal impact in this scenario, as can be seen from the comparison of the (overlapping dashed) PHOKHARA and McMULE LO results in the bottom panel. However, at NLO there is a significant difference of the order of 2%. The difference between the parton-shower improved LO results of BABAYAGA@NLO and PHOKHARA are of similar size. Hence, to obtain reliable results with an error at or below the percent level, it is imperative to include photon radiation also from the final state pions. Given that currently there is no code that includes the structure-dependent radiative corrections (see Section 3.4.2) in a model-independent way, it is prudent to assume a sizeable inherent uncertainty in the NLO corrections of this process.

## 5.4   KLOE-like large-angle scenario

This scenario with $\sqrt{s} = 1.02 \, \text{GeV}$ assumes that at least one photon with energy $E_\gamma > 20 \, \text{MeV}$ is detected. Thus, we are dealing with radiative processes $e^+ e^- \to X^+ X^- \gamma$ and the acceptance cuts are

$$E_\gamma > 20 \, \text{MeV} \qquad \text{and} \qquad 50° \leq \theta_\gamma \leq 130° \, , \qquad (5.17a)$$

$$|\mathsf{p}^z_\pm| > 90 \, \text{MeV} \ \text{or} \ \mathsf{p}^\perp_\pm > 160 \, \text{MeV} \qquad \text{and} \qquad 50° \leq \theta^\pm \leq 130° \, , \qquad (5.17b)$$

$$0.1 \, \text{GeV}^2 \leq M^2_{XX} \leq 0.85 \, \text{GeV}^2 \, , \qquad (5.17c)$$

where (5.17a) is to be understood that there is at least one such photon.

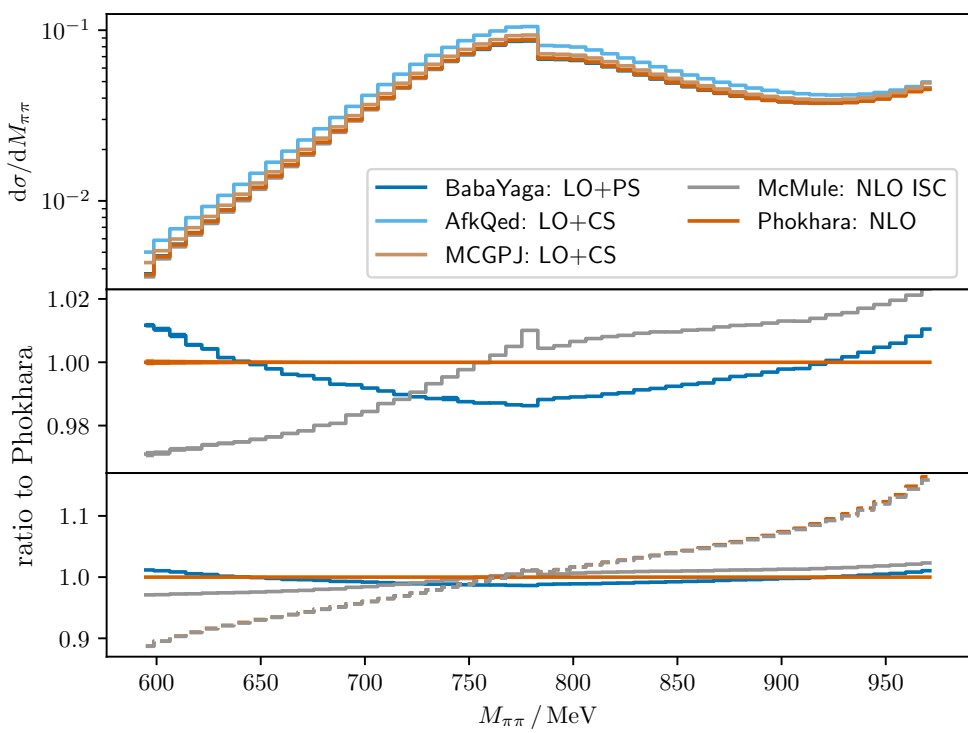

Figure 27: Distribution of $M_{\pi\pi}$ in the KLOE-like small-angle scenario for $e^+ e^- \to \pi^+ \pi^- \gamma$. The dashed lines are the LO results, with (PHOKHARA) and without (MCMULE) final-state photon emission.

### 5.4.1 Muon final state

In the upper panel of Figure 28 we show the invariant-mass distribution for all seven codes, even though MCGPJ is not designed to be used for radiative processes. The remaining six codes – including BABAYAGA@NLO which is also not designed for this process – show reasonable agreement within a few percent. The main difference is due to the lack of VPC in KKMC and SHERPA. To illustrate the impact of VPC we show the perfectly agreeing PHOKHARA and MCMULE NLO results without any VPC as dashed lines in the lower panel. The dashed BABAYAGA@NLO line includes leptonic but not hadronic VPC. We can infer that VPC amount to several percent and their inclusion is mandatory. The full NLO PHOKHARA and MCMULE results differ by up to 1%, due to the different inclusion of VPC. MCMULE implements them strictly at NLO, whereas PHOKHARA performs a Dyson resummation of them (see (5.5)). The difference is formally at NNLO. Hence, if an accuracy better than 1% is required, NNLO corrections are needed. The KKMC result does not include VPC and agrees within $1-2\%$ with the corresponding PHOKHARA and MCMULE results. Also BABAYAGA@NLO with VPC agrees within about 2% with the full NLO calculations. In AFKQED, PHOTOS for NLO final-state radiation is switched off, so that the LO photon is always detected as the main ISR photon after selection cuts. The cross section is dominated by the region $M_{\mu\mu} \gtrsim 800 \,\mathrm{MeV}$, where the differences between the various codes are about 1%. However, typically the full distribution is required and for smaller values of $M_{\mu\mu}$ the differences are larger.

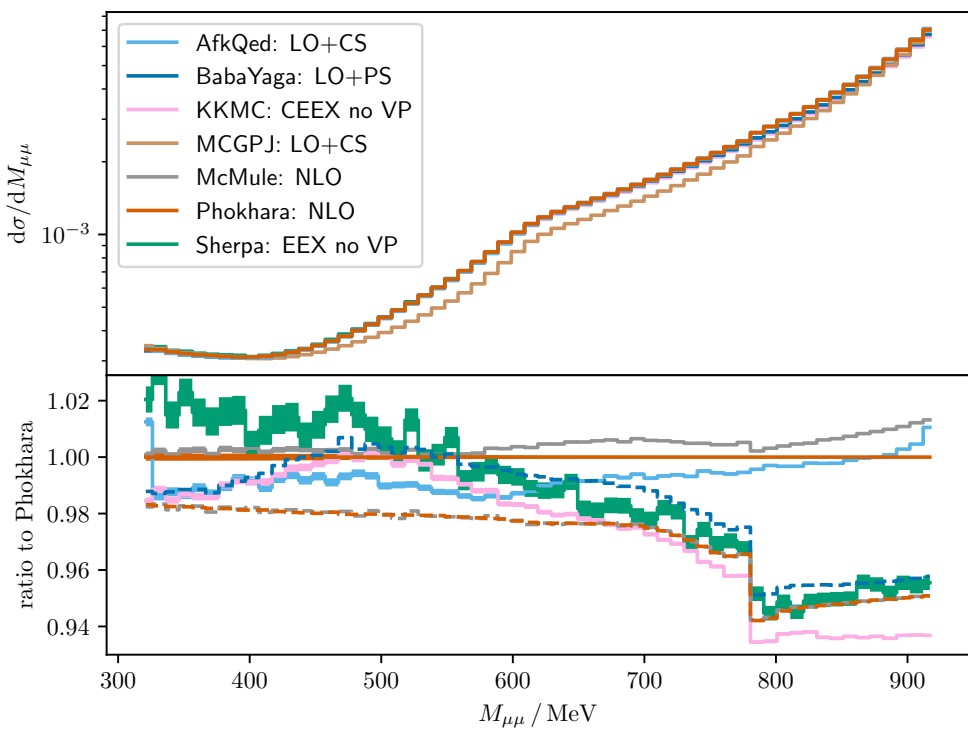

Figure 28: Distribution of $M_{\mu\mu}$ in the KLOE-like large-angle scenario for $e^+ e^- \to \mu^+ \mu^- \gamma$. The dashed BABAYAGA@NLO line shows the LO+PS result with leptonic VP, but no HVP contributions. The (overlapping) dashed PHOKHARA and MCMULE lines show the NLO result without any VPC.

### 5.4.2 Pion final state

Contrary to previous scenarios for the process $e^+ e^- \to \pi^+ \pi^- \gamma$, in the KLOE-like large-angle scenario FSC are very large. This is illustrated in Figure 29, where we show the $\theta^+$ distribution. At LO, PHOKHARA, BABAYAGA@NLO, and MCGPJ contain the full $2 \to 3$ matrix element using F×sQED and agree perfectly. MCMULE and AFKQED only contain initial-state radiation of photons, as PHOTOS is switched off in AFKQED for this scenario. The neglect of FSC at LO leads to an error more than 10%, as can be seen by comparing the dashed MCMULE line with the (overlapping) BABAYAGA@NLO and MCGPJ lines in the lower panel of Figure 29.

The effect is enhanced at NLO. AFKQED and MCMULE only contain ISC and initial-state radiation. Within this approach, the results of these two codes agree as can be seen by the (overlapping) lines in Figure 29. PHOKHARA on the other hand has a full NLO result using F×sQED. The FSC and mixed corrections included in PHOKHARA lead to differences of up to 30%, in particular for small values of $\theta^+$. The large effect of FSC and mixed corrections in the KLOE-like large-angle scenario is also confirmed for the process $e^+ e^- \to \mu^+ \mu^- \gamma$, where they can be computed reliably in QED. Hence, it is clear that a reliable theoretical prediction of this scenario has to include FSC and mixed corrections in a solid framework. In this respect, the lack of a coherent description of structure-dependent radiative corrections in the currently available codes is concerning.

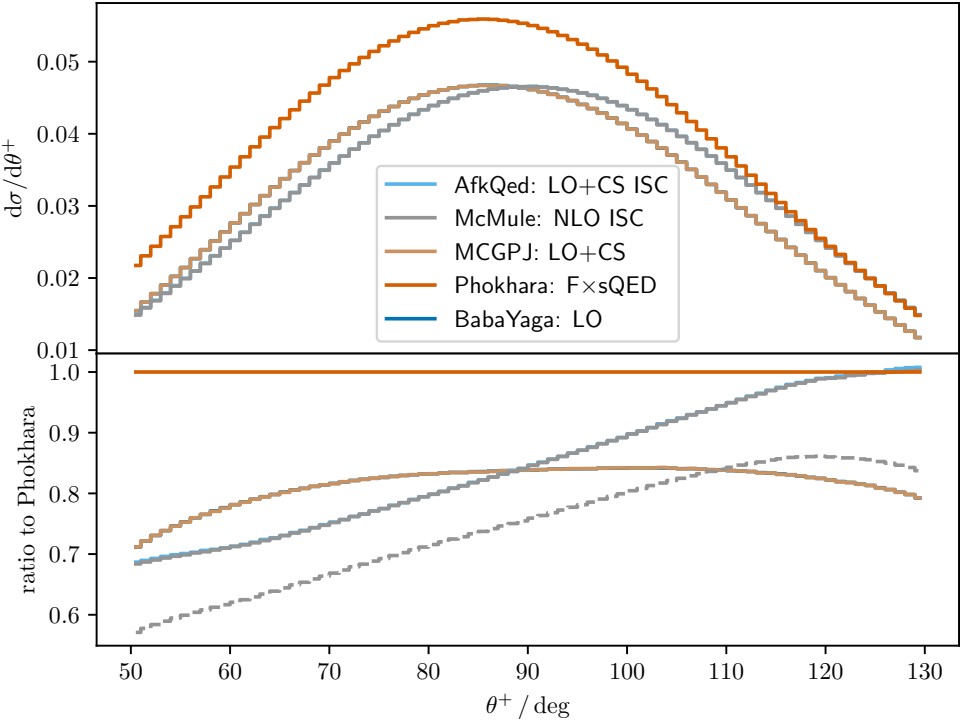

Figure 29: Distribution of $\theta^+$ in the KLOE-like large-angle scenario for $e^+ e^- \to \pi^+ \pi^- \gamma$. The dashed McMule line shows the LO result.

## 5.5 BESIII-like scenario

In this scenario we deal with radiative processes at an energy of $\sqrt{s} = 4\,\mathrm{GeV}$. In order to detect a charged particle $X^\pm$ or photon, they have to pass the selection cuts

$$|\cos\theta^\pm| < 0.93 \qquad \text{and} \quad \mathsf{p}_\pm^\perp > 300\,\mathrm{MeV}\,, \tag{5.18a}$$
$$\left(|\cos\theta_\gamma| < 0.8 \text{ and } E_\gamma > 25\,\mathrm{MeV}\right) \quad \text{or} \quad \left(0.86 < |\cos\theta_\gamma| < 0.92 \text{ and } E_\gamma > 50\,\mathrm{MeV}\right) \tag{5.18b}$$

In addition, we require that precisely one such photon has $E_\gamma \geq 400\,\mathrm{MeV}$.

MCGPJ is not meant to be used for this scenario and is only shown for illustration.

### 5.5.1 Muon final state

For the process with a muon pair in the final state we present a comparison of the codes for the results of the invariant-mass distribution $M_{\mu\mu}$ without VPC. This is to be understood as a technical comparison. Indeed, as is evident from the grey dashed lines in the lower two panels of Figure 30, the VPC are larger than 2% and, therefore, not negligible. The NLO corrections are larger than 10% in some regions of the distribution. Still, the NLO results (without VPC) of Phokhara, AfkQed, KKMC, and McMule agree within about 1%. As for previous scenarios, a theoretical description of $e^+ e^- \to \mu^+ \mu^- \gamma$ with an accuracy better than 1% requires the inclusion of NNLO corrections in a systematic way.

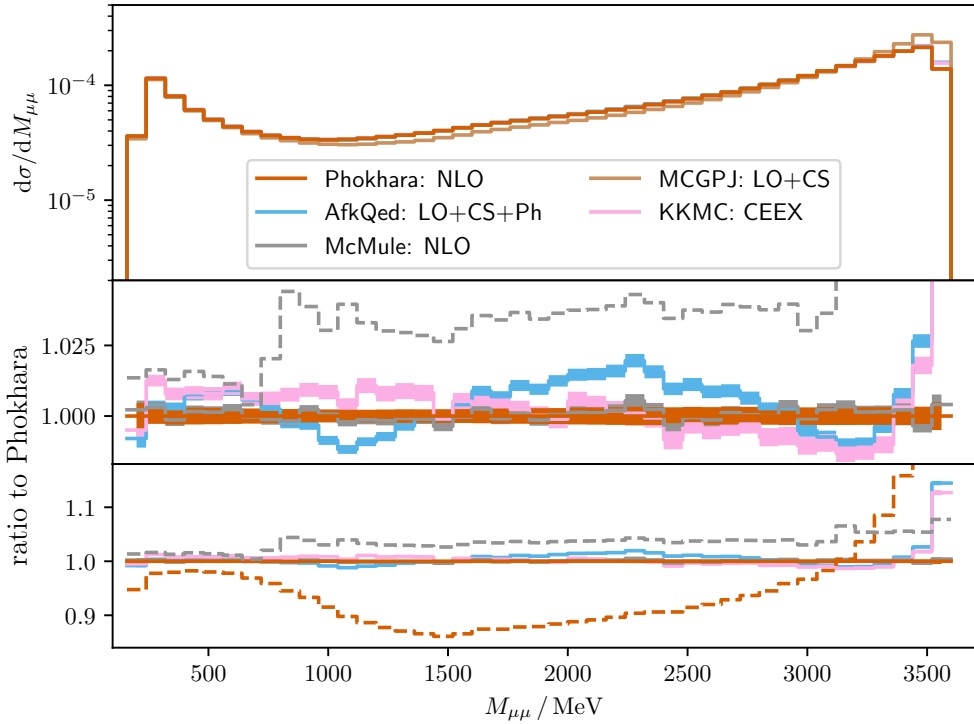

Figure 30: Distribution of $M_{\mu\mu}$ in the BESIII-like scenario for $e^+e^- \to \mu^+\mu^-\gamma$ without VP contributions. For comparison, the LO (dashed PHOKHARA line) and NLO with VP (dashed MCMULE line) results are also shown.

### 5.5.2 Pion final state

As for the KLOE-like scenarios, the computation of $e^+e^- \to \pi^+\pi^-\gamma$ leads to the issue of how to include radiation off the final-state pions. However, in the BESIII-like scenario, the ISC are dominant. In particular, the LO results for the $|\cos\theta^+|$ distribution of PHOKHARA and MCMULE agree within 0.1%, as shown by the dashed lines in the lowest panel of Figure 31, even though MCMULE only includes ISC, whereas PHOKHARA has a complete computation using F×sQED. Part of this suppression of FSR is due to the enhancement of initial-state collinear emission, which can also be noted in the process with a muon pair in the final state. However, a much larger effect is the form-factor suppression. In the case of FSR, the VFF is evaluated at $q^2 = s$, whereas in the case of ISR, the typical $q^2$ is considerably smaller. The sharp fall of $F_\pi(q^2)$ with increasing $q^2$ therefore leads to a strong dominance of ISR relative to FSR.

The NLO corrections amount to roughly 5%. Again, PHOKHARA presents a full NLO result in the F×sQED framework. MCMULE only includes ISC, whereas AFKQED combines complete ISC with FSC implemented through PHOTOS. The impact of FSC is larger for the NLO cross section. But the various results agree to within 1%, except for large values of $|\cos\theta^+|$. It appears that the collinear approximation used by AFKQED for ISC is leading to an underestimation of events in this region.

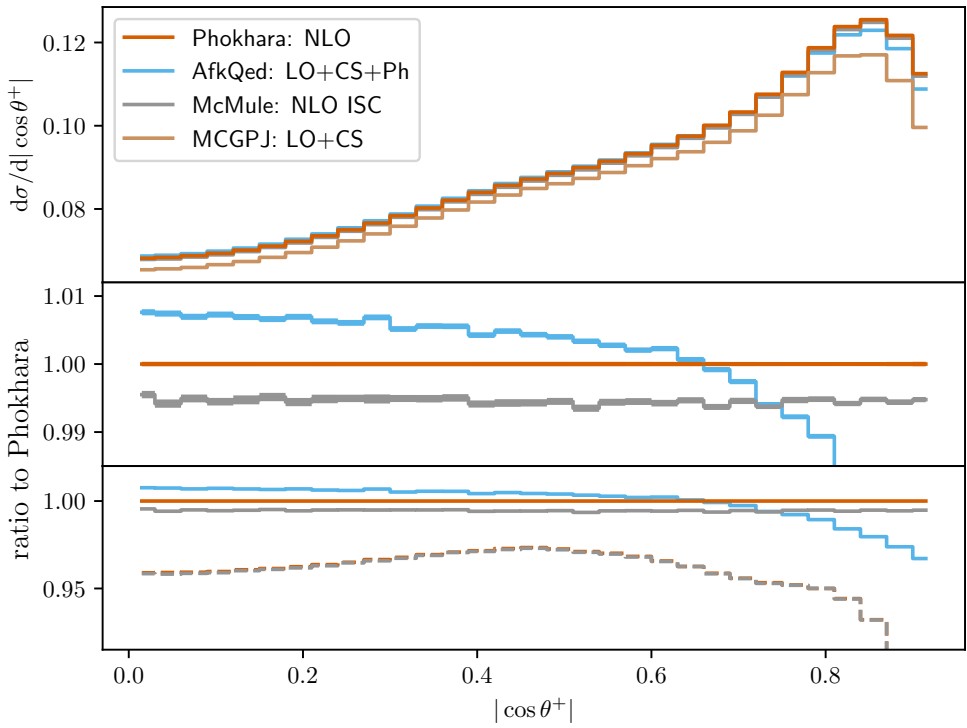

Figure 31: Distribution of $\cos\theta^+$ in the BESIII-like scenario for $e^+e^- \to \pi^+\pi^-\gamma$. The (overlapping) dashed PHOKHARA and MCMULE lines show the LO results with and without final-state radiation.

## 5.6  *B* scenario

This scenario is inspired by $B$ factories. We consider the radiative processes $e^+e^- \to X^+X^-\gamma$ with $X \in \{e, \mu, \pi\}$ at $\sqrt{s} = 10\,\mathrm{GeV}$ with symmetric beams. Consequently, all selection cuts are understood to be in the centre-of-mass frame. In order to be detected, we require for the charged particles and photons

$$0.65\,\mathrm{rad} \leq \theta^\pm \leq 2.75\,\mathrm{rad} \qquad \text{and} \qquad \mathsf{p}_\pm > 1\,\mathrm{GeV}\,, \tag{5.19a}$$

$$0.6\,\mathrm{rad} \leq \theta_\gamma \leq 2.7\,\mathrm{rad} \qquad \text{and} \qquad E_\gamma > 3\,\mathrm{GeV}\,. \tag{5.19b}$$

Furthermore, denoting the most energetic photon passing the cut (5.19b) by $\gamma^{(h)}$ and introducing $M^2_{XX\gamma} \equiv (p_+ + p_- + p_{\gamma^{(h)}})^2$ we require

$$\theta_{\gamma^{(h)},\widetilde{\gamma}} = \sphericalangle(\vec{p}_{\gamma^{(h)}}, \vec{p}_{\widetilde{\gamma}}) < 0.3\,\mathrm{rad} \qquad \text{and} \qquad M_{XX\gamma} > 8\,\mathrm{GeV}\,. \tag{5.19c}$$

The second cut in (5.19c) is to suppress secondary photons. In the case of $X = e$ we also demand

$$M_{ee} > 0.3\,\mathrm{GeV} \tag{5.19d}$$

while for $X \in \{\mu, \pi\}$ no cut on the invariant mass of the charged final-state pair is made.

### 5.6.1  Muon final state

In Figure 32 we show the $M_{\mu\mu}$ distribution for $e^+e^- \to \mu^+\mu^-\gamma$. In the main plot we drop the VPC to perform a technical comparison of photonic corrections in various approaches. In fact,

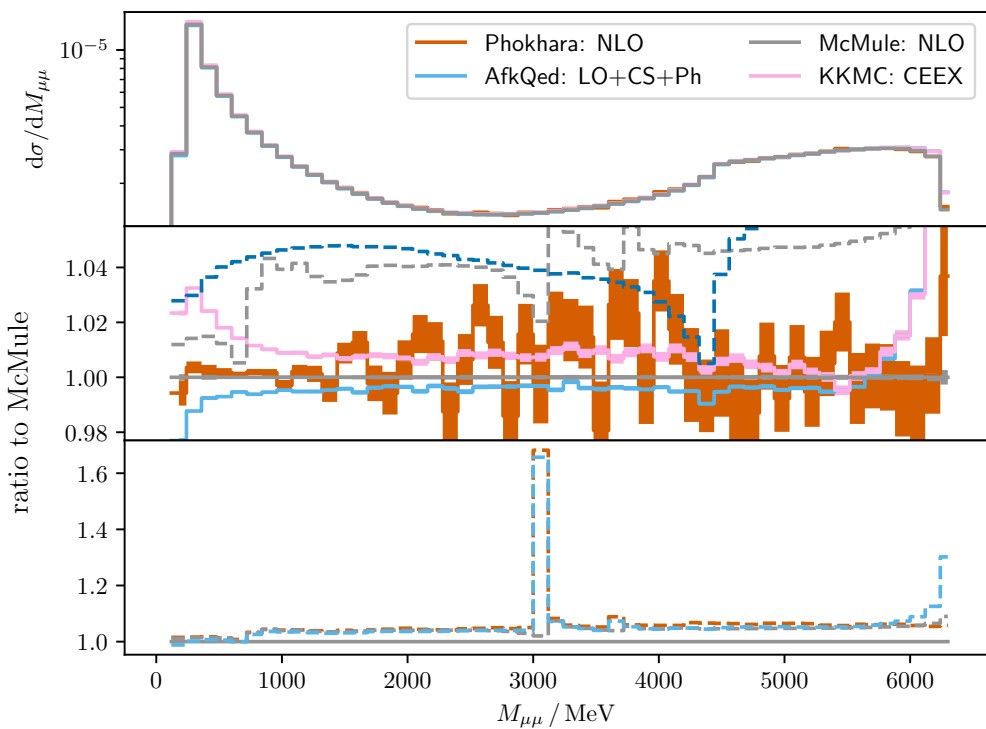

Figure 32: Distribution of $M_{\mu\mu}$ in the $B$ scenario for $e^+e^- \to \mu^+\mu^-\gamma$ without VPC. For comparison, the LO (dashed blue line) and NLO with VP (dashed McMule line) are shown in the middle panel. In the bottom panel we show the results with VPC as dashed lines.

the VPC are about as large as the full NLO corrections and, hence, cannot be neglected at all. This is evident from the dashed lines in the middle panel that show the LO (blue) and NLO with VPC (grey) results. The full NLO results of Phokhara and McMule agree perfectly. Since the Phokhara results have rather large statistical fluctuations we show the ratio w.r.t. McMule in the lower panels. As for the BESIII-like scenario, AfkQed includes the first initial-state emission exactly and combines this with the collinear approximation for further ISC and Photos for FSC. This agrees within 1% with the exact NLO result, except at the very high end of the tail, where multiple emission of photons might be particularly relevant. Indeed, the AfkQed result there agrees well with KKMC, which also includes multiple emission through CEEX. The cross section is dominated by small values of $M_{\mu\mu}$, where there are differences larger than 2% between KKMC and the other codes. These differences are as large as the full photonic NLO corrections.

In the bottom panel we point out the huge impact of the VPC in the region of the $J/\psi$ resonance. The McMule implementation is strictly fixed order NLO, i.e., with one insertion of $\Pi$ in the intermediate photon line. Phokhara and AfkQed resum multiple $\Pi$ insertions through a modified photon propagator, as discussed in Section 5.1.1. Formally, the difference is NNLO, but due to the resonance structure in $\Pi_h$, these differences are numerically extremely large. This region is expected to receive large NNLO corrections also due to the sensitivity to additional soft photon emission.

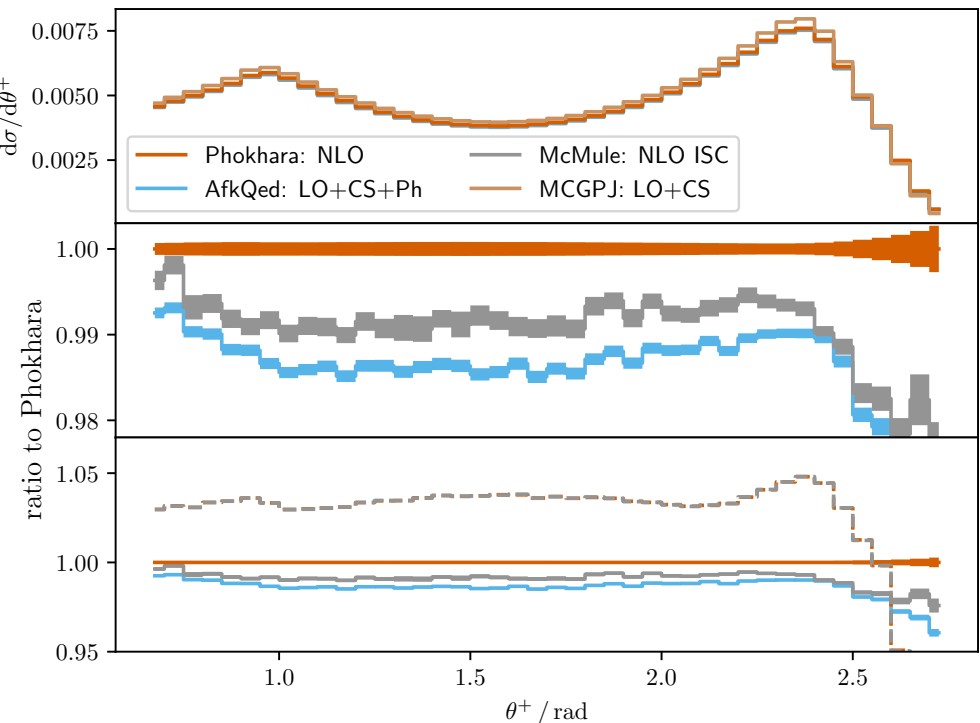

Figure 33: Distribution of $\theta^+$ in the $B$ scenario for $e^+ e^- \to \pi^+ \pi^- \gamma$. The dashed PHOKHARA and McMULE lines in the lower panel show the LO results, with and without final-state radiation.

### 5.6.2 Pion final state

As for previous scenarios, also in the $B$ scenario we consider the $\theta^+$ distribution for the process $e^+ e^- \to \pi^+ \pi^- \gamma$. The comparison of the PHOKHARA and McMULE LO results, shown as dashed lines in the lowest panel Figure 33, reveal that also in this case final-state radiation (included in PHOKHARA but not in McMULE) is strongly suppressed at LO. In fact, the bin-by-bin differences between the LO McMULE and PHOKHARA results are zero within the numerical Monte Carlo error of $\mathcal{O}(10^{-4})$.

At NLO, the difference between the full F×sQED computation of PHOKHARA and the results of McMULE and AFKQED with ISC only (plus PHOTOS for AFKQED) amounts to about 1%. As for the BESIII-like scenario, this is a sizeable fraction of the complete NLO correction that is about 4%. As a preliminary conclusion it could be argued that the PHOKHARA result is sufficient for a description at the $1-2\%$ level. But if a more precise result is required, once more, an improved treatment of structure-dependent corrections would be beneficial.

### 5.6.3 Electron final state

As a final example, in Figure 34 we show results for $e^+ e^- \to e^+ e^- \gamma$, namely the $\theta^+$ distribution. For this process, currently only McMULE provides a fixed-order NLO result. Comparing the LO and NLO results in the bottom panel reveals NLO corrections of about 10%. There are also rather large differences compared to the results of MCGPJ and BABAYAGA@NLO, which provide parton-shower improved LO results. However, it should be kept in mind that these codes have not

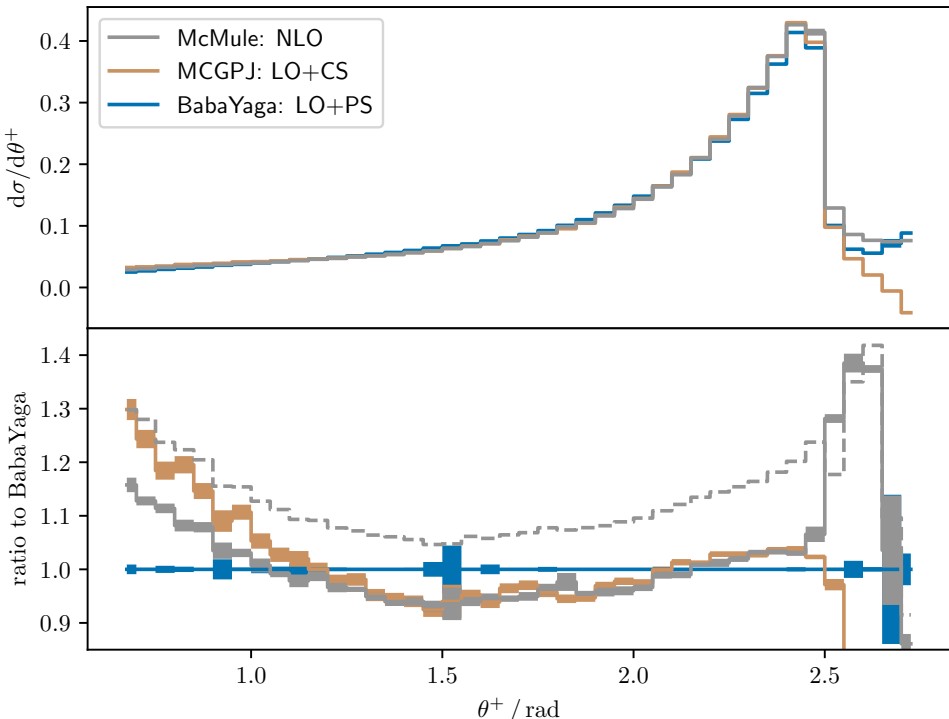

Figure 34: Distribution of $\theta^+$ in the $B$ scenario for $e^+ e^- \to e^+ e^- \gamma$. The dashed McMule line shows the LO result.

been designed for radiative processes. Still, a reliable theoretical description of radiative Bhabha scattering at the percent level in this scenario does require to include corrections beyond NLO.

# 6   Outlook

The work reported in this document is meant to be Phase I of an ongoing community effort to improve the theoretical description of scattering processes at electron–positron colliders. We have set up a framework to coordinate the activities of several groups working on Monte Carlo codes, on hadronic physics, and on experimental aspects related to such processes. At this stage, we have focused on the core processes (1.1) and (1.2). We have made comparisons between various codes, either as technical validation, or as an investigation into the impact of particular contributions included in some but not all of the codes. All codes that have been used to obtain the results presented in this report are publicly available and further developments are foreseen. These developments concern improvements for the core processes as well as extensions of the scope.

First, several improvements in perturbative QED computations are feasible with currently available techniques. This includes NNLO QED corrections to ISC, i.e., to $e^+ e^- \to \gamma \gamma^*$ and even to the full $2 \to 3$ processes $e^+ e^- \to \mu^+ \mu^- \gamma$ and $e^+ e^- \to e^+ e^- \gamma$. While it is a formidable task to obtain the two-loop integrals with full mass dependence required for the $2 \to 3$ processes, there are well established procedures to obtain sufficiently precise approximations for mass effects at NNLO. Furthermore, such fixed-order calculations can in principle be combined with dominant effects beyond NNLO through a parton shower or YFS resummation. Hence, for the pure QED

part it is a realistic goal to achieve a status that removes any doubts about a precision well below the percent level for arbitrary differential distributions.

The path towards such a precision for processes with pions in the final state is much steeper. While there is in principle a systematic way forward with a dispersive approach, currently there is no dispersive study of the $2\pi3\gamma^{(*)}$ amplitude. Hence, at a practical level, there is a need to compromise. This involves the comparison of different approaches and approximations. A broader activity and cross talk between groups working on Monte Carlo implementations and dispersive studies will be instrumental for making further progress.

The improvements mentioned above are closely linked to obtaining not only a good theory description, but also a reliable theory error. This is a notoriously difficult aspect and there is no unambiguous procedure. However, with an increased number of approaches and a deeper understanding of their strengths and limitations, a more reliable error estimate is one of the future goals of this effort.

Concerning additional extensions, an obvious next step is also to enlarge the list of processes to be considered. In addition to $e^+e^- \to \pi^+\pi^-\pi^0$ only briefly discussed in this report,

$$e^+e^- \to \pi^+\pi^-\pi^0\pi^0\,, \qquad\qquad e^+e^- \to \pi^+\pi^-\pi^+\pi^-\,, \tag{6.1a}$$

$$e^+e^- \to \pi^0\gamma\,, \qquad\qquad e^+e^- \to \pi^+\pi^-\pi^0\gamma\,, \tag{6.1b}$$

$$e^+e^- \to K^+K^-\,, \qquad\qquad e^+e^- \to K_L K_S\,, \tag{6.1c}$$

$$e^+e^- \to \tau^+\tau^-\,, \qquad\qquad e^+e^- \to \gamma\gamma\,, \tag{6.1d}$$

are among the processes of particular interest. So far, only in a small subset of codes these processes have been implemented. Since (most of them) suffer from the complications related to hadronic final states, it would be useful to extend this activity and contrast different implementations. Again, this is related to the determination of a reliable theory error.

The observables presented in this report are fairly simple in that they only include generic scenarios with some acceptance cuts. However, the Monte Carlo tools provided in connection with this work can and are being used for full experimental analyses. Through this community effort we also offer the possibility for a close collaboration between the experimental collaborations and the developers of the theoretical tools. We hope that this is of mutual benefit and also triggers further developments on the theory side. This will not only advance the physics of low-energy electron–positron collisions, but also have an impact for improved Monte Carlo tools for other experiments, such as lepton–proton scattering, the electron–ion collider, or even a future circular electron–positron collider.

## Acknowledgements

We thank V. Druzhinin and H. Czyż for their help in setting up and running AFKQED and PHOKHARA, respectively. CCC and FP, together with EB and FU, wish to thank G. Montagna and O. Nicrosini for the continuous collaboration on the development of BABAYAGA@NLO. We further thank the Mainz Institute for Theoretical Physics (MITP) of the DFG Cluster of Excellence PRISMA$^+$ (Project ID 39083149), for supporting the topical workshop "The Evaluation of the Leading Hadronic Contribution to the Muon $g-2$: Consolidation of the MUonE Experiment and Recent Developments in Low Energy $e^+e^-$ Data," and the University of Zurich for support of the 5th Workstop/Thinkstart "Radiative corrections and Monte Carlo tools for Strong 2020". These meetings were instrumental in moving this project forward.

**Monte Carlo contacts**

The following authors are responsible for the codes: PB and LC for AFKQED, CCC and AG for BABAYAGA@NLO, JP and AnS for KKMC, FI for MCGPJ, SK and MR for MCMULE, PPR and WTB for PHOKHARA, and AP and LF for SHERPA. This does not necessarily mean that they developed the code.

**Funding information**

The work in this report is supported by the Leverhulme Grant LIP-2021-014, the STFC (grants ST/P001246/1, ST/T000988/1, and ST/X000699/1), the DFG through the funds provided to the Sino-German Collaborative Research Center TRR110 "Symmetries and the Emergence of Structure in QCD" (DFG Project-ID 196253076 – TRR 110), the SNSF (Project Nos. 200020_200553, 200020_207386, PCEFP2_181117, PCEFP2_194272, and TMSGI2_211209), the Italian Ministero dell'Università e Ricerca (MUR) and European Union - Next Generation EU through the research grants 2022BCXSW9, 20225X52RA, and 2022ENJMRS under the programme PRIN 2022, the European Union STRONG2020 project under Grant Agreement Number 824093, the Polish National Science Centre through the Grants Nos. 2019/35/O/ST2/02907 and 2023/50/A/ST2/0022, and by the Priority Research Area Digiworld under the programme "Excellence Initiative – Research University" at the Jagiellonian University in Krakow.

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
