# Peer review of "Radiative corrections and Monte Carlo tools for low-energy hadronic cross sections in $e^+ e^-$ collisions"

_SciPost Physics Reviews_

## Round 2 · Referee Report · Anonymous (Referee 1) · 2025-3-17

Strengths

This is important to review and list the programs which may be useful for experimental data analysis as well as list people involved in program development or just managing codes.

Collecting short descriptions of the codes and their comparisons can be helpful for the comunity

Weaknesses

1-, general theoretical introduction is missing.:
2-,In particular what is the relation between vacuum polarization and bremsstrahlung. It should be mentioned, including comments on dispersion relations, especially in context of effects beyond first order.
3-, Logarithms (at least some of them) appear from phase space integration over the regions of detector sensitivity. This should be mentioned and consequences for observable build indicated. Especially for the one involving detector granularity. Known since long limitations of fixed order calculations (cancelation of infrared virtual and real emission infinities, should be at least mentioned and better explained.
4-, these are the most important points. Some recall of references, in particular to work of Denner should be mentioned in context of general principles of use of dispersion relations and separation of predictions for QED and hard interaction part (effectively running of alpha_QED). Some of this is covered in programs descriptions. It would be OK to list the topics and write that every program has the issues covered by the authors.
5-) It would be nice if authors of individual programs contributions would
extend descriptions of their theoretical basis, but I would not expect this to be possible.

Report

After modifications and statement that responsibility of particular overall ambiguities has to be consulted with the authors of individual programs contributions, and that in general ambiguities are bound to be observable dependent, report may be accepted.

Requested changes

1.) see point 1 of weakness
2.) see point 2 of weakness
3.) see point 3 of weakness
4.) see point 4 of weakness

Recommendation

Ask for major revision

---

## Editorial Decision

resubmitted